

# Generalized symmetries and anomalies of 3d $\mathcal{N} = 4$ SCFTs

Lakshya Bhardwaj[1], Mathew Bullimore[2],
Andrea E. V. Ferrari[2] and Sakura Schäfer-Nameki[1]

**1** Mathematical Institute, University of Oxford,
Woodstock Road, Oxford, OX2 6GG, United Kingdom
**2** Department of Mathematical Sciences, Durham University,
Upper Mountjoy, Stockton Road, Durham, DH1 3LE, United Kingdom

## Abstract

We study generalized global symmetries and their 't Hooft anomalies in 3d $\mathcal{N} = 4$ superconformal field theories (SCFTs). Following some general considerations, we focus on good quiver gauge theories, comprised of balanced unitary nodes and unbalanced unitary and special unitary nodes. While the global form of the Higgs branch symmetry group may be determined from the UV Lagrangian, the global form of Coulomb branch symmetry groups and associated mixed 't Hooft anomalies are more subtle due to potential symmetry enhancement in the IR. We describe how Coulomb branch symmetry groups and their mixed 't Hooft anomalies can be deduced from the UV Lagrangian by studying center charges of various types of monopole operators, providing a concrete and unambiguous way to implement 't Hooft anomaly matching. The final expression for the symmetry group and 't Hooft anomalies has a concise form that can be easily read off from the quiver data, specifically from the positions of the unbalanced and flavor nodes with respect to the positions of the balanced nodes. We provide consistency checks by applying our method to compute symmetry groups of 3d $\mathcal{N} = 4$ theories corresponding to magnetic quivers of 4d Class S theories and 5d SCFTs. We are able to match these results against the flavor symmetry groups of the 4d and 5d theories computed using independent methods. Another strong consistency check is provided by comparing symmetry groups and anomalies of two theories related by 3d mirror symmetry.



# 1  Introduction

Gauge theories in three spacetime dimensions are extremely interesting to study from a theoretical viewpoint. Since the gauge coupling has positive mass dimension, any gauge theory can be given an ultraviolet (UV) complete definition, but in the infrared (IR) the effective gauge coupling becomes strong, opening up the possibility of interesting strong coupling behaviour. The case of 3d gauge theories with eight supercharges, i.e. $\mathcal{N}=4$ supersymmetry, has been very well studied in this context, where it is known that with enough matter the gauge theory flows in the IR to a 3d $\mathcal{N}=4$ superconformal field theory (SCFT).

There are many interesting non-perturbative phenomena that arise in the context of $\mathcal{N}=4$ supersymmetry. Of these, the most well-known phenomenon is that of 3d mirror symmetry [1–4] which relates two different 3d $\mathcal{N}=4$ gauge theories such that the corresponding IR 3d $\mathcal{N}=4$ SCFTs are the same, up to the exchange of Coulomb and Higgs branches.

Arguably the most interesting aspect of 3d $\mathcal{N} = 4$ supersymmetric gauge theories and mirror symmetry is symmetry enhancement in the IR. The flavor symmetries of 3d $\mathcal{N} = 4$ gauge theories arise from hyper-Kähler isometries of the Higgs and Coulomb branch moduli spaces. While the Higgs branch and associated symmetries may be understood classically, the Coulomb branch receives 1-loop and non-perturbative corrections and the hyper-Kähler metric depends on the gauge coupling. This allows for the emergence of additional hyper-Kähler isometries and associated symmetry enhancement on the Coulomb branch in the IR SCFT. This phenomenon plays a fundamental role in mirror symmetry. This symmetry enhancement was systematically explained in [5], using techniques developed in [6–8] based on the study of monopole operators [9, 10].

In this paper, we extend the discussion of symmetries in 3d $\mathcal{N} = 4$ supersymmetric gauge theories to include generalized symmetries [11], including global aspects of traditional 0-form symmetries, 1-form symmetries, 2-groups symmetries and discrete 't Hooft anomalies.

A key result is to identify the monopole operators in the UV gauge theory that allow a determination of the global form of the IR Coulomb branch 0-form symmetry group. The result matches the global form of the Higgs branch 0-form symmetry group of the mirror UV gauge theory, which can be computed classically from the matter content pf the mirror gauge theory without considering its monopole operators.

Once the global form of the IR Coulomb 0-form symmetry group is known, an important question is to determine the 't Hooft anomalies of the Coulomb 0-form symmetry with the Higgs 0-form symmetry. In three space-time dimensions, such anomalies are necessarily discrete. The general structure of such anomalies for 3d gauge theories was explored in our previous work [12]. This captured the information about the anomaly in terms of flavor charges carried by mixed flavor-gauge monopole operators, which are in general non-genuine local operators that arise at the end points of flavor-gauge vortex line defects. In $\mathcal{N} = 4$ supersymmetric gauge theories, there exist BPS configurations of flavor-gauge monopole operators sitting at the ends of flavor-gauge vortex lines. These configurations thus descend to configurations of local operators sitting at the ends of line operators in the corresponding IR $\mathcal{N} = 4$ SCFTs. During the flow the flavor charges of these non-genuine local operators get mixed, and the mixing can be deduced using the methods of [11]. Finally, reversing the logic of [12], one can use the information about the charges of these non-genuine local operators to deduce the 't Hooft anomaly of the IR SCFT. This provides a concrete and unambiguous way of implementing *'t Hooft anomaly matching* from UV 3d $\mathcal{N} = 4$ gauge theories to IR 3d $\mathcal{N} = 4$ SCFTs.[1]

In a similar fashion, we also determine the 't Hooft anomalies of the IR Coulomb 0-form symmetry with 1-form symmetries. The requisite operators are now what were dubbed fractional gauge monopole operators in [12], which are non-genuine local operators living at the ends of topological line operators generating the 1-form symmetry. For 3d $\mathcal{N} = 4$ gauge theories, there exist BPS configurations of fractional gauge monopole operators living at the ends of topological line operators.[2] These configurations survive in the IR SCFT and the charges of such non-genuine local operators under IR Coulomb symmetry determine the precise form of the mixed 't Hooft anomaly between the Coulomb 0-form symmetry and the 1-form symmetry of the IR 3d $\mathcal{N} = 4$ SCFT. While this work was being written, we received [14] which also used the analysis of [12] to obtain some of the results appearing in sections 3 and 6 of this paper.

The analysis of this paper also opens the door for the determination of symmetries and anomalies of higher-dimensional (i.e. in $d > 3$) SCFTs with at least eight supercharges. This can be done by applying the analysis of this paper to 3d $\mathcal{N} = 4$ *magnetic quivers* (MQs) associated to these higher-dimensional SCFTs. MQs are 3d $\mathcal{N} = 4$ gauge theories whose IR behaviour

---

[1]Note that similar logic of using BPS objects to match symmetry properties between UV and IR was recently employed in [13].

[2]Note that a topological operator automatically preserves all supersymmetry.

captures information about the Higgs branch of vacua of the corresponding higher-dimensional SCFTs. MQs have been a subject of much interest and exploration recently [15–44]. In particular they have been instrumental in studying Higgs branches for 4d $\mathcal{N} = 2$ and 5d $\mathcal{N} = 1$ SCFTs, which are otherwise more difficult to access due to quantum corrections.

In some instances, we expect the symmetries of the 4d or 5d SCFT and that of its 3d MQ theories to agree. This is in particular the case, when the higher-dimensional theory only exhibits 0-form symmetries, which then should then agree with the 0-form symmetries of the MQ theory.[3] We test our proposal by computing the global 0-form symmetries of MQs, and compare them to the global form of flavor symmetries of 4d class S theories and 5d SCFTs, and find agreement.

**Outlook.** This paper opens up many interesting avenues to explore in the connection between generalized symmetries and their 't Hooft anomalies and SCFTs (in particular with 8 supercharges).

As mentioned above, 0-form global symmetries act by hyper-Kähler isometries of Higgs and Coulomb branch moduli spaces of vacua. A natural question is therefore whether there exists such a geometric realization for generalised symmetries such as 1-form symmetries and 2-groups and their 't Hooft anomalies. In forthcoming work [45], we will show that such a realization may be found in the algebraic setting by promoting the Higgs and Coulomb branch moduli spaces to moduli stacks. These stacky enhancements of moduli spaces keep track of unbroken discrete gauge symmetry when flowing to the IR at points on the underlying moduli space and carry actions of generalized symmetries. Moreover, the 't Hooft anomalies for generalised symmetries considered in this paper may be understood geometrically in terms of equivariance properties of distinguished line bundles on these moduli stacks associated to half-BPS line operators.

Another natural generalization in light of the fact that we consider invertible symmetries in 3d, is the extension to non-invertible symmetries. There is a multitude of realizations now. Most relevant for the field theoretic approach that was the focus in this paper are the following constructions, which have direct 3d realizations [46–54]. Non-invertible symmetries in 3d are of course very well explored in the context of modular tensor categories, however here the interesting question is related to the interplay between non-invertible symmetries and superconformal symmetry in 3d. One construction, which relies on the presence of mixed anomalies has been explored in 3d in [46, 49]. To explore these in full it will be useful to characterize systematically the symmetry topological field theories for SCFTs in 3d, as started in [55, 56].

Finally one can extend the considerations of this paper involving mostly continuous 0-form symmetries to also include discrete 0-form symmetries and their associated 't Hooft anomalies. The two types of symmetries in general combine to form a disconnected 0-form symmetry (Lie) group, which when combined with 1-form symmetries generally gives rise to disconnected 2-group symmetries introduced recently in [57].

The paper is organized as follows. In section 2, we provide a discussion of supersymmetric defect operators and generalized symmetries of A/B-type (Coulomb/Higgs in the gauge theory setting) and their 't Hooft anomalies in 3d $\mathcal{N} = 4$ theories. This is an application of our general results in [12] to this supersymmetric setting.

In section 3, we study the simplest example, in great detail, where one can apply the considerations of this paper. This concerns $T[SU(n)]$ theories and related theories that can be obtained by gauging Higgs 0-form symmetry of $T[SU(n)]$.

---

[3]If the higher dimensional theory has higher-form symmetries, then these can in principle contribute to the 0-form symmetry of the 3d MQ theory.

In section 4, we generalize the methods employed in the previous section 3 to study a large class of 3d $\mathcal{N} = 4$ SCFTs that can be obtained in the IR of good 3d $\mathcal{N} = 4$ quiver gauge theories composed of balanced unitary and unbalanced unitary and special unitary gauge groups along with matter hypermultiplets transforming in fundamental and bifundamental representations. We describe how the Coulomb 0-form symmetry groups and its mixed 't Hooft anomalies with 1-form and Higgs 0-forms symmetries can be obtained easily by a visual analysis of the UV quiver and noting the placement of unbalanced and flavor nodes with respect to the positions of balanced nodes.

In section 5, we present a variety of consistency checks of our general results of section 4. We check that the IR Coulomb 0-form symmetry groups of magnetic quivers of Class S theories and 5d SCFTs match the flavor symmetry groups of these higher dimensional SCFTs computed via other methods. We also check that the IR Coulomb 0-form symmetry group matches the Higgs 0-form symmetry group of the 3d mirror gauge theory.

In the final section 6, we study a few interesting theories that lie outside the general class of theories studied in section 4. Our methods presented in section 4 can be easily generalized to include such theories and lead to many interesting phenomena not observed in the class of theories studied in section 4. These include pure 't Hooft anomalies for 1-form symmetry, the existence of 2-group symmetries in 3d $\mathcal{N} = 4$ SCFTs, and mixed 't Hooft anomalies between 2-group and 0-form symmetries of 3d $\mathcal{N} = 4$ SCFTs. The latter two phenomena are exhibited by the 3d $\mathcal{N} = 4$ SCFT called $\text{T}[SU(2)]/\mathbb{Z}_2^C$ that can be obtained from $\text{T}[SU(2)]$ by gauging a $\mathbb{Z}_2$ subgroup of $SO(3)_C$ Coulomb 0-form symmetry of $\text{T}[SU(2)]$, and can be obtained as the IR SCFT corresponding to 3d $\mathcal{N} = 4$ SQED with $U(1)$ gauge group and 2 hypermultiplets of charge 2.

Appendix A provides details on the computation of global forms of flavor symmetry groups of 5d SCFTs from Calabi-Yau threefold singularities.

## 2 Generalized symmetries of 3d $\mathcal{N} = 4$ theories

In this section, we consider general aspects of invertible generalized symmetries in 3d $\mathcal{N} = 4$ supersymmetric theories, including 0-form symmetries, 1-form symmetries and 2-group symmetries as well as their 't Hooft anomalies. For ordinary 0-form symmetries, we distinguish between $R$-symmetries and flavor symmetries. We focus on flavor symmetries, which commute with all of the supercharges, and consider possible 2-group symmetries that combine flavor symmetries and 1-form symmetries.

We will introduce two types of such symmetries called "A-type" and "B-type" depending on which class of BPS operators are charged under them. For continuous 0-form symmetries, this corresponds to the known classification of supermultiplets that conserved currents or background gauge fields for continuous symmetries may transform in, or equivalently central extensions of the supersymmetry algebra. However, we explain how this classification can be applied more broadly to both finite and continuous symmetry groups, in addition to 1-form and 2-group symmetries.[4]

These symmetries may have various 't Hooft anomalies, which we study using the techniques introduced in our previous paper [12]. In particular, we consider "A-type" and "B-type" BPS solitonic local operators and line defects that source background fields for the above symmetries and explain how their properties capture different types of 't Hooft anomaly. We also explain how gauging discrete symmetries interchanges A-type and B-type symmetries in a

---

[4]This does not preclude the existence of additional discrete global symmetries that are not of this type. Examples of such symmetries include outer automorphisms of gauge groups such as charge conjugation or automorphisms of quiver diagrams, and anomalous 1-form symmetries that arise when coupling to a 3d TQFT.

manner compatible with such 't Hooft anomalies and how this leads to examples of mirror symmetry involving generalised symmetries.

Our primary example throughout this section will be standard 3d $\mathcal{N} = 4$ supersymmetric gauge theories built from vectormultiplets and hypermultiplets. In such case, the A-type and B-type symmetries are associated to Coulomb and Higgs branch geometry respectively. It is also possible to consider gauge theories with less supersymmetry that flow to $\mathcal{N} = 4$ supersymmetry in the IR [58–60]. In these cases, the identification of the A-type and B-type symmetries from a UV perspective is more intricate.

## 2.1 BPS defects

We begin with a discussion of BPS local operators, line defects and junctions that will play a role in the classification of flavor symmetries. First recall that a theory with 3d $\mathcal{N} = 4$ supersymmetry has R-symmetry algebra $\mathfrak{so}(4) \cong \mathfrak{su}(2) \oplus \mathfrak{su}(2)$ and supercharges $Q_\alpha^{A\dot{A}}$ where $\alpha$ is a euclidean space-time spinor index and the indices $A, \dot{A}$ denote the spinor representation of the two factors of the *R*-symmetry group.

**Local operators.**    The half-BPS genuine local operators come in two types:

- **A-type**: annihilated by the four supercharges $Q_\alpha^{A\dot{+}}$.

- **B-type**: annihilated by the four supercharges $Q_\alpha^{+\dot{A}}$.

The A-type operators are constructed from the bottom scalar components of vectormultiplets and twisted hypermultiplets, while the *B*-type operators are constructed from the bottom scalar components of hypermultiplets and twisted vectormultiplets. This classification into A-type and B-type applies equally well to non-genuine twisted sector local operators attached to a topological line defect.

The two sets of half-BPS genuine local operators generate two chiral rings $\mathsf{C}_A, \mathsf{C}_B$ whose spectra define complex affine moduli spaces

$$X_A := \mathrm{Spec}(\mathsf{C}_A), \tag{1}$$
$$X_B := \mathrm{Spec}(\mathsf{C}_B). \tag{2}$$

In a standard supersymmetric gauge theory constructed from vectormultiplets and hypermultiplets, they coincide with the Coulomb and Higgs branch respectively, in the absence of resolution or deformation parameters, viewed as complex algebraic varieties.

We might consider the possibility that there are local operators annihilated by all of the supercharges. Such operators are necessarily topological. We will assume that there is a unique (up to multiplication by a complex number) such topological local operator, namely the identity operator. This is tantamount to the statement that the theory is irreducible or equivalently that there are no 2-form symmetries.

In the opposite direction, we may have occasion to consider more general quarter-BPS operators annihilated by two supercharges $Q_\alpha^{+\dot{+}}$. They are half-BPS for the 3d $\mathcal{N} = 2$ supersymmetry algebra generated by $Q_\alpha^{+\dot{+}}, Q_\alpha^{-\dot{-}}$.

**Line operators.**    We consider half-BPS line defects along the $x^3$-axis preserving a 1d $\mathcal{N} = 4$ supersymmetric quantum mechanics sub-algebra of the 3d $\mathcal{N} = 4$ supersymmetry algebra. Such line operators were first introduced in supersymmetric gauge theories in [61] and have been further studied in [62–64].

There are two classes of half-BPS lines:

- **A-type**: annihilated by four supercharges $Q_+^{A\dot{+}}$, $Q_-^{A\dot{-}}$.

- **B-type**: annihilated by four supercharges $Q_+^{+\dot{A}}$, $Q_-^{-\dot{A}}$.

The line defects can be described uniformly by consistent couplings to 1d $\mathcal{N} = 4$ supersymmetric quantum mechanics with super-multiplets obtained by dimensional reduction from 2d $\mathcal{N} = (2,2)$ and $\mathcal{N} = (0,4)$ supersymmetry respectively. Examples include B-type Wilson lines for dynamical vectormultiplets and A-type Wilson lines for dynamical twisted vectormultiplets. We note that this classification applies equally well to half-BPS twisted sector line defects that are attached to a topological surface.

A special case is line defects annihilated by all of the supercharges, which are simultaneously A-type and B-type and therefore necessarily topological line defects. Such line defects are normally considered as generators of 1-form symmetries, but may also be charged under them in the presence of 't Hooft anomalies. This situation may arise when coupling to a general 3d TQFT in a way that preserves $\mathcal{N} = 4$ supersymmetry but does not arise in theories constructed from standard supermultiplets. Incorporating such topological line defects as charged objects will require a refinement of the classification of symmetries presented here and some examples are presented in subsequent sections.

In the opposite direction, we may have occasion to consider more general quarter-BPS line defects preserving the common pair of supercharges $Q_+^{+\dot{+}}$, $Q_-^{-\dot{-}}$. They can be regarded as half-BPS line defects for the 3d $\mathcal{N} = 2$ supersymmetry algebra generated by the supercharges $Q_\alpha^{+\dot{+}}$, $Q_\alpha^{-\dot{-}}$.

**Junctions.** Finally we consider various local junction operators between pairs of line defects. We consider two classes of quarter-BPS junctions between pairs of A-type and B-type lines and preserve two supercharges lying in the intersections of the two sets of four supercharges preserved by genuine local operators and line defects:

- **A-type**: annihilated by two supercharges $Q_+^{A\dot{+}}$.

- **B-type**: annihilated by two supercharges $Q_+^{+\dot{A}}$.

It is also possible to consider local junction operators between a half-BPS A-type and a B-type line defect, or alternatively between a pair of quarter-BPS line defects, which both preserve the single supercharge $Q_+^{+\dot{+}}$.

**Comment on relation to topological twist** The above classification of BPS operators is related but distinct to the classification of operators in topological twists of 3d $\mathcal{N} = 4$ supersymmetry, where A-type and B-type operators are defined as those in the cohomology of the nilpotent supercharges

$$Q_A := Q_+^{+\dot{+}} + Q_-^{-\dot{+}}, \tag{3}$$

$$Q_B := Q_+^{+\dot{+}} + Q_-^{+\dot{-}}. \tag{4}$$

Correspondingly, we are interested only in genuine symmetries generated by extended operators that are topological in the full 3d $\mathcal{N} = 4$ theory, not merely after performing a topological twist.

## 2.2 A- and B-type symmetries

We now consider the classification of invertible flavor symmetries in 3d $\mathcal{N} = 4$ theories. As mentioned above, we assume that the theory is irreducible and therefore restrict ourselves to at most 2-group symmetries. In particular, there is a unique genuine local operator that is simultaneously A-type and B-type, which is the identity operator.

The proposal is then that the most general flavor symmetry is a product of A-type and B-type 2-group symmetries associated to the above classification of BPS defects.

### 2.2.1 0-form symmetry

For continuous 0-form symmetries, it is well known that the flavor symmetry takes the form of a product $\mathcal{F}_A \times \mathcal{F}_B$ for compact Lie groups $\mathcal{F}_A$, $\mathcal{F}_B$. The two factors are known as A-type and B-type symmetry groups.

At the level of the associated Lie algebra $\mathfrak{f}_A \oplus \mathfrak{f}_B$, this decomposition may be understood from the allowed central extensions of the 3d $\mathcal{N} = 4$ supersymmetry algebra. This admits a pair of central charges $Z^{AB}$, $Z^{\dot{A}\dot{B}}$ transforming in the adjoint representations of the two $\mathfrak{su}(2)$ R-symmetries. The central charges are proportional to the generators of the A-type and B-type symmetries respectively with coefficients given by scalar fields $\sigma^{AB}$, $\sigma^{\dot{A}\dot{B}}$ in vectormultiplets and twisted vectormultiplets respectively. In summary, A-type symmetries couple to vectormultiplets and B-type symmetries to twisted vectormultiplets.

However, in order to provide a definition of the flavor symmetry group $\mathcal{F}_A \times \mathcal{F}_B$, which also applies to discrete symmetries, and in addition to formulate obstruction classes that appear in 't Hooft anomalies for these symmetries, it is convenient to define symmetries starting from the BPS operators on which they act.

**Definitions.** The flavor symmetry groups $\mathcal{F}_A$, $\mathcal{F}_B$ are defined as the maximal compact Lie groups with Lie algebras $\mathfrak{f}_A$, $\mathfrak{f}_B$ that act faithfully on A-type and B-type genuine half-BPS local operators respectively. This definition also applies when $\mathfrak{f}_A$, $\mathfrak{f}_B$ are trivial, in which case the flavor symmetry groups are discrete. These symmetry groups (or rather their complexification in the continuous case) will act by complex isometries on the moduli spaces $X_A$, $X_B$.

In the construction of 2-group symmetries involving these flavor symmetries and their 't Hooft anomalies, we will also need to consider non-genuine local operators that sit at the junctions between half-BPS line defects.

Let us consider A-type or B-type half-BPS line defects that preserve the whole symmetry group $\mathcal{F}_A$ or $\mathcal{F}_B$ respectively. Such line defects may then end on A-type or B-type quarter-BPS local operators that transform in representations of central extensions of $\mathcal{F}_A$, $\mathcal{F}_B$ by discrete abelian groups, that are not representations of $\mathcal{F}_A$, $\mathcal{F}_B$. It is convenient to write down the short exact sequences

$$0 \longrightarrow \mathcal{Z}_A \longrightarrow F_A \longrightarrow \mathcal{F}_A \longrightarrow 0 \, , \tag{5}$$

$$0 \longrightarrow \mathcal{Z}_B \longrightarrow F_B \longrightarrow \mathcal{F}_B \longrightarrow 0 \, , \tag{6}$$

where $\mathcal{Z}_A$, $\mathcal{Z}_B$ are finite abelian groups and $F_A$, $F_B$ denotes the extended symmetry groups. Equivalently, we have the quotients $\mathcal{F}_A = F_A / \mathcal{Z}_A$, $\mathcal{F}_B = F_B / \mathcal{Z}_B$. In summary, local operators at the end of line defects may be charged under $\mathcal{Z}_A$, $\mathcal{Z}_B$.

There are associated obstruction classes

$$w_2^A \in H^2(B\mathcal{F}_A, \mathcal{Z}_A) \, , \tag{7}$$

$$w_2^B \in H^2(B\mathcal{F}_B, \mathcal{Z}_B) \, , \tag{8}$$

for lifting $\mathcal{F}_A, \mathcal{F}_B$ bundles to $F_A, F_B$ bundles, which play an important role in the description of 2-groups and 't Hooft anomalies involving these symmetries. In particular, introducing background fields $B_1^A, B_1^B : M \to B\mathcal{F}_A, B\mathcal{F}_B$, there are associated obstruction classes on spacetime via pull-back $(B_1^A)^* w_2^A$, $(B_1^B)^* w_2^B$. In what follows, we will often abuse notation and denote these spacetime obstruction classes also by $w_2^A$, $w_2^B$.

**Gauge theories.** Let us consider standard supersymmetric gauge theories constructed from vectormultiplets and hypermultiplets. In such cases, it is appropriate to replace the monikers $A/B$ by $C/H$, which refer to Coulomb and Higgs respectively.

The $B$-type symmetry $\mathcal{F}_H$ acts faithfully on gauge-invariant combinations of hypermultiplet fields, while the central extension $F_H$ is constructed by examining the charges of non-gauge invariant combinations of hypermultiplet fields attached to B-type half-BPS Wilson lines for the dynamical vectormultiplet.

This is conveniently captured by introducing the structure group $\mathcal{S}$, which captures the combination of gauge and B-type flavor symmetries acting faithfully on all supermultiplets. In other words, the bundles for $\mathcal{S}$ correspond to the most general combination of gauge and B-type flavor symmetry bundles (transforming in dynamical and background vectormultiplets respectively) to which the theory may be consistently coupled. It takes the form

$$\mathcal{S} = \frac{\mathcal{G} \times F_H}{\mathcal{E}}, \tag{9}$$

where $\mathcal{G}$ denotes the gauge group, which we assume is connected, and $\mathcal{E}$ is a subgroup of the center $Z(\mathcal{G} \times F_H)$ of $\mathcal{G} \times F_H$ such that $p_H(\mathcal{E}) = \mathcal{Z}_H$ where $p_H : Z(\mathcal{G}) \times Z(F_H) \to Z(F_H)$ is the natural projection.

On the other hand, the A-type symmetry group $\mathcal{F}_C$ acts faithfully on genuine half-BPS monopole operators. This is the topological symmetry

$$\mathcal{F}_C = \widehat{\pi_1(\mathcal{G})}, \tag{10}$$

where the hat denotes the Pontryagin dual. It measures the topological class of the $\mathcal{G}$-bundles on a sphere surrounding the monopole operator. It may be continuous or discrete. Unlike the B-type symmetry group, this may undergo enhancement at an IR superconformal fixed point. Determining the precise global form of the enhanced symmetry group is a major goal of this paper.

The gauge theory may couple to bundles for the structure group $\mathcal{S}$. Correspondingly, there exist A-type half-BPS line defects corresponding to gauge-flavour vortex lines for the structure group labelled by a co-character $\phi : U(1) \to \mathcal{S}$.

These vortex lines may include fractional gauge vortex lines, which by definition are vortices associated to co-characters for the quotient group

$$G = \mathcal{G}/\mathcal{Z}_g, \tag{11}$$

that do not lift to co-characters for the gauge group $\mathcal{G}$. Here $\mathcal{Z}_g = p_g(\mathcal{E})$ where $p_g : Z(\mathcal{G}) \times Z(F_H) \to Z(\mathcal{G})$ is the natural projection. Such configurations must, in general, be paired with fractional flavour vortex lines to determine a well defined line defect labelled by a co-character for the structure group $\phi : U(1) \to \mathcal{S}$.

The gauge-flavour vortex line defects may then end on non-genuine monopole operators of fractional magnetic charge, which results in a short exact sequence

$$1 \longrightarrow \mathcal{Z}_C \longrightarrow F_C \longrightarrow \mathcal{F}_C \longrightarrow 1, \tag{12}$$

where

$$F_C = \widehat{\pi_1(G)}, \tag{13}$$

and we identify $\mathcal{Z}_C = \widehat{\mathcal{Z}_g}$.

Let us note that in general 3d gauge theories it is necessary to include such topological symmetries as part of the structure group, thus extending the above discussed structure group $\mathcal{S}$ into an extended structure group $\widetilde{\mathcal{S}}$. Thus is due to the fact that monopole operators receive charges under gauge and flavor symmetries due to effective Chern-Simons levels, as discussed in our previous paper [12]. However, with $\mathcal{N} = 4$ supersymmetry and only standard supermultiplets, this extended structure group factorises as $\widetilde{\mathcal{S}} = \mathcal{S} \times \mathcal{F}_C$.

**Example.** A basic example to illustrate these points is supersymmetric QED with $\mathcal{G} = U(1)$ and $N$ hypermultiplets of charge $q$. We assume without loss of generality that $q > 0$.

The hypermultiplets contain complex scalar fields $X_j, Y_j$ of charge $q, -q$ transforming in the fundamental and anti-fundamental representations of the flavor symmetry algebra $\mathfrak{f}_H = \mathfrak{su}(N)$. The B-type genuine local operators are the gauge-invariant combinations $X_i Y_j$ transforming in the adjoint representation. The B-type flavor symmetry group is therefore $\mathcal{F}_H = PSU(N)$.

There are B-type Wilson lines $W_n$ labelled by an integer charge $n$. Consider the case where $n > 0$. If $n$ is a multiple of the minimal charge $q$, the Wilson line may end on local operators consisting of homogeneous polynomials in $X_j$ of degree $m = n/q$, which transform in the $m$-th symmetric power of the fundamental representation of $\mathfrak{su}(N)$. This includes representations of the central extension $F_H = SU(N)$ that are not representations of $\mathcal{F}_H = PSU(N)$ forming a short exact sequence

$$1 \longrightarrow \mathbb{Z}_N \longrightarrow SU(N) \longrightarrow PSU(N) \longrightarrow 1, \tag{14}$$

with $\mathcal{Z}_H = \mathbb{Z}_N$. The associated obstruction class may be denoted by $w_2^H \in H^2(X, \mathbb{Z}_N)$. This information is also catpured in the structure group

$$\mathcal{S} = \frac{U(1) \times SU(N)}{\mathbb{Z}_{qN}}, \tag{15}$$

where the denominator is generated by the central element $(e^{2\pi i/qN}, e^{2\pi i/N} 1_N)$.

The genuine A-type local operators correspond to half-BPS monopole operators labelled by a co-character $m : U(1) \to \mathcal{G}$, which is an integer magnetic charge $m \in \mathbb{Z}$. Correspondingly, the A-type symmetry group is the topological symmetry

$$\mathcal{F}_C = \widehat{\pi_1(\mathcal{G})} = U(1), \tag{16}$$

whose charge measures the topological type of a $\mathcal{G}$-bundle on a two-sphere surrounding a monopole operator.

To determine the required extension of the topological symmetry, consider A-type fractional gauge-flavour vortex line defects labelled by co-characters of the structure group $\phi : U(1) \to \mathcal{S}$. Such co-characters take the explicit form

$$\phi : e^{i\theta} \mapsto \left( e^{im\theta}, \mathrm{diag}\left( e^{im_1\theta}, \ldots, e^{im_N\theta} \right) \right), \tag{17}$$

where

$$m = n - \frac{\ell}{qN}, \qquad m_j = n_j - \frac{\ell}{N}, \tag{18}$$

where $n, n_j \in \mathbb{Z}$ and $\ell \in \{0, 1, \ldots, qN - 1\}$ with the condition $\sum_j n_j = \ell$ to ensure unit determinant. Note that $\theta \to \theta + 2\pi$ multiplies by $(e^{-2\pi\ell/qN}, e^{-2\pi\ell/N} 1_N)$, which is trivial under the quotient. In particular, $\ell \in \{0, 1, \ldots, qN - 1\}$, viewed additively as an element of $\mathcal{E} \cong \mathbb{Z}_{qN}$, is the obstruction to lifting to a co-character for the numerator $U(1) \times SU(N)$.

Let us summarise some examples that will play a role in what follows:

- Let us first consider pure gauge fractional vortex lines. This requires choosing $m_j = 0$ and therefore $n_j = \ell/N$ for all $j = 1,\ldots,N$. This is only possible if $\ell = Np$ and $n_j = p$. There are then two distinct situations:

  - $p = 0$: this is a dynamical gauge vortex and therefore corresponds to a trivial line defect. It ends on genuine A-type gauge monopole operators of integer magnetic charge $m = n$.

  - $p = 1,\ldots,N-1$: this is a fractional gauge vortex and defines a non-trivial A-type line defect. It ends on non-genuine gauge monopole operators of fractional magnetic charge $m = n - \frac{p}{q}$.

- More general mixed gauge-flavour fractional vortex lines with $\ell \neq 0$ may end on A-type monopole operators of fractional magnetic charge $m = n - \frac{\ell}{qN}$.

The final bullet point means we must introduce an $qN$-fold cover of the topological symmetry group, $F_C = \widehat{\pi_1(G)} \cong U(1)$, which is an extension of the topological symmetry by $\mathcal{Z}_C = \mathbb{Z}_{qN}$. The associated obstruction class for background fields is $w_2^C = c_1^C \bmod qN$, where $c_1^C$ denotes the first Chern class of a background $\mathcal{F}_C = U(1)$ bundle.

### 2.2.2 1-form symmetry

Following the same philosophy, we define A/B-type 1-form symmetries by applying the recipe studied in [12], but restricted to half-BPS A/B-type line defects.

**Definition.** The construction begins by considering equivalence classes of A-type or B-type line defects. We say that two line defects $L_1$, $L_2$ are equivalent $L_1 \sim L_2$ if there exists a non-trivial quarter-BPS junction of the appropriate type connecting them. In other words, equivalence classes capture the half-BPS line defects that cannot be screened by quarter-BPS junctions of A-type or B-type.

The equivalence classes of A-type and B-type lines inherit the structure of abelian groups $\widehat{\Gamma}_A$, $\widehat{\Gamma}_B$ from the OPE of parallel line defects. The A-type and B-type 1-form symmetries are defined as the Pontryagin dual groups

$$\begin{aligned}
\Gamma_A &:= \mathrm{Hom}(\widehat{\Gamma}_A, U(1)), \\
\Gamma_B &:= \mathrm{Hom}(\widehat{\Gamma}_B, U(1)),
\end{aligned} \tag{19}$$

such that these 1-form symmetries act on half-BPS lines via the natural pairings $\Gamma \times \widehat{\Gamma} \to U(1)$.

Correspondingly, we can introduce $\Gamma_A, \Gamma_B$-valued 2-cochain backgrounds $B_2^A, B_2^B$ for these 1-form symmetries. If the 1-form symmetries do not participate in 2-groups, the background field are closed and define $\Gamma_A, \Gamma_B$-valued 2-cocycles.

**Gauge theories.** In standard gauge theories built from vectormultiplets and hypermultiplets, the A-type and B-type 1-form symmetries may be determined from the properties of vortex lines and dynamical Wilson lines respectively.

The B-type symmetry arises from half-BPS Wilson lines in representations of the gauge group $\mathcal{G}$. In the absence of hypermultiplets, quarter-BPS junctions may only arise from vector-multiplet fields in the adjoint representation of the gauge group. In this case, Wilson lines in representations $R_1, R_2$ are equivalent if and only if the central characters of the representations coincide. Therefore

$$\widehat{\Gamma}_H = \mathrm{Hom}(Z(\mathcal{G}), U(1)), \tag{20}$$

is the abelian group of central characters and the 1-form symmetry coincides with the centre of the gauge group $\Gamma_H = Z(\mathcal{G})$.

More generally, incorporating hypermultiplet fields, the 1-form symmetry $\Gamma_H$ is the subgroup of the center of the gauge group that acts trivially on hypermultiplets. This can be formulated in terms of the structure group

$$\mathcal{S} = \frac{\mathcal{G} \times F_H}{\mathcal{E}}, \tag{21}$$

where the B-type 1-form symmetry may be identified with the intersection $\Gamma_H = Z(\mathcal{G}) \cap \mathcal{E}$. This naturally forms a short exact sequence

$$1 \longrightarrow \Gamma_H \longrightarrow \mathcal{E} \longrightarrow \mathcal{Z}_H \longrightarrow 1, \tag{22}$$

which will play a role in the construction of 2-group symmetries below.

An A-type 1-form symmetry may arise in gauge theories with discrete or continuous but disconnected gauge groups, which we will not discuss here. We will instead explain below how A-type 1-form symmetries arise generally when gauging discrete B-type 0-form symmetries.

**Example.** Let us again consider supersymmetric QED with $\mathcal{G} = U(1)$ and $N$ hypermultiplets of charge $q > 0$. It is a standard result that this has a B-type 1-form symmetry $\Gamma_B = \mathbb{Z}_q$ as B-type Wilson lines $W_n$ cannot be screened unless $n$ is a multiple of $q$.

### 2.2.3 2-group symmetry

The 0-form and 1-form symmetries defined above may combine to form A-type and B-type 2-group symmetries. See [65] for the first systematic study of 2-group global symmetries, and references therein for previous discussions about 2-groups in quantum field theory. In order to define the 2-group symmetry structure, we will need to consider a more refined equivalence relation for line defects that takes into account the fact that junctions may transform in representations of central extensions of symmetry groups. For further background on the perspective taken here see [12, 66–69].

**Definition.** We first define another equivalence relation such that $L_1 \sim' L_2$ if the two line operators admit quarter-BPS junctions of the appropriate type transforming in honest representations of $\mathcal{F}_A$, $\mathcal{F}_B$ that are not charged under $\mathcal{Z}_A$, $\mathcal{Z}_B$.

These equivalence classes form larger abelian groups $\widehat{\mathcal{E}_A}$, $\widehat{\mathcal{E}_B}$ sitting in short exact sequences

$$0 \longrightarrow \widehat{\mathcal{Z}_A} \longrightarrow \widehat{\mathcal{E}_A} \longrightarrow \widehat{\Gamma_A} \longrightarrow 0, \tag{23}$$

$$0 \longrightarrow \widehat{\mathcal{Z}_B} \longrightarrow \widehat{\mathcal{E}_B} \longrightarrow \widehat{\Gamma_B} \longrightarrow 0. \tag{24}$$

The first terms in the sequence can be understood as follows. The quarter-BPS local operators screening line operators in equivalence classes corresponding to elements $\widehat{z}_A \in \widehat{\mathcal{Z}_A} \subset \widehat{\mathcal{E}_A}$, $\widehat{z}_B \in \widehat{\mathcal{Z}_B} \subset \widehat{\mathcal{E}_B}$ transform in representations of $F_A$, $F_B$ with charges $\widehat{z}_A, \widehat{z}_B$ under $\mathcal{Z}_A, \mathcal{Z}_B$.

The Pontryagin dual exact sequences are

$$1 \longrightarrow \Gamma_A \longrightarrow \mathcal{E}_A \longrightarrow \mathcal{Z}_A \longrightarrow 1, \tag{25}$$

$$1 \longrightarrow \Gamma_B \longrightarrow \mathcal{E}_B \longrightarrow \mathcal{Z}_B \longrightarrow 1. \tag{26}$$

The 0-form symmetries $\mathcal{F}_A$, $\mathcal{F}_B$ and 1-form symmetries $\Gamma_A$, $\Gamma_B$ now combine into 2-groups whose Postnikov classes are given by

$$\Theta^A = \mathrm{Bock}(w_2^A), \tag{27}$$

$$\Theta^B = \mathrm{Bock}(w_2^B), \tag{28}$$

using the appropriate Bockstein homomorphisms Bock : $H^2(X, \mathcal{Z}_A) \rightarrow H^3(X, \Gamma_A)$ or Bock : $H^2(X, \mathcal{Z}_B) \rightarrow H^3(X, \Gamma_B)$ associated to the above short exact sequences. If the Postnikov classes are trivial the 2-group symmetry is a product of a 0-form and a 1-form symmetry.

We may then introduce backgrounds for the 2-group symmetry given by the $\mathcal{E}_A, \mathcal{E}_B$-valued combinations

$$B_w^A = i(B_2^A) + \widetilde{w}_2^A, \tag{29}$$

$$B_w^B = i(B_2^B) + \widetilde{w}_2^B, \tag{30}$$

where $i : \Gamma_A, \Gamma_B \rightarrow \mathcal{E}_A, \mathcal{E}_B$ denotes the relevant inclusion maps and $\widetilde{w}_2^A, \widetilde{w}_2^B$ are co-chain lifts of $w_2^A, w_2^B$ under the projections $p : \mathcal{E}_A, \mathcal{E}_B \rightarrow \mathcal{Z}_A, \mathcal{Z}_B$ in the above short exact sequences. These combinations are closed by construction and define $\mathcal{E}_A, \mathcal{E}_B$-valued co-cycles. If the Postnikov class is trivial, we may work independently with closed backgrounds $B_2^A, B_2^B$ and $w_2^A, w_2^B$ for the 1-form and 0-form symmetries respectively.

**Gauge theories.** Consider a standard supersymmetric 3d $\mathcal{N} = 4$ gauge theory built from ordinary vectormultiplets and hypermultiplets. The data determining the B-type 2-group is encoded in the structure group

$$\mathcal{S} = \frac{\mathcal{G} \times F_H}{\mathcal{E}}. \tag{31}$$

In particular, we have already identified $\mathcal{Z}_H = p_H(\mathcal{E})$ and the 1-form symmetry $\Gamma_B = Z(\mathcal{G}) \cap \mathcal{E}$. The remaining ingredient is simply the identification $\mathcal{E}_H = \mathcal{E}$, which forms the appropriate short exact sequence.

**Example.** Let us consider again supersymmetric QED with $\mathcal{G} = U(1)$ and $N$ hypermultiplets of charge $q > 1$. Recall that there are B-type symmetry groups $\mathcal{F}_H = PSU(N)$ and $\Gamma_H = \mathbb{Z}_q$ sitting in short exact sequences

$$1 \longrightarrow \mathbb{Z}_N \longrightarrow SU(N) \longrightarrow PSU(N) \longrightarrow 1, \tag{32}$$

and

$$1 \longrightarrow \mathbb{Z}_q \longrightarrow \mathbb{Z}_{qN} \longrightarrow \mathbb{Z}_N \longrightarrow 1, \tag{33}$$

respectively. There is therefore a potential Postnikov class $\Theta = \text{Bock}(w_2^H)$ where $w_2^H$ is the obstruction class for the first sequence and Bock : $H^2(PSU(N), \mathbb{Z}_N) \rightarrow H^3(PSU(N), \mathbb{Z}_q)$ is the Bockstein homomorphism for the second. The Bockstein homomorphism may or may not be trivial. In the former case, there is no 2-group symmetry. An example of vanishing Bockstein is provided if the first sequence splits, which requires $\gcd(q, N) = 1$. A non-supersymmetric version of this example was considered already in [12].

There is also an A-type 0-form symmetry $\mathcal{F}_A = U(1)$, which does not participate in a 2-group. However, we will show later that it has a mixed 't Hooft anomaly with the above B-type 2-group.

## 2.3 Solitonic defects

The A-type and B-type local, line and junctions operators may induce background fields for flavor symmetries and correspond to solitonic defects in the terminology of [12]. More specifically they induce vortex and monopole configurations for background fields associated to flavor symmetries. Such defects play a crucial role in determining 't Hooft anomalies from the spectrum of BPS charged objects.

The proposal is that A-type defects may source background fields for B-type flavor symmetries and vice-versa. We substantiate this claim for vortex and monopole backgrounds in the remainder of this subsection.

**Definitions.** For concreteness, let us first consider A-type line defects. The most general situation is that they induce a background field configuration for the B-type 2-group symmetry such that

$$\int_{D_2} B_w^B = \alpha_B \,, \tag{34}$$

where $D_2$ denotes a small disk intersecting the line defect transversely and $\alpha_B \in \mathcal{E}_B$. We refer to this as a background vortex configuration for the 2-group symmetry. Such line defects may end on A-type quarter-BPS local operators with the property that

$$\int_{S^2} B_w^B = \alpha_B \,, \tag{35}$$

where $S^2$ is now a small 2-sphere surrounding the local operator and intersecting the line defect transversely. We refer to this as a background monopole configuration for the 2-group symmetry. Entirely analogous statements hold with A-type and B-type symmetries and defects interchanged.

This reduces to simpler statements in special cases of individual 0-form and 1-form symmetry groups. For example, an A-type line defect may induce a vortex background for a B-type 0-form symmetry such that

$$\int_{D_2} w_2^B = \alpha_B \,, \tag{36}$$

where now $\alpha_B \in \mathcal{Z}_B$. Similarly, if there is a B-type 1-form symmetry that does not participate in a 2-group symmetry then an A-type line defect may induce a vortex background for the 1-form symmetry such that

$$\int_{D_2} B_2^B = \alpha_B \,, \tag{37}$$

where now $\alpha_B \in \Gamma_B$. Similar comments apply to local operators and monopole backgrounds. Again, entirely analogous statements fold with A-type and B-type symmetries and defects interchanged.

**Gauge theory.** In a standard supersymmetric gauge theory, the B-type symmetry backgrounds sourced by A-type line defects can be understood systematically. Let us consider fractional vortex lines labelled by co-characters of the structure group $\phi : U(1) \to \mathcal{S}$.

They source background field configurations for the B-type 2-group symmetry such that

$$\int_{D_2} B_2^H = \alpha_H \,, \tag{38}$$

where $\alpha_H \in \mathcal{E}$ is the obstruction for lifting $\phi$ to a co-character for $\mathcal{G} \times F$. This reduces to corresponding simpler statements for individual 0-form and 1-form symmetries. There are many special cases of interest considered in the example below.

In the opposite direction, B-type Wilson lines for a dynamical vectormultiplet source a background vortex configuration for the dual A-type topological symmetry. This is discussed for $\mathcal{G} = U(1)$ in the example below.

**Example.** Consider again supersymmetric QED with $N$ hypermultiplets of charge $q > 0$. We consider A-type gauge-flavor vortex lines labelled by a co-character of the structure group $\phi : U(1) \to \mathcal{S}$. Recall that such co-characters are labelled by fractional magnetic fluxes

$$m = n - \frac{\ell}{qN} \,, \qquad m_j = n_j - \frac{\ell}{N} \,, \tag{39}$$

where $n, n_j \in \mathbb{Z}$ and $\sum_j n_j = \ell$ with obstruction $\ell = 0, 1, \ldots, qN - 1$ to lifting to a co-character for $U(1) \times SU(N)$. Such a fractional vortex line sources a background for the B-type 2-group symmetry such that

$$\int_{D_2} B_w^H = \ell \,. \tag{40}$$

Let us now assume for simplicity that $\gcd(q, N) = 1$ so that the 2-group is split. Then we have the following decomposition:

- Consider fractional gauge vortex lines, which correspond to co-characters with $\ell = Np$ and $n_j = p$ where $p = 0, \ldots, q - 1$. They exclusively source backgrounds for the B-type 1-form symmetry $\Gamma_B = \mathbb{Z}_q$ such that

$$\int_{D_2} B_2^H = p \,. \tag{41}$$

- More general fractional gauge-flavour vortex lines induce combinations of 0-form and 1-form symmetry backgrounds $w_2^H, B_2^H$. Let us parametrize the obstruction by $\ell = Np + r$ with $p = 0, 1, \ldots, q - 1$ and $r = 0, \ldots, N - 1$. Then they source backgrounds

$$\begin{aligned} \int_{D_2} w_2^H &= r \,, \\ \int_{D_2} B_2^H &= p \,. \end{aligned} \tag{42}$$

In the opposite direction, the $B$-type Wilson lines $W_n$ source a background for the A-type topological symmetry $\mathcal{F}_C = U(1)$ such that

$$\int_{D_2} w_2^C = n \mod qN \,. \tag{43}$$

These statements will be utilised to derive mixed 't Hooft anomalies below.

## 2.4  't Hooft anomalies

We now consider the 't Hooft anomalies captured by BPS operators considered so far. These are primarily mixed 't Hooft anomalies between A-type and B-type symmetries.

The most general situation assuming potential A-type and B-type 2-group symmetries is as follows. Let us consider A-type line defects that source background field configurations for the B-type 2-group symmetry labelled by elements $\alpha_B \in \mathcal{E}_B$. Such line defects define equivalence classes in $\widehat{\mathcal{E}_A}$ and this provides a homomorphism

$$\widehat{\gamma} : \widehat{\mathcal{E}_B} \to \mathcal{E}_A \,. \tag{44}$$

In this situation there is a mixed 't Hooft anomaly represented by the four-dimensional SPT phase

$$\mathcal{A}_4 = \int B_w^A \cup \gamma(B_w^B) \,. \tag{45}$$

This construction may be performed exchanging A-type and B-type defects and symmetries and these constructions must be compatible.

There are various simpler special cases that are worth considering:

- Let us assume there are 1-form symmetries $\Gamma_A$, $\Gamma_B$ that do not participate in 2-groups. The A-type line defects may source backgrounds for a B-type 1-form symmetry labelled by elements $\alpha_B \in \Gamma_B$ and simultaneously charged under the A-type 1-form symmetry $\Gamma_A$. This determined a homomorphism

$$\gamma : \Gamma_B \to \widehat{\Gamma}_A, \tag{46}$$

and mixed 't Hooft anomaly

$$\mathcal{A}_4 = \int B_2^A \cup \gamma(B_2^B). \tag{47}$$

- Consider an A-type 0-form symmetry $\mathcal{F}_A$ and a B-type 1-form symmetry $\Gamma_B$ not participating in a 2-group. The A-type line defects may source backgrounds for a B-type 1-form symmetry labelled by elements $\alpha_B \in \Gamma_B$ and simultaneously end on A-type local operators charged under $\mathcal{Z}_A$. This determines a homomorphism

$$\gamma : \Gamma_B \to \widehat{\mathcal{Z}_A}, \tag{48}$$

and mixed 't Hooft anomaly

$$\mathcal{A}_4 = \int w_2^A \cup \gamma(B_2^B). \tag{49}$$

- Consider an A-type 0-form symmetry $\mathcal{F}_A$ and a B-type 0-form symmetry $\mathcal{F}_B$. The A-type line defects may source backgrounds $w_2^B$ for a B-type 0-form symmetry labelled by elements $\alpha_H \in \mathcal{Z}_B$ and simultaneously end on A-type local operators charged under $\mathcal{Z}_A$. This determines a homomorphism

$$\gamma : \mathcal{Z}_B \to \widehat{\mathcal{Z}_A}, \tag{50}$$

and mixed 't Hooft anomaly

$$\mathcal{A}_4 = \int w_2^A \cup \gamma(w_2^B). \tag{51}$$

**Gauge theory.** For a standard supersymmetric gauge theory with connected gauge group, there may be a mixed 't Hooft anomaly between the A-type topological symmetry and the B-type 2-group symmetry. This may be determined, for example, by examining gauge-flavor vortex line defects that induce backgrounds for the B-type flavor symmetry and the fractional A-type topological charges of the monopoles on which they end. An example is presented below.

**Example.** Let us consider again supersymmetric QED with $N$ hypermultiplets of charge $q > 0$. The symmetries are summarised as follows:

- An A-type topological symmetry $\mathcal{F}_A = U(1)$ whose background field has an $\mathbb{Z}_{qN}$-valued obstruction class $w_2^C = c_1 \bmod qN$.

- A B-type 2-group symmetry with $\mathcal{F}_H = PSU(N)$ and $\Gamma_H = \mathbb{Z}_q$ with $\mathbb{Z}_{qN}$-valued background field $B_w^H = NB_2 + \widetilde{w}_2^H$.

The mixed 't Hooft anomaly between these symmetries is derived from the fractional magnetic charges of the A-type local operators on which gauge-flavour vortex lines end. A slight generalisation of the examples presented in [12] shows that this is represented by the 4d SPT phase

$$\mathcal{A}_4 = \exp\left(\frac{2\pi i}{qN} \int w_2^C \cup B_w^H\right). \tag{52}$$

When $\gcd(q, N) = 1$ and the 2-group is split, this simplifies to a sum of mixed anomalies for the individual B-type 0-form and 1-form symmetries

$$\mathcal{A}_4 = \exp\left(\frac{2\pi i}{q} \int (c_1^C \bmod q) \cup B_2^H + \frac{2\pi i}{N} \int (c_1^C \bmod N) \cup w_2^B\right). \tag{53}$$

## 2.5 Discrete gauging

In three dimensions, gauging a discrete abelian 0/1-form symmetry $\Gamma$ group results in a Pontryagin dual 1/0-form symmetry group $\widehat{\Gamma} := \mathrm{Hom}(A, U(1))$. In the context of symmetries in theories with $\mathcal{N} = 4$ supersymmetry these operations interchange A-type and B-type symmetries. In summary:

- Gauging an A-type discrete abelian 0/1-form symmetry $\Gamma$ results in a B-type Pontryagin dual 1/0-form symmetry $\widehat{\Gamma} := \mathrm{Hom}(\Gamma, U(1))$.

- Gauging an B-type discrete abelian 0/1-form symmetry $\Gamma$ results in a A-type Pontryagin dual 1/0-form symmetry $\widehat{\Gamma} := \mathrm{Hom}(\Gamma, U(1))$.

This is compatible with our discussion of mixed 't Hooft anomalies between A-type and B-type symmetries. A slight generalisation of the above is that gauging a normal subgroup $\Gamma \subset \Gamma'$ of a 0-form symmetry results in a 1-form symmetry $\widehat{\Gamma}$ with a mixed anomaly with the remaining quotient group $\Gamma'/\Gamma$ controlled by the extension class [70].[5] In theories with $\mathcal{N} = 4$ supersymmetry and with the above identifications, this is always a mixed anomaly between A-type and B-type symmetries considered above.

This provides a clean and general method to construct new examples of mirror symmetry that involved 1-form symmetries and their anomalies can be explicitly matched. Examples are presented below and in the remainder of the paper.

**Example.** An example of this phenomenon arises in $U(1)$ supersymmetric gauge theories, which have an A-type topological symmetry $\mathcal{F}_C = U(1)$ under which genuine monopole operators are charged. Gauging a subgroup $\mathbb{Z}_q \subset U(1)$ of the topological symmetry is equivalent to multiplying the charges of all hypermultiplet fields by $q$. This results in a B-type 1-form symmetry $\Gamma_H = \mathbb{Z}_q$ due since a subgroup of the gauge group now acts trivially on all hypermultiplet fields. This B-type symmetry has a mixed anomaly with the remaining topological symmetry after gauging.

As an example consider supersymmetric QED with $N$ hypermultiplets of charge 1. This has A-type topological symmetry $\mathcal{F}_C = U(1)$ and B-type symmetry $\mathcal{F}_H = PSU(N)$ with mixed 't Hooft anomaly

$$\mathcal{A}_4 = \exp\left(\frac{2\pi i}{N} \int (c_1^C \bmod N) \cup w_2^H\right). \tag{54}$$

We now gauge $\mathbb{Z}_q \subset \mathcal{F}_C$ assuming $\gcd(q, N) = 1$. This results in a dual B-type 1-form symmetry $\Gamma_H = \mathbb{Z}_q$ and an additional mixed anomaly with the remaining topological symmetry such that

---

[5]This conclusion holds even when $\Gamma'$ is continuous.

the total anomaly is

$$\mathcal{A}_4 = \exp\left( \frac{2\pi i}{q} \int (c_1^C \bmod q) \cup B_2^H + \frac{2\pi i}{N} \int (c_1^C \bmod N) \cup w_2^H \right). \tag{55}$$

This is indeed the anomaly of supersymmetric QED with $N$ hypermultiplets of charge $q$. A slightly more intricate argument is required when $\gcd(q,N) > 1$.

The mirror of supersymmetric QED with $N$ hypermultiplets of charge $q$ can therefore be obtained from the mirror of supersymmetric QED with $N$ hypermultiplets of charge 1 by gauging a $\mathbb{Z}_q$ symmetry. This results in a circular quiver gauge theory with $(N-1)$ $U(1)$ nodes and 1 $\mathbb{Z}_q$ node, which has an A-type 1-form symmetry $\Gamma_A = \mathbb{Z}_q$. In particular, when $N = 1$, the mirror theory is a $\mathbb{Z}_q$-quotient of a free hypermultiplet.

# 3 Warmup: T[SU($n$)] and its gaugings

In this section, we study the simplest example, which is provided by 3d $\mathcal{N} = 4$ SCFTs known as T[SU($n$)] theories. We study the global forms of flavor symmetry groups of T[SU($n$)] along with their 't Hooft anomalies. We also study some other 3d $\mathcal{N} = 4$ SCFTs closely related to T[SU($n$)], in that they can be obtained by gauging (along with the addition of extra flavors for balancing purposes) the Higgs branch flavor symmetries of the UV unitary quiver theory whose IR fixed point is T[SU($n$)].

## 3.1 T[SU(2)] and its gaugings

Let us begin with T[SU(2)] and theories related to it by gauging. Some of the results on the symmetries and anomalies of the T[SU(2)] theory are known already in the literature [71,72]. We derive them here using the perspective of our earlier paper [12].

### 3.1.1 T[SU(2)]

The T[SU(2)] theory arises as an IR fixed point of the following 3d $\mathcal{N} = 4$ Lagrangian theory

$$U(1) \text{——} [\mathfrak{su}(2)_H], \tag{56}$$

having a $U(1)$ gauge group and 2 hypermultiplets of charge 1 that are rotated by an $\mathfrak{su}(2)_H$ flavor symmetry algebra,[6] where the subscript $H$ indicates that the $\mathfrak{su}(2)_H$ symmetry acts non-trivially (i.e. is spontaneously broken) on the Higgs branch of vacua of the theory.

**B-type 0-form symmetry.** The genuine local operators charged under $\mathfrak{su}(2)_H$ arise from gauge invariant combinations of hypermultiplets. Such gauge invariant combinations all form representations of the Lie group $SO(3)_H$ with Lie algebra $\mathfrak{su}(2)_H$.

Thus, the Higgs branch 0-form symmetry group is

$$\mathcal{F}_H = SO(3)_H = SU(2)_H/\mathbb{Z}_2, \tag{57}$$

where $SU(2)_H$ is the simply connected group with Lie algebra $\mathfrak{su}(2)_H$.

Another way of deducing the above 0-form symmetry group is as follows. Let us put the theory (56) on a non-trivial compact 3-manifold $M_3$. The charges of the vector and hypermultiplets allow us to turn on bundles for the structure group

$$\mathcal{S} = \frac{U(1) \times SU(2)_H}{\mathbb{Z}_2}, \tag{58}$$

---

[6]Note that the flavor symmetry is not $\mathfrak{u}(2)_H = \mathfrak{su}(2)_H \oplus \mathfrak{u}(1)_H$. One way to see it is to note that the $\mathfrak{u}(1)_H$ part acts in the same way as the $\mathfrak{u}(1)$ gauge algebra and hence is absorbed into that.

where $U(1)$ is the gauge group. The $\mathbb{Z}_2$ in the denominator is the diagonal combination of the $\mathbb{Z}_2$ element in $U(1)$ and the non-trivial element in the $\mathbb{Z}_2$ center of $SU(2)_H$. This diagonal combination leaves all vector and hyper multiplets invariant. Thus the flavor symmetry group associated to $\mathfrak{su}(2)_H$ is

$$SO(3)_H = SU(2)_H/\mathbb{Z}_2 \,, \tag{59}$$

because, according to (58), we can couple the theory (56) to non-trivial background bundles for $SO(3)_H$, provided we turn on non-trivial bundles for $U(1)/\mathbb{Z}_2$. In more detail, let $w_2^H$ be the obstruction class for lifting the $SO(3)_H$ bundle to an $SU(2)_H$ bundle, i.e. the second Stiefel-Whitney class of the $SO(3)_H$ bundle. The obstruction class for lifting the $U(1)/\mathbb{Z}_2$ bundle to a $U(1)$ bundle can then be written as $c_1 \pmod 2$, where $c_1$ is the first Chern class for the $U(1)/\mathbb{Z}_2$ bundle. The group (58) then requires that

$$w_2^H = c_1 \pmod 2 \,. \tag{60}$$

That is, $SO(3)_H$ bundle can be lifted to $SU(2)_H$ bundle if and only if $U(1)/\mathbb{Z}_2$ bundle can be lifted to $U(1)$ bundle. Once an $SO(3)_H$ bundle is specified, the gauge theory sums over all $U(1)/\mathbb{Z}_2$ bundles satisfying the constraint (60).

**A-type 0-form symmetry.** In addition of the $SO(3)_H$ 0-form symmetry, the Lagrangian theory (56) admits a magnetic

$$\mathcal{F}_C^{\text{UV}} = U(1)_C \,, \tag{61}$$

0-form symmetry whose associated topological operators are

$$\exp\left( i\alpha \oint F \right) \,, \tag{62}$$

parametrized by $\alpha \in [0, 2\pi)$, where $F$ is the field strength of the $U(1)$ gauge group. The subscript $C$ denotes that the $U(1)_C$ symmetry acts non-trivially on the Coulomb branch of vacua of the theory. Indeed, the monopole operators, whose vacuum expectation values parametrize the Coulomb branch, are charged under this symmetry, with the charges valued in $\mathbb{Z}$.

It is well-known, from the analysis of [5], that at the level of Lie algebras, the $\mathfrak{u}(1)_C$ 0-form symmetry enhances in the IR to an $\mathfrak{su}(2)_C$ 0-form symmetry. In particular, the Cartan of the simply connected Lie group $SU(2)_C$ with Lie algebra $\mathfrak{su}(2)_C$ is a double cover of $U(1)_C$, or in other words we have an inclusion map

$$U(1)_C \hookrightarrow SO(3)_C = SU(2)_C/\mathbb{Z}_2 \,, \tag{63}$$

which embeds $U(1)_C$ as the maximal torus of $SO(3)_C$.

Since the gauge monopole operators have integer charges under $U(1)_C$ in the UV, they descend to genuine local operators of the IR SCFT transforming in representations of $SO(3)_C$. Thus the Coulomb 0-form symmetry group of the IR SCFT is

$$\mathcal{F}_C^{\text{IR}} = SO(3)_C \,. \tag{64}$$

In other words, at the level of Lie groups the $U(1)_C$ 0-form symmetry group enhances in the IR to $SO(3)_C$ 0-form symmetry group.

**Mixed 0-form symmetry anomaly.** As is well-known, there is a mixed 't Hooft anomaly between the $SO(3)_H$ and $SO(3)_C$ 0-form symmetries of T[SU(2)], see [71,72]. Here we describe how this 't Hooft anomaly can be derived as a consequence of a mixed 't Hooft anomaly between the $SO(3)_H$ and $U(1)_C$ 0-form symmetries in the UV gauge theory. For this purpose, we

will use the analysis of [12] which described a general way of deducing 't Hooft anomalies for any 3d gauge theory by computing center charges of flavor-gauge monopole operators. The relevant monopole operator is associated to a co-character

$$U(1) \to \mathcal{S}, \tag{65}$$

where $\mathcal{S}$ is the structure group of the gauge theory appearing in (58), with winding number half around the $U(1)$ factor in the numerator of $\mathcal{S}$ and winding number half around the $U(1)$ maximal torus of $SU(2)_H$. This is an allowed co-character because of the $\mathbb{Z}_2$ quotient appearing in the denominator of $\mathcal{S}$.

The mixed flavor-gauge monopole operator $O$ associated to such a co-character is necessarily a non-genuine local operator of the gauge theory lying at the end of a vortex line defect. From the methods of [12], we can compute that $O$ carries a charge $1/2$ under $U(1)_C$. A quick way to see this is to note that a purely gauge monopole operator associated to a co-character with winding number 1 around $U(1)$ gauge group has charge 1 under $U(1)_C$; a purely flavor monopole operator associated to a co-character with winding number 1 around $U(1)$ maximal torus of $SU(2)_H$ is uncharged under $U(1)_C$; and twice of the co-character associated to $O$ is a product of such purely gauge and purely flavor co-characters.

As explained in [12], the half-integral charge under $U(1)_C$ of the monopole operator $O$ is equivalent to a mixed 't Hooft anomaly between the Coulomb and Higgs 0-form symmetry groups of the UV gauge theory

$$\mathcal{A}_4^{\mathrm{UV}} = \exp\left( \pi i \int w_2^H \cup \left[ c_1\big(U(1)_C\big) \,(\mathrm{mod}\,2) \right] \right), \tag{66}$$

where $w_2^H$ denotes the Stiefel-Whitney class for the background $SO(3)_H$ bundle and $c_1\big(U(1)_C\big)$ denotes the first Chern class of the background $U(1)_C$ bundle.

After flowing to the IR, the monopole operator $O$ descends to a non-genuine local operator of the T[SU(2)] theory that is now a purely flavor monopole operator (because there is no gauge group in the IR SCFT) associated to a co-character having winding number 1 around the $SO(3)_H$ 0-form symmetry group of T[SU(2)]. Any flavor monopole operator is a non-genuine local operator living at the end of a flavor vortex line defect associated to the same co-character. The flavor monopole operator $O$ must transform in a representation of $\mathfrak{su}(2)_C$ which is not an allowed representation of $SO(3)_C$. This is a straightforward consequence of the embedding (63). Again using the analysis of [12], this fact is equivalent to a mixed 't Hooft anomaly between the Coulomb and Higgs 0-form symmetry groups of the IR SCFT T[SU(2)]

$$\mathcal{A}_4^{\mathrm{IR}} = \exp\left( \pi i \int w_2^H \cup w_2^C \right), \tag{67}$$

where $w_2^C$ is the second Stiefel-Whitney class of the background $SO(3)_C$ bundle.

One can think of the IR anomaly (67) as being obtained from the UV anomaly (66) by 't Hooft anomaly matching. The above analysis in terms of charges of flavor-monopole operators thus provides a precise and unambiguous way of performing such a 't Hooft anomaly matching.

### 3.1.2 $SU(2)_H$ gauging

Consider now the 3d $\mathcal{N} = 4$ theory

$$\mathrm{T[SU(2)]} \relbar\joinrel\relbar SU(2)_H, \tag{68}$$

obtained by gauging the $\mathfrak{su}(2)_H$ flavor symmetry of T[SU(2)] by an $SU(2)_H$ gauge group. We can reach a very closely related cousin of this theory by flowing from the 3d $\mathcal{N} = 4$ quiver

$$U(1) \relbar\joinrel\relbar\joinrel\relbar SU(2)_H, \tag{69}$$

if the gauge coupling for $SU(2)_H$ is extremely small compared to the $U(1)$ gauge coupling. However, in this way we never land precisely on the theory (68).

Note that (68) is a "bad" theory in the sense of [5]. After a discussion of the symmetries and anomalies of this theory, we will add flavors for the $SU(2)_H$ gauge group converting the above theory into a "good" theory and study its symmetries.

**1-form symmetry.** This theory carries a

$$\Gamma^{(1)} = \mathbb{Z}_2 \,, \tag{70}$$

1-form symmetry, as can be seen by the following argument. After gauging we obtain Wilson line defects valued in representations of $SU(2)_H$. A genuine local operator of T[SU(2)] transforming in representation $R$ of $SU(2)_H$ becomes a non-genuine local operator of the gauged theory (68) that lives at the end of Wilson line defect in representation $R$. In other words, the Wilson line in representation $R$ is screened in the gauged theory (68). From our previous discussion, we know that the genuine local operators transform only in those representations of $SU(2)_H$ that are also representations of $SO(3)_H$. Thus, only Wilson lines in representations of $SO(3)_H$ are screened, but the Wilson lines in representations of $SU(2)_H$ that are not representations of $SO(3)_H$ are not screened. The non-screened Wilson lines are non-trivially charged under the $\mathbb{Z}_2^H$ center of $SU(2)_H$, which descends to a $\mathbb{Z}_2$ 1-form symmetry of the gauged theory (68).

**0-form symmetry.** We claim that the $SO(3)_C$ 0-form symmetry of T[SU(2)] is not impacted by the gauging procedure and the gauged theory (68) also has

$$\mathcal{F}_C = SO(3)_C \,, \tag{71}$$

0-form symmetry. To see this, we need to show that all genuine local operators of the gauged theory form representations of $SO(3)_C$. It is sufficient to show this fact for the following two types of genuine local operators:

1. The genuine local operators of T[SU(2)] that are uncharged under $SU(2)_H$ descend to genuine local operators of the gauged theory (68). Since such operators form $SO(3)_C$ representations before gauging, they also form $SO(3)_C$ representations after gauging.

2. The flavor monopole operators for $\mathfrak{su}(2)_H$ are non-genuine in T[SU(2)] as they are attached to vortex line defects. Some of these vortex line defects become invisible after gauging and the attached flavor monopole operators thus become genuine local operators of the gauged theory (68). The co-characters associated to such monopoles are those which have even winding numbers around the maximal torus of $SO(3)_H$. Such monopole operators form $SO(3)_C$ representations before gauging, so only lead to $SO(3)_C$ representations after gauging.

**Is there a 2-group symmetry?** The question we would now like to address is whether the above $\mathbb{Z}_2$ 1-form and $SO(3)_C$ 0-form symmetries combine to form a 2-group symmetry with a non-trivial Postnikov class. We claim that the answer is *negative*, due to the following reason.

The existence of such a 2-group symmetry requires the presence of a local operator $O$ in the gauged theory (68) which sits at the end of a Wilson line operator transforming in an allowed representation of $SO(3)_H$ and transforms in a representation of $SU(2)_C$ that is *not* an allowed representation of $SO(3)_C$. This means that in the ungauged theory T[SU(2)], $O$ must be a local operator (transforming in the same representations of $SO(3)_H$ and $SU(2)_C$) of one of the following two types:

1. $O$ is a genuine local operator in T[SU(2)]. But then $O$ must transform in an $SO(3)_C$ representation because the Coulomb 0-form symmetry group of T[SU(2)] is $SO(3)_C$.

2. $O$ is a flavor monopole operator in T[SU(2)] associated to a co-character with even winding number around the maximal torus of $SO(3)_H$. But then $O$ must transform in an $SO(3)_C$ representation as already discussed above.

Thus, we conclude that there is no non-trivial 2-group symmetry in the gauged theory (68).

**Mixed 0-/1-form symmetry 't Hooft anomaly.** A flavor monopole operator in T[SU(2)] with odd winding number around maximal torus of $SO(3)_H$ becomes a gauge monopole operator in the gauged theory (68). Such a gauge monopole operator is not a standard gauge monopole operator usually discussed in the literature. The latter monopole operators are genuine local operators while the former monopole operator must be non-genuine attached to a non-trivial solitonic line defect which induces a non-trivial flux for the background field $B_2$ for the $\mathbb{Z}_2$ 1-form symmetry [12]. For this reason, such monopole operators were referred to as *fractional* gauge monopole operators in [12] to distinguish them from the standard (non-fractional) gauge monopole operators. Some fractional monopole operators lie in the twisted sector of the $\mathbb{Z}_2$ 1-form symmetry, that is, they lie at the end of a topological line operator generating the $\mathbb{Z}_2$ 1-form symmetry.

Since such a fractional gauge monopole operator forms a representation of $SU(2)_C$ that is not an allowed representation of $SO(3)_C$, using the analysis of [12] we learn that there is a mixed 't Hooft anomaly

$$\mathcal{A}_4 = \exp\left( \pi i \int B_2 \cup w_2^C \right) , \tag{72}$$

between the $\mathbb{Z}_2$ 1-form symmetry and $SO(3)_C$ 0-form symmetry of the gauged theory (68).

A more straightforward way of deriving the above anomaly is to first note that the background field $B_2$ for the $\mathbb{Z}_2$ 1-form symmetry can be identified with the obstruction class $w_2^H$

$$B_2 = w_2^H , \tag{73}$$

and then the anomaly (72) follows simply from the anomaly (67) of T[SU(2)].

**Adding flavors.** We are interested in understanding the global form of the Coulomb symmetry group of the IR SCFT obtained (for large enough $N$) from the UV 3d $\mathcal{N} = 4$ Lagrangian theory

$$U(1) \rule{2em}{0.4pt} SU(2)_H \rule{1.5em}{0.4pt} N\,\mathsf{F} , \tag{74}$$

where we have $N$ hypermultiplets transforming in fundamental representation of $SU(2)_H$ along with the bifundamental hypermultiplet between $U(1)$ and $SU(2)_H$. Note that the corresponding IR SCFT can also be obtained by starting from the good version

$$\text{T[SU(2)]} \rule{2em}{0.4pt} SU(2)_H \rule{1.5em}{0.4pt} N\,\mathsf{F} , \tag{75}$$

of the theory (68), obtained from (68) by adding $N$ fundamental hypers for $SU(2)_H$. The relationship between theories (74) and (75) is that the theory (74) flows very close to the theory (75) at intermediate energy scales if the gauge coupling for $SU(2)_H$ is extremely small compared to the gauge coupling for $U(1)$.

We can thus use the symmetry properties of the theory (68) to deduce symmetry properties for the IR SCFT associated to (74) as follows. The additional flavors in the theory (75) screen the fundamental Wilson line of $SU(2)_H$ and thus the $\mathbb{Z}_2$ 1-form symmetry of the theory (68) is lost in the theory (75). On the other hand, the only new genuine local operators we obtain

are gauge invariant combinations of these hypers. These carry trivial charge under $SO(3)_C$, and hence the Coulomb 0-form symmetry group of the theory (75) is $SO(3)_C$. For generic $N$, there is no enhancement of $SO(3)_C$ and the IR SCFT originating from (74) (which is the same as the IR SCFT originating from (75)) has Coulomb 0-form symmetry group

$$\mathcal{F}_C^{\text{IR}} = SO(3)_C. \tag{76}$$

### 3.1.3 $SO(3)_H$ gauging

Consider now the 3d $\mathcal{N} = 4$ theory

$$\text{T[SU(2)]} \;\rule[0.5ex]{2em}{0.4pt}\; SO(3)_H, \tag{77}$$

obtained by gauging the $\mathfrak{su}(2)_H$ flavor symmetry of T[SU(2)] by an $SO(3)_H$ gauge group. One can reach a very closely related theory by flowing from a 3d $\mathcal{N} = 4$ Lagrangian theory

$$\mathfrak{u}(1) \;\rule[0.5ex]{3em}{0.4pt}\; \mathfrak{su}(2)_H \tag{78}$$

(where we have only displayed gauge algebras), with the gauge group

$$\mathcal{G} = \frac{U(1) \times SU(2)_H}{\mathbb{Z}_2}, \tag{79}$$

where the $\mathbb{Z}_2$ being quotiented out is the diagonal combination of the $\mathbb{Z}_2$ subgroup of $U(1)$ and the $\mathbb{Z}_2^H$ center of $SU(2)_H$. Note that (77) is again a bad theory just like (68). Later, we also consider its good versions obtained by adding adjoint flavors for the $SO(3)_H$ gauge group.

**0-form symmetry group.** In fact, this theory (77) can be obtained by gauging the $\mathbb{Z}_2$ 1-form symmetry of the theory (68). As a consequence, we expect the above theory (77) to contain a dual $\mathbb{Z}_2$ 0-form symmetry, alongside the residual $SO(3)_C$ 0-form symmetry descending from (68). However, the combined group structure of the 0-form symmetry is not $\mathbb{Z}_2 \times SO(3)_C$. Instead, the mixed anomaly (72) between $\mathbb{Z}_2$ 1-form and $SO(3)_C$ 0-form symmetries of the theory (68) dualizes to a non-trivial extension between the $\mathbb{Z}_2$ and $SO(3)_C$ 0-form symmetries of the theory (77). Thus, in total the 0-form symmetry of the theory (77) is

$$\mathcal{F}_C = SU(2)_C, \tag{80}$$

which is a non-trivial extension of the form

$$1 \rightarrow \mathbb{Z}_2 \rightarrow SU(2)_C \rightarrow SO(3)_C \rightarrow 1. \tag{81}$$

To see it more concretely, following [73] let us explicitly perform the gauging over $B_2$ with the term $B_2 \cup B_1$ added to the 3d action, where $B_1$ is background field for dual $\mathbb{Z}_2$ 0-form symmetry. This modifies the anomaly as

$$\mathcal{A}_4 \rightarrow B_2 \cup w_2^C + \delta B_2 \cup B_1 + B_2 \cup \delta B_1. \tag{82}$$

The second term $\delta B_2 \cup B_1 = 0$ as $B_2$ is closed. The rest of the anomaly $B_2 \cup (\delta B_1 + w_2^C)$ is a gauge anomaly (because $B_2$ is a gauge field now), so it must vanish. This gives us the constraint

$$\delta B_1 = w_2^C, \tag{83}$$

which implies that $\mathbb{Z}_2$ and $SO(3)_C$ 0-form symmetries indeed combine to form $SU(2)_C$ 0-form symmetry.

The same conclusion can also be reached by studying the monopole operators discussed above. Recall that (68) contains a fractional gauge monopole operator $O$ living at the end of topological line operator generating the $\mathbb{Z}_2$ 1-form symmetry, such that $O$ transforms in a representation of $SU(2)_C$ which is not a representation of $SO(3)_C$. As we gauge the 1-form symmetry, the topological line generating the $\mathbb{Z}_2$ 1-form symmetry disappears, and $O$ becomes a genuine local operator of the theory (77). Thus, the 0-form symmetry group associated to $\mathfrak{su}(2)_C$ 0-form symmetry algebra in the theory (77) is $SU(2)_C$.

**Adding flavors.** We are interested in understanding the global form of the Coulomb symmetry group of the IR SCFT obtained (for large enough $N$) from the UV 3d $\mathcal{N} = 4$ Lagrangian theory

$$\mathfrak{u}(1) \,\text{————}\, \mathfrak{su}(2)_H \,\text{——}\, N\,\text{A}\,, \tag{84}$$

where we have $N$ hypers transforming in adjoint representation of $\mathfrak{su}(2)_H$ along with the bifundamental hyper between $\mathfrak{u}(1)$ and $\mathfrak{su}(2)_H$, and the gauge group is chosen to be (79). Note that the corresponding IR SCFT can also be obtained by starting from the good version

$$\text{T}[\text{SU}(2)] \,\text{——}\, SO(3)_H \,\text{——}\, N\,\text{A}\,, \tag{85}$$

of the theory (77), obtained from (77) by adding $N$ adjoint hypers for $SO(3)_H$. The relationship between theories (84) and (85) is that the theory (84) flows very close to the theory (75) at intermediate energy scales if the gauge coupling for $\mathfrak{su}(2)_H$ is extremely small compared to the gauge coupling for $\mathfrak{u}(1)$.

We can thus use the symmetry properties of the theory (77) to deduce symmetry properties for the IR SCFT associated to (84) as follows. Recall that (77) has a gauge monopole operator transforming in $SU(2)_C$ representation that is not an $SO(3)_C$ representation. This operator is not impacted by the addition of adjoint hypers. Thus, we deduce that the Coulomb 0-form symmetry group of the theory (85) is $SU(2)_C$. For generic $N$, there is no enhancement of $SU(2)_C$ and the IR SCFT originating from (84) (which is the same as the IR SCFT originating from (85)) has Coulomb 0-form symmetry group

$$\mathcal{F}_C^{\text{IR}} = SU(2)_C\,. \tag{86}$$

### 3.1.4 $U(2)$ gauging

We are interested in understanding the global form of Coulomb and Higgs symmetry groups of the IR SCFT obtained (for large enough $N$) from the UV 3d $\mathcal{N} = 4$ Lagrangian theory

$$U(1) \,\text{————}\, U(2) \,\text{——}\, [\mathfrak{su}(N)_H]\,, \tag{87}$$

where we have $N$ hypers transforming in fundamental representation of $U(2)$ along with the bifundamental hyper between $U(1)$ and $U(2)$.

**0-form symmetry algebras.** As shown in (87), there is a Higgs branch flavor symmetry

$$\mathfrak{f}_H = \mathfrak{su}(N)_H\,, \tag{88}$$

rotating the $N$ fundamental hypers.

We will focus on the case $N > 3$ for which the IR Coulomb symmetry algebra is

$$\mathfrak{f}_C^{\text{IR}} = \mathfrak{u}(2)_C = \mathfrak{su}(2)_C \oplus \mathfrak{u}(1)_C\,. \tag{89}$$

The $N = 3$ case has a further enhancement of IR Coulomb symmetry algebra to $\mathfrak{su}(3)_C$ that will be discussed in the following subsection on $\text{T}[\text{SU}(n)]$ theories.

The $\mathfrak{su}(2)_C$ subalgebra of $\mathfrak{f}_C^{\text{IR}}$ is usually associated to the balanced node provided by the $U(1)$ gauge group, and the $\mathfrak{u}(1)_C$ subalgebra of $\mathfrak{f}_C^{\text{IR}}$ is usually associated to the unbalanced node provided by the $U(2)$ gauge group. However, we will need a more precise identification of these subalgebras, which we discuss in what follows.

**0-form symmetry groups.** The Higgs 0-form symmetry group of the IR SCFT is the same as that of the UV Lagrangian theory

$$\mathcal{F}_H = PSU(N)_H = SU(N)_H/\mathbb{Z}_N \,, \tag{90}$$

and can be easily deduced by noticing that the gauge invariant combinations of hypermultiplets all form representations of $PSU(N)_H$.

On the other hand, the Coulomb 0-form symmetry group of the IR SCFT is more subtle to deduce. Let us begin with the Coulomb 0-form symmetry group of the UV theory. We have a $U(1)_1$ 0-form symmetry associated to the $U(1)$ gauge node and a $U(1)_2$ 0-form symmetry associated to the $U(2)$ gauge node. First, we need to understand the precise identification of the $U(1)$ subgroup of UV 0-form symmetry group

$$\mathcal{F}_C^{\text{UV}} = U(1)_1 \times U(1)_2 \,, \tag{91}$$

which becomes the maximal torus of the Lie group $SU(2)_C$ associated to the $\mathfrak{su}(2)_C$ subalgebra of $\mathfrak{f}_C^{\text{IR}}$. This is done by studying $\mathcal{F}_C^{\text{UV}}$ charges of BPS monopole operators of low R-charge. We look for a combination[7] $q = n_1 q_1 + n_2 q_2$, where $q_i$ is the $U(1)_i$ charge of monopole and $n_i \in \mathbb{Z}$, such that the set of monopole operators at a fixed value of R-charge have $q$-charges coinciding with the Dynkin coefficients of the weights of a representation of $\mathfrak{su}(2)_C$. This fixes

$$q = 2q_1 - q_2 \,. \tag{92}$$

Now the $\mathfrak{u}(1)_C$ factor is determined such that the BPS monopole operators furnishing the weights of adjoint of $\mathfrak{su}(2)_C$ are uncharged under $\mathfrak{u}(1)_C$. This determines that the $\mathfrak{u}(1)_C$ charge is proportional to $q_2$, or more precisely there is a Lie group $U(1)_C$ with Lie algebra $\mathfrak{u}(1)_C$, such that the charge $q_C$ under $U(1)_C$ is

$$q_C = q_2 \,. \tag{93}$$

In order to determine the global form of the IR Coulomb flavor group $\mathcal{F}_C^{\text{IR}}$ we need to determine the charges

$$\big(q \,(\text{mod } 2), q_C\big), \tag{94}$$

of non-fractional gauge monopole operators, where the charge $q \,(\text{mod } 2)$ is the charge of the monopole operator under the $\mathbb{Z}_2$ center of the IR $SU(2)_C$. The fundamental monopole operator from $U(1)$ gauge node has $(q_1, q_2) = (1, 0)$ implying $\big(q \,(\text{mod } 2), q_C\big) = \big(0 \,(\text{mod } 2), 0\big)$, while the fundamental monopole operators from $U(2)$ gauge node have $(q_1, q_2) = (0, 1)$ implying $\big(q \,(\text{mod } 2), q_C\big) = \big(1 \,(\text{mod } 2), 1\big)$. The only non-trivial charge is the latter one, which implies that the Coulomb 0-form symmetry group of the IR SCFT is

$$\mathcal{F}_C^{\text{IR}} = U(2)_C = \frac{SU(2)_C \times U(1)_C}{\mathbb{Z}_2} \,. \tag{95}$$

## 3.2 T[SU($n$)] and its gaugings

In this subsection we generalize to T[SU($n$)] theories the results obtained in the previous subsection regarding T[SU(2)] theories.

---

[7]We thank Antoine Bourget for a discussion regarding this point.

### 3.2.1 T[SU($n$)]

**B-type 0-form symmetry.** The T[SU($n$)] theory arises as an IR fixed point of the following 3d $\mathcal{N} = 4$ Lagrangian theory

$$U(1) \text{------} U(2) \text{---} \cdots \text{---} U(n-1) \text{------} [\mathfrak{su}(n)_H], \tag{96}$$

where we have a bifundamental hyper between adjacent unitary gauge nodes, and $n$ fundamental hypers for $U(n-1)$ that are rotated by an

$$\mathfrak{f}_H = \mathfrak{su}(n)_H, \tag{97}$$

Higgs flavor symmetry algebra.

The genuine local operators charged under $\mathfrak{su}(n)_H$ arise from gauge invariant combinations of hypermultiplets. Such gauge invariant combinations all form representations of the Lie group $PSU(n)_H$ with Lie algebra $\mathfrak{su}(n)_H$. Thus, the Higgs branch 0-form symmetry group is

$$\mathcal{F}_H = PSU(n)_H = SU(n)_H/\mathbb{Z}_n, \tag{98}$$

where $SU(n)_H$ is the simply connected group with Lie algebra $\mathfrak{su}(n)_H$.

Equivalently, we can deduce the above 0-form symmetry group by putting the theory (96) on a non-trivial compact 3-manifold $M_3$. The charges of the vector and hypermultiplets allow us to turn on bundles for the group

$$\mathcal{S} = \frac{U(1) \times U(2) \times \cdots \times U(n-1) \times SU(n)_H}{\mathbb{Z}_n}. \tag{99}$$

The $\mathbb{Z}_n$ in the denominator is the diagonal combination of the $\mathbb{Z}_n$ subgroups of $U(1)$ centers of $U(i)$ gauge groups and the $\mathbb{Z}_n$ center of $SU(n)_H$. This diagonal combination leaves all vector and hyper multiplets invariant. Thus the Higgs flavor symmetry group is $PSU(n)_H$ because, according to (99), we can couple the theory (96) to non-trivial (in the sense that they cannot be lifted to $SU(n)_H$ bundles) background bundles for $PSU(n)_H$, provided we turn on non-trivial (in the sense that they cannot be lifted to $U(i)$ bundles) bundles for $U(i)/\mathbb{Z}_n$.

**A-type 0-form symmetry.** In addition to the $PSU(n)_H$ 0-form symmetry, the Lagrangian theory (96) admits a magnetic

$$\mathcal{F}_C^{\text{UV}} = U(1)_C^{n-1} = \prod_{i=1}^{n-1} U(1)_{C,i}, \tag{100}$$

0-form symmetry, where $U(1)_{C,i}$ is the magnetic symmetry arising from the $U(i)$ gauge node.

The above Coulomb 0-form symmetry enhances in the IR to an $\mathfrak{su}(n)_C$ Coulomb 0-form symmetry. The fundamental monopole operators associated to each node $i$ describe roots of $\mathfrak{su}(n)_C$, which carry charge 0 (mod $n$) under the center $\mathbb{Z}_n$ of $SU(n)_C$. As a consequence, all gauge monopole operators form representations of $SU(n)_C$ having charge 0 (mod $n$) under the center $\mathbb{Z}_n$. Thus the Coulomb 0-form symmetry group of the IR SCFT is

$$\mathcal{F}_C^{\text{IR}} = PSU(n)_C. \tag{101}$$

**Mixed 0-form anomaly.** There is a mixed 't Hooft anomaly between the $PSU(n)_H$ and $PSU(n)_C$ 0-form symmetries of T[SU($n$)]. The relevant flavor-gauge monopole operator is associated to a co-character of the structure group $\mathcal{S}$ appearing in (99) with winding number $1/n$ around the $U(1)$ center of each $U(i)$ gauge group and winding number $1/n$ around a $U(1)$

subgroup of the maximal torus of $SU(n)_H$. This is an allowed co-character because of the $\mathbb{Z}_n$ quotient appearing in the denominator of $\mathcal{S}$.

The mixed flavor-gauge monopole operator $O$ descends to a flavor monopole operator in the IR SCFT T[$SU(n)$], which is a non-genuine local operator lying at the end of a flavor vortex line defect. $O$ carries a charge $q_i = i/n$ under $U(1)_{C,i}$. This implies that the Dynkin coefficients $d_i$ of the weight of $\mathfrak{su}(n)_C$ carried by $O$ are

$$
\begin{aligned}
(d_1, d_2, \cdots, d_{n-2}, d_{n-1}) &= (2q_1 - q_2, -q_1 + 2q_2 - q_3, \cdots, -q_{n-3} + 2q_{n-2} - q_{n-1}, -q_{n-2} + 2q_{n-1}) \\
&= (0, 0, \cdots, 0, 1).
\end{aligned}
\tag{102}
$$

The charge of $O$ under the $\mathbb{Z}_n$ center of $SU(n)_C$ is computed in terms of $d_i$ as

$$
\sum_{i=1}^{n-1} i \times d_i \ (\mathrm{mod}\ n) = -1 \ (\mathrm{mod}\ n).
\tag{103}
$$

Using the analysis of [12], this fact is equivalent to a mixed 't Hooft anomaly between the Coulomb and Higgs 0-form symmetry groups of the IR T[$SU(n)$] SCFT

$$
\mathcal{A}_4^{\mathrm{IR}} = \exp\left(-\frac{2\pi i}{n} \int w_2^H \cup w_2^C\right),
\tag{104}
$$

where $w_2^C, w_2^H$ are $\mathbb{Z}_n$ valued obstruction classes capturing the obstruction of lifting background $PSU(n)_C, PSU(n)_H$ bundles to $SU(n)_C, SU(n)_H$ bundles.

### 3.2.2  $SU(n)_H$ gauging

Consider now the 3d $\mathcal{N} = 4$ theory

$$
\mathrm{T}[SU(n)] \relbar\joinrel\relbar SU(n)_H,
\tag{105}
$$

obtained by gauging the $\mathfrak{su}(n)_H$ flavor symmetry of T[$SU(n)$] by an $SU(n)_H$ gauge group. We can reach very close to this theory by flowing from the 3d $\mathcal{N} = 4$ quiver

$$
U(1) \relbar\joinrel\relbar U(2) \relbar\joinrel\relbar \cdots \relbar\joinrel\relbar U(n-1) \relbar\joinrel\relbar SU(n)_H,
\tag{106}
$$

if the gauge coupling for $SU(n)_H$ is extremely small compared to the $U(i)$ gauge couplings.

Note that (105) is a bad theory. We will later also discuss its good versions obtained by adding flavors for the $SU(n)_H$ gauge group.

**1-form symmetry.** This theory carries a

$$
\Gamma^{(1)} = \mathbb{Z}_n,
\tag{107}
$$

1-form symmetry, as can be seen by the following argument. After gauging we obtain Wilson line defects valued in representations of $SU(n)_H$. A genuine local operator of T[$SU(n)$] transforming in representation $R$ of $SU(n)_H$ becomes a non-genuine local operator of the gauged theory (105) that lives at the end of Wilson line defect in representation $R$. In other words, Wilson line in representation $R$ is screened in the gauged theory (105). From our previous discussion, we know that the genuine local operators transform only in representations of $PSU(n)_H$. Thus, Wilson lines transforming in representations of $SU(n)_H$ with non-zero charge (modulo $n$) under its $\mathbb{Z}_n$ center are left unscreened, implying that $\mathbb{Z}_n$ center of $SU(n)_H$ descends to a $\mathbb{Z}_n$ 1-form symmetry of the gauged theory (105).

**0-form symmetry.** There are two types of genuine local operators of the gauged theory (105) to consider:

1. The genuine local operators of T[SU($n$)] that are uncharged under $SU(n)_H$ descend to genuine local operators of the gauged theory (105). Since such operators form $PSU(n)_C$ representations before gauging, they also form $PSU(n)_C$ representations after gauging.

2. The flavor monopole operators of T[SU($n$)] associated to co-characters of $SU(n)_H$ become genuine local operators (non-fractional gauge monopole operators) after gauging, despite being non-genuine before gauging. Such monopole operators form $PSU(n)_C$ representations before gauging, so only lead to $PSU(n)_C$ representations after gauging.

Thus, the $PSU(n)_C$ 0-form symmetry of T[SU($n$)] is not impacted by the gauging procedure and the gauged theory (105) also has

$$\mathcal{F}_C = PSU(n)_C \,, \tag{108}$$

0-form symmetry.

**Is there a 2-group symmetry?** The question we would now like to address is whether the above $\mathbb{Z}_n$ 1-form and $PSU(n)_C$ 0-form symmetries combine to form a 2-group symmetry with a non-trivial Postnikov class. We claim that the answer is *negative*, due to the following reason.

The existence of such a 2-group symmetry requires the presence of a local operator $O$ in the gauged theory (105) which sits at the end of a Wilson line operator transforming in a representation of $PSU(n)_H$ and transforms in a representation of $SU(n)_C$ that is *not* an allowed representation of $PSU(n)_C$. This means that in the ungauged theory T[SU($n$)], $O$ must be a local operator (transforming in the same representations of $PSU(n)_H$ and $SU(n)_C$) of one of the following two types:

1. $O$ is a genuine local operator in T[SU($n$)]. But then $O$ must transform in a $PSU(n)_C$ representation because the Coulomb 0-form symmetry group of T[SU($n$)] is $PSU(n)_C$.

2. $O$ is a flavor monopole operator in T[SU($n$)] associated to a co-character of the group $SU(n)_H$. But then $O$ must transform in a $PSU(n)_C$ representation as already discussed above.

Thus, we conclude that there is no non-trivial 2-group symmetry in the gauged theory (105).

**Mixed 1-form 0-form 't Hooft anomaly.** Flavor monopole operators of T[SU($n$)] become (possibly fractional) gauge monopole operators in the gauged theory (105). This includes the operator $O$ discussed around equation (104), which becomes a fractional gauge monopole operator after gauging and can be converted into a local operator living at the end of the topological line defect generating the $\mathbb{Z}_n$ 1-form symmetry. The fact that $O$ transforms in a representation of $SU(n)_C$ having charge $-1$ (mod $n$) under its $\mathbb{Z}_n$ center is equivalent to the mixed 't Hooft anomaly between the $\mathbb{Z}_n$ 1-form symmetry and $PSU(n)_C$ 0-form symmetry of the gauged theory (68)

$$\mathcal{A}_4 = \exp\left(-\frac{2\pi i}{n} \int B_2 \cup w_2^C\right), \tag{109}$$

where $B_2$ is the background field for the $\mathbb{Z}_n$ 1-form symmetry. This anomaly can also be derived as a consequence of the anomaly (104) using the identification $B_2 = w_2^H$.

**Adding flavors.** We are interested in understanding the global form of the Coulomb symmetry group of the IR SCFT obtained (for large enough $N$) from the UV 3d $\mathcal{N} = 4$ Lagrangian theory

$$U(1) \longrightarrow U(2) \longrightarrow \cdots \longrightarrow U(n-1) \longrightarrow SU(n)_H \longrightarrow N\,\mathsf{F}. \tag{110}$$

Note that the corresponding IR SCFT can also be obtained by starting from the good version

$$\mathsf{T[SU(n)]} \longrightarrow SU(n)_H \longrightarrow N\,\mathsf{F}, \tag{111}$$

of the theory (105), obtained from (105) by adding $N$ fundamental hypers for $SU(n)_H$. The relationship between theories (110) and (111) is that the theory (110) flows very close to the theory (111) at intermediate energy scales if the gauge coupling for $SU(n)_H$ is extremely small compared to the $U(i)$ gauge couplings.

We can thus use the symmetry properties of the theory (105) to deduce symmetry properties for the IR SCFT associated to (110) as follows. The additional flavors in the theory (111) screen the fundamental Wilson line of $SU(n)_H$ and thus the $\mathbb{Z}_n$ 1-form symmetry of the theory (105) is lost in the theory (111). On the other hand, the only new genuine local operators we obtain are gauge invariant combinations of these hypers. These carry trivial charge under $PSU(n)_C$, and hence the Coulomb 0-form symmetry group of the theory (111) is $PSU(n)_C$. For generic $N$, there is no enhancement of $PSU(n)_C$ and the IR SCFT originating from (110) (which is the same as the IR SCFT originating from (111)) has Coulomb 0-form symmetry group

$$\mathcal{F}_C^{\mathrm{IR}} = PSU(n)_C. \tag{112}$$

### 3.2.3 Other $\mathfrak{su}(n)_H$ gaugings

We can also consider gauging $SU(n)_H/\mathbb{Z}_m$ (where $\mathbb{Z}_m < \mathbb{Z}_n$ is a subgroup, i.e. $m|n$) to obtain the 3d $\mathcal{N} = 4$ theory

$$\mathsf{T[SU(n)]} \longrightarrow SU(n)_H/\mathbb{Z}_m. \tag{113}$$

One can reach very close to this theory by flowing from a 3d $\mathcal{N} = 4$ Lagrangian theory

$$\mathfrak{u}(1) \longrightarrow \mathfrak{u}(2) \longrightarrow \cdots \longrightarrow \mathfrak{u}(n-1) \longrightarrow \mathfrak{su}(n)_H \tag{114}$$

(where we have only displayed gauge algebras), with the gauge group

$$\mathcal{G} = \frac{U(1) \times U(2) \times \cdots \times U(n-1) \times SU(n)_H}{\mathbb{Z}_m}, \tag{115}$$

where the $\mathbb{Z}_m$ being quotiented out is the $\mathbb{Z}_m$ subgroup of the $\mathbb{Z}_n$ group appearing in the denominator of (99). Note that (113) is again a bad theory just like (105). Later, we also consider its good versions obtained by adding adjoint flavors for the $SU(n)_H/\mathbb{Z}_m$ gauge group.

**1-form symmetry group.** In fact, this theory (113) can be obtained by gauging the $\mathbb{Z}_m$ 1-form symmetry of the theory (105). Consequently, there is a residual

$$\Gamma^{(1)} = \mathbb{Z}_p, \tag{116}$$

1-form symmetry in the theory (113) where $p = n/m$.

**0-form symmetry group.** There is also thus a dual $\mathbb{Z}_m$ 0-form symmetry in (113) alongside the residual $PSU(n)_C$ 0-form symmetry descending from (105). By similar arguments as in the previous subsection on $\mathsf{T[SU(2)]}$, the two 0-form symmetries combine non-trivially and the full 0-form symmetry group of (113) is

$$\mathcal{F}_C = SU(n)_C/\mathbb{Z}_p. \tag{117}$$

**Mixed 1-form 0-form 't Hooft anomaly.** There is a mixed 't Hooft anomaly between $\mathbb{Z}_p$ 1-form and $SU(n)_C/\mathbb{Z}_p$ 0-form symmetries arising as a residue of the anomaly (109)

$$\mathcal{A}_4 = \exp\left(-\frac{2\pi i}{p}\int B_2 \cup w_2^C\right), \tag{118}$$

where $w_2^C$ is the $\mathbb{Z}_p$ valued obstruction class for lifting $SU(n)_C/\mathbb{Z}_p$ bundles to $SU(n)_C$ bundles, and $B_2$ is the $\mathbb{Z}_p$ valued background field for 1-form symmetry.

**Particular case: $m = n$.** In this particular case, we are studying a $PSU(n)_H$ gauging of T[SU(n)]. There is no 1-form symmetry and the 0-form symmetry group is $SU(n)_C$.

**Adding adjoint flavors.** We can add $N$ adjoint flavors for $SU(n)_C/\mathbb{Z}_m$. For large enough $N$, this flows to a 3d $\mathcal{N} = 4$ SCFT in the IR. The global form of the Coulomb symmetry group and mixed anomaly between Coulomb 0-form symmetry and 1-form symmetry in the IR SCFT for generic $N$ are the same as those described above.

### 3.2.4 $U(n)$ gauging

We are interested in understanding the global form of Coulomb and Higgs symmetry groups of the IR SCFT obtained (for large enough $N$) from the UV 3d $\mathcal{N} = 4$ Lagrangian theory

$$U(1) \text{——} U(2) \text{—} \cdots \text{——} U(n) \text{——} [\mathfrak{su}(n)_H], \tag{119}$$

where we have $N$ hypers transforming in fundamental representation of $U(n)$ along with bifundamental hypers between adjacent $U(i)$ gauge nodes.

**0-form symmetry algebras.** As shown in (119), there is a

$$\mathfrak{f}_H = \mathfrak{su}(N)_H, \tag{120}$$

Higgs branch symmetry rotating the $N$ fundamental hypers.

We will focus on the case $N > n + 1$ for which the IR Coulomb symmetry algebra is

$$\mathfrak{f}_C^{\text{IR}} = \mathfrak{u}(n)_C = \mathfrak{su}(n)_C \oplus \mathfrak{u}(1)_C. \tag{121}$$

The $N = n+1$ case has a further enhancement of IR Coulomb symmetry algebra to $\mathfrak{su}(n+1)_C$.

The $\mathfrak{su}(n)_C$ subalgebra of $\mathfrak{f}_C^{\text{IR}}$ is usually associated to the balanced nodes provided by the $U(i)$ gauge groups for $1 \le i \le n-1$, and the $\mathfrak{u}(1)_C$ subalgebra of $\mathfrak{f}_C^{\text{IR}}$ is usually associated to the unbalanced node provided by the $U(n)$ gauge group. However, as before we will need a more precise identification of these subalgebras.

**0-form symmetry groups.** The Higgs 0-form symmetry group of the IR SCFT is the same as that of the UV Lagrangian theory

$$\mathcal{F}_H = PSU(N)_H = SU(N)_H/\mathbb{Z}_N. \tag{122}$$

To deduce the Coulomb 0-form symmetry group of the IR SCFT, we need a precise identification of the Dynkin coefficients $d_i$ for $\mathfrak{su}(n)_C$ weights in terms $U(1)_{C,i}$ charges. This is the same as discussed around equation (102), except now $d_{n-1}$ is modified to

$$d_{n-1} = -q_{n-2} + 2q_{n-1} - q_n. \tag{123}$$

Moreover, the $\mathfrak{u}(1)_C$ factor has a global form $U(1)_C$ such that the charge $q_C$ under $U(1)_C$ is

$$q_C = q_n. \tag{124}$$

Now we see that the fundamental monopole operators coming from the $U(n)$ gauge node transform in anti-fundamental representation of $\mathfrak{su}(n)_C$ and simultaneously have charge $+1$ under $\mathfrak{u}(1)_C$. Thus the Coulomb 0-form symmetry group of the IR SCFT is

$$\mathcal{F}_C^{\text{IR}} = U(n)_C = \frac{SU(n)_C \times U(1)_C}{\mathbb{Z}_n}. \tag{125}$$

# 4 General symmetry and anomaly analysis for 3d $\mathcal{N} = 4$ SCFTs

Now we are ready to describe the most general result of the considerations of this paper. Consider a 3d $\mathcal{N} = 4$ good quiver gauge theory composed of unitary and special unitary gauge algebras and no Higgs flavor algebras. Let us write the gauge algebra as

$$\mathfrak{g} = \bigoplus_i \mathfrak{g}_i, \tag{126}$$

where each $\mathfrak{g}_i$ is either $\mathfrak{su}(n_i)$ or $\mathfrak{u}(n_i)$. We assume there is at least one special unitary node and all special unitary nodes are unbalanced. Also let $G_i$ be $SU(n_i)$ or $U(n_i)$ for the two cases. The matter content is comprised entirely of bifundamental hypers. Say we have $m_{ij} \geq 0$ bifundamental hypers between $\mathfrak{g}_i$ and $\mathfrak{g}_j$. We assume that the quiver is connected, which means that we can go from any node $i$ to any other node $j$ by choosing a sequence of nodes $i_a$ for $0 \leq a \leq b$ such that $i_0 = i$, $i_b = j$ and $m_{i_a i_{a+1}} > 0$. Moreover, the balanced unitary nodes are taken to form the Dynkin diagram of a finite semi-simple Lie algebra

$$\mathfrak{f} = \bigoplus_a \mathfrak{f}_a, \tag{127}$$

where each $\mathfrak{f}_a$ is a finite simple Lie algebra. Let $F_a$ be the simply connected group associated to $\mathfrak{f}_a$ with center $Z_a$ and define $F = \prod_a F_a$ with center $Z = \prod_a Z_a$. Let $\mathcal{U}$ be the set of unbalanced unitary nodes. Let $\mathfrak{u}(1)_i$ for a node $i \in \mathcal{U}$ be the associated Coulomb 0-form symmetry algebra and $U(1)_i$ be a group with Lie algebra $\mathfrak{u}(1)_i$ such that the fundamental BPS monopoles associated to node $i$ have charge $g\delta_{ij}$ under $U(1)_j$, where $g$ is defined around equation (130) below and $\delta_{ij}$ is the Kronecker delta. We have scaled the $U(1)_i$ charges of fundamental monopoles by $g$ for later convenience in computing 't Hooft anomalies. The Coulomb 0-form symmetry algebra of the corresponding 3d $\mathcal{N} = 4$ IR SCFT is

$$\mathfrak{f}_C^{\text{IR}} = \bigoplus_{i \in \mathcal{U}} \mathfrak{u}(1)_i \oplus \mathfrak{f}. \tag{128}$$

Let us define $F_C^{\text{IR}} = \prod_{i \in \mathcal{U}} U(1)_i \times F$ with center $Z_C^{\text{IR}} = \prod_{i \in \mathcal{U}} U(1)_i \times Z$.

## 4.1 Symmetries

**1-form symmetry group.** Consider choosing first the gauge group

$$\mathcal{G} = \prod_i G_i. \tag{129}$$

Then the theory with the above-described matter content has a 1-form symmetry group given by

$$\Gamma^{(1)} = \mathbb{Z}_g, \tag{130}$$

where $g$ is the GCD (greatest common divisor) of all $n_i$ for which $\mathfrak{g}_i = \mathfrak{su}(n_i)$.

**(Coulomb) 0-form symmetry group.** Let us determine the Coulomb 0-form symmetry group $\mathcal{F}_C^{\mathrm{IR}}$ of the IR SCFT. Each node $i \in \mathcal{U}$ provides a genuine local operator $O_i$ in the IR SCFT whose charge under $Z_C^{\mathrm{IR}}$ is what we want to determine. This local operator can be chosen to be IR image of any fundamental BPS monopole associated to the node $i$.

First of all, $O_i$ has charge $q_{i,j} = g\delta_{ij}$ under $U(1)_j$ factor of $Z_C^{\mathrm{IR}}$, where $j \in \mathcal{U}$. The charge $q_{i,a}$ of $O_i$ under $Z_a$ is given by

$$q_{i,a} = -\sum_{j \in \mathcal{N}_a} m_{i,j} n_{a,j} \in \widehat{Z}_a \,, \tag{131}$$

where $\mathcal{N}_a$ is the set of nodes forming the Dynkin diagram of $\mathfrak{f}_a$ and $n_{a,j} \in \widehat{Z}_a$ is the charge of the representation of $\mathfrak{f}_a$ with highest weight having Dynkin coefficients $d_i = \delta_{ij}$ for $i \in \mathcal{N}_a$. Combining the charges $q_{i,j}$ for all $j \in \mathcal{U}$ and charges $q_{i,a}$ for all $a$, we obtain a charge

$$q_i \in \widehat{Z}_C^{\mathrm{IR}} \,, \tag{132}$$

of $O_i$ under $Z_C^{\mathrm{IR}}$. The charges $q_i$ for all $i \in \mathcal{U}$ span a subgroup $Y_C^{\mathrm{IR}}$ of the abelian group $\widehat{Z}_C^{\mathrm{IR}}$. We thus obtain the information of a surjective map

$$\widehat{Z}_C^{\mathrm{IR}} \to \widehat{\mathcal{Z}}_C^{\mathrm{IR}} := \frac{\widehat{Z}_C^{\mathrm{IR}}}{Y_C^{\mathrm{IR}}} \,, \tag{133}$$

which can be Pontryagin dualized to an injective map

$$\mathcal{Z}_C^{\mathrm{IR}} \to Z_C^{\mathrm{IR}} \,, \tag{134}$$

providing a subgroup $\mathcal{Z}_C^{\mathrm{IR}}$ of $Z_C^{\mathrm{IR}}$. The Coulomb 0-form symmetry group of the IR SCFT is

$$\mathcal{F}_C^{\mathrm{IR}} = \frac{F_C^{\mathrm{IR}}}{\mathcal{Z}_C^{\mathrm{IR}}} \,. \tag{135}$$

**Visual representation.** The charges $q_{i,a}$ of operators $O_i$ under the centers $Z_a$ of Coulomb flavor algebras $\mathfrak{f}_a$ can be deduced visually from the UV quiver. First of all, note that $q_{i,a}$ is the same as the charge under $Z_a$ of the representation

$$\bigotimes_{j \in \mathcal{N}_a} \mathsf{F}_j^{m_{i,j}} \,, \tag{136}$$

of $\mathfrak{f}_a$, where $\mathsf{F}_j$ is the so-called 'fundamental representation associated to node $j$' of the Dynkin diagram of $\mathfrak{f}_a$, which is the representation with highest weight having Dynkin coefficients $d_i = \delta_{ij}$ for $i \in \mathcal{N}_a$. This representation is deduced visually from the UV quiver: we just see how many times the node $i \in \mathcal{U}$ hits a node $j$ in $\mathcal{N}_a$ and include that many copies of the representation $\mathsf{F}_j$ of $\mathfrak{f}_a$.

## 4.2 Anomaly

There is a mixed 't Hooft anomaly between $\Gamma^{(1)}$ and $\mathcal{F}_C^{\mathrm{IR}}$ which is computed using the charge of a fractional gauge monopole operator $O$ associated to co-character of the group

$$G = \frac{\mathcal{G}}{\mathbb{Z}_g} \,, \tag{137}$$

where $\mathbb{Z}_g$ being quotiented out is the diagonal combination of the $\mathbb{Z}_g$ subgroup of the $U(1)$ center of each unitary gauge group and the $\mathbb{Z}_g$ subgroup of the $\mathbb{Z}_{n_i}$ center of the gauge group

$SU(n_i)$ for each special unitary gauge group. The co-character associated to $O$ has winding number $1/g$ around the $U(1)$ center of each unitary gauge group and winding number $1/g$ around a $U(1)$ subgroup of the maximal torus of $SU(n_i)$ for each special unitary gauge group.

The charge of $O$ under $U(1)_i$ for $i \in \mathcal{U}$ is $q_i$. The charge $q_a$ of $O$ under $Z_a$ is the same as the charge of the representation of $\mathfrak{f}_a$ having highest weight with Dynkin coefficients

$$d_i = \sum_{j \in \mathcal{N}_a} M_{a,ij} \frac{n_j}{g} - \sum_{j \in \mathcal{U}} m_{i,j} \frac{n_j}{g}, \tag{138}$$

for $i \in \mathcal{N}_a$, where $M_{a,ij}$ is the Cartan matrix of $\mathfrak{f}_a$ and $n_j$ is the rank of the gauge algebra $\mathfrak{u}(n_j)$ associated to the node $j$. Combining the $q_i$ and $q_a$ charges, we obtain a charge $q_O \in \widehat{Z}_C^{\mathrm{IR}}$ of $O$ under $Z_C^{\mathrm{IR}}$. Projecting it using the map (133), we obtain an element $q \in \widehat{\mathcal{Z}}_C^{\mathrm{IR}}$, letting us define a homomorphism

$$\gamma : \Gamma^{(1)} \to \widehat{\mathcal{Z}}_C^{\mathrm{IR}}, \tag{139}$$

via

$$\Gamma^{(1)} = \mathbb{Z}_g \ni 1 \mapsto q \in \widehat{\mathcal{Z}}_C^{\mathrm{IR}}. \tag{140}$$

The 't Hooft anomaly between the 1-form and 0-form symmetries of the IR SCFT is then

$$\mathcal{A}_4^{\mathrm{IR}} = \exp\left( 2\pi i \int \gamma(B_2) \cup w_2^C \right), \tag{141}$$

where $B_2$ is the $\Gamma^{(1)}$ valued background field for the 1-form symmetry, $w_2^C$ is the $\mathcal{Z}_C^{\mathrm{IR}}$ valued class capturing the obstruction of lifting $\mathcal{F}_C^{\mathrm{IR}}$ bundles to $F_C^{\mathrm{IR}}$ bundles, and the cup product uses the natural pairing $\widehat{\mathcal{Z}}_C^{\mathrm{IR}} \times \mathcal{Z}_C^{\mathrm{IR}} \to \mathbb{R}/\mathbb{Z}$.

**Visual representation.** The charges $q_a$ of the operator $O$ under the centers $Z_a$ of Coulomb flavor algebras $\mathfrak{f}_a$ can be deduced visually from the UV quiver. First of all, note that $q_a$ is the same as the charge under $Z_a$ of the representation

$$\bigotimes_{i \in \mathcal{S}} \bigotimes_{j \in \mathcal{N}_a} \mathsf{F}_j^{\frac{n_i m_{i,j}}{g}}, \tag{142}$$

of $\mathfrak{f}_a$, where $\mathcal{S}$ is the set of special unitary gauge nodes. To see this, one needs to use the balancing condition.

This representation is deduced visually from the UV quiver: for each special unitary node $i \in \mathcal{S}$, we just see how many times $i$ hits a node $j$ in $\mathcal{N}_a$ and include that many copies of the representation $\mathsf{F}_j$ of $\mathfrak{f}_a$ weighted by a factor of $n_i/g$.

## 4.3 Other gauge groups

We can change the gauge group by gauging a subgroup $\mathbb{Z}_h \subseteq \mathbb{Z}_g$ of the 1-form symmetry. The resulting gauge group is

$$\mathcal{G}_h = \mathcal{G}/\mathbb{Z}_h. \tag{143}$$

The 1-form symmetry group of the resulting IR SCFT is now

$$\Gamma_h^{(1)} = \mathbb{Z}_g/\mathbb{Z}_h = \mathbb{Z}_k, \tag{144}$$

where $k = g/h$.

Because of the extra $\mathbb{Z}_h$ quotient in the new gauge group $\mathcal{G}_h$, some of the gauge monopole operators which were fractional (i.e. were non-genuine local operators) now become non-fractional gauge monopole operators (i.e. become genuine local operators). We need to account for center charges of these new genuine local operators to compute the Coulomb 0-form

symmetry group $\mathcal{F}^{\mathrm{IR}}_{C,h}$ of the new IR SCFT. To account for these new charges, it is sufficient to consider the contribution of a single monopole operator $O_h$ for $\mathcal{G}_h$ with co-characters having winding number $k$ times the winding numbers associated to the fractional gauge monopole operator $O$ considered above. The charge $q_{i,h}$ of $O_h$ under $U(1)_i$ for $i \in \mathcal{U}$ is $k$, and the charge $q_{a,h}$ of $O_h$ under $Z_a$ is the same as the charge of the representation of $\mathfrak{f}_a$ having highest weight with Dynkin coefficients $d_{i,h} = kd_i$, where $d_i$ are defined in (138). Combining the $q_{i,h}$ and $q_{a,h}$ charges for various values of $i$ and $a$, we obtain a charge $q_h \in \widehat{Z}^{\mathrm{IR}}_C$. Appending $q_h$ to $Y^{\mathrm{IR}}_C$ and spanning, we obtain a larger subgroup $Y^{\mathrm{IR}}_{C,h}$ of $\widehat{Z}^{\mathrm{IR}}_C$. This provides a surjective map

$$\widehat{Z}^{\mathrm{IR}}_C \to \widehat{\mathscr{Z}}^{\mathrm{IR}}_{C,h} := \frac{\widehat{Z}^{\mathrm{IR}}_C}{Y^{\mathrm{IR}}_{C,h}}, \tag{145}$$

whose Pontryagin dual is an injective map

$$\mathscr{Z}^{\mathrm{IR}}_{C,h} \to Z^{\mathrm{IR}}_C, \tag{146}$$

providing a subgroup $\mathscr{Z}^{\mathrm{IR}}_{C,h}$ of $Z^{\mathrm{IR}}_C$. The Coulomb 0-form symmetry group of the new IR SCFT is

$$\mathcal{F}^{\mathrm{IR}}_{C,h} = \frac{F^{\mathrm{IR}}_C}{\mathscr{Z}^{\mathrm{IR}}_{C,h}}. \tag{147}$$

There is a residual mixed 't Hooft anomaly between $\Gamma^{(1)}_h$ and $\mathcal{F}^{\mathrm{IR}}_{C,h}$ descending from (141). This is still a consequence of the operator $O$ which remains a fractional gauge monopole operator for $h < g$. Its charge is still $q_O \in \widehat{Z}^{\mathrm{IR}}_C$ which projects to an element $q_h \in \widehat{\mathscr{Z}}^{\mathrm{IR}}_{C,h}$, letting us define a homomorphism

$$\gamma_h : \Gamma^{(1)}_h \to \widehat{\mathscr{Z}}^{\mathrm{IR}}_{C,h}, \tag{148}$$

via

$$\Gamma^{(1)} = \mathbb{Z}_k \ni 1 \mapsto q_h \in \widehat{\mathscr{Z}}^{\mathrm{IR}}_{C,h}. \tag{149}$$

The 't Hooft anomaly between the 1-form and 0-form symmetries of the new IR SCFT is then

$$\mathcal{A}^{\mathrm{IR}}_{4,h} = \exp\left( 2\pi i \int \gamma_h(B_{2,h}) \cup w^C_{2,h} \right), \tag{150}$$

where $B_{2,h}$ is the $\Gamma^{(1)}_h$ valued background field for the new 1-form symmetry, $w^C_{2,h}$ is the $\mathscr{Z}^{\mathrm{IR}}_{C,h}$ valued class capturing the obstruction of lifting $\mathcal{F}^{\mathrm{IR}}_{C,h}$ bundles to $F^{\mathrm{IR}}_C$ bundles, and the cup product uses the natural pairing $\widehat{\mathscr{Z}}^{\mathrm{IR}}_{C,h} \times \mathscr{Z}^{\mathrm{IR}}_{C,h} \to \mathbb{R}/\mathbb{Z}$.

## 4.4 Including flavors

Let us now ungauge a few special unitary gauge algebras in the above UV theory, converting those gauge nodes into flavor nodes. The resulting theory is still a good theory and flows to a 3d $\mathcal{N} = 4$ SCFT in the IR. Let the set $\mathcal{R}$ parametrize the nodes which remain as gauge nodes. We choose the gauge group to be

$$\mathcal{G}_R = \prod_{i \in \mathcal{R}} G_i. \tag{151}$$

Because of the presence of flavors, the theory has no 1-form symmetry

$$\Gamma^{(1)} = 0. \tag{152}$$

The Higgs flavor symmetry algebra is taken to be (there might be some extra abelian factors that we ignore)

$$\mathfrak{f}_H = \bigoplus_{i \notin \mathcal{R}} \mathfrak{su}(n_i) \,. \tag{153}$$

The structure group of the UV theory including Higgs flavor symmetries is

$$\mathcal{S}_R = \mathcal{G}/\mathbb{Z}_g \,. \tag{154}$$

In particular, the Higgs 0-form symmetry group of the UV theory and the corresponding IR SCFT is

$$\mathcal{F}_H = \frac{\prod_{i \notin \mathcal{R}} SU(n_i)}{\mathbb{Z}_g} \,. \tag{155}$$

The Coulomb 0-form symmetry group is the same as before $\mathcal{F}_C^{\mathrm{IR}}$.

The fractional gauge monopole $O$ discussed above is now instead a mixed flavor-gauge monopole operator providing a mixed 't Hooft anomaly between the Higgs and Coulomb 0-form symmetries

$$\mathcal{A}_{4,R}^{\mathrm{IR}} = \exp\left( 2\pi i \int \gamma(w_2^H) \cup w_2^C \right), \tag{156}$$

where $w_2^H$ is the $\mathbb{Z}_g$ valued class capturing the obstruction of lifting $\mathcal{F}_H$ bundles to $\prod_{i \notin \mathcal{R}} SU(n_i)$ bundles and $\gamma$ is the homomorphism appearing in (139).

## 4.5 Special case 1: Single special unitary node

In this and the following subsections, we discuss two special cases for which the symmetry groups and anomalies of the IR SCFT take a simple form. These two special cases will be used frequently in the rest of this paper.

In this subsection, we consider the first special case, which occurs when we have a single (unbalanced) special unitary gauge node carrying $\mathfrak{su}(n)_H$ gauge algebra.

First choose the gauge group

$$\mathcal{G} = \prod_i U(n_i) \times SU(n)_H \,. \tag{157}$$

The 1-form symmetry is

$$\Gamma^{(1)} = \mathbb{Z}_n \,. \tag{158}$$

As explained around (136), the 0-form symmetry group $\mathcal{F}_C^{\mathrm{IR}}$ is read simply from the positions where the unbalanced unitary nodes hit the balanced unitary nodes.

The IR SCFT has a mixed 't Hooft anomaly between the 1-form and 0-form symmetry groups. The charge $q_a$ under $Z_a$ of the fractional gauge monopole operator $O$ is read simply visually from the UV quiver, and coincides with the charge of the representation

$$\bigotimes_{i \in \mathcal{N}_a} \mathsf{F}_i^{m_i} \,, \tag{159}$$

of $\mathfrak{f}_a$, where $m_i$ is the number of bifundamental hypers between the $SU(n)_H$ gauge node and the balanced unitary gauge node $i \in \mathcal{N}_a$. That is, we simply observe the number of times the $SU(n)_H$ gauge node hits the balanced node $i$ and include that many copies of the representation $\mathsf{F}_i$. Combining the charges $q_a \in \widehat{Z}_a$ with the charge $n_i$ under each Coulomb symmetry $U(1)_i$ associated to unbalanced unitary node $i \in \mathcal{U}$, we obtain the charge $q \in \widehat{Z}$ of $O$, which provides the homomorphism (139) appearing in the mixed anomaly (141).

Let us gauge the 1-form symmetry group $\Gamma^{(1)} = \mathbb{Z}_n$. The new gauge group is

$$\mathcal{G}_n = \frac{\prod_i U(n_i) \times SU(n)_H}{\mathbb{Z}_n} \,. \tag{160}$$

There is no residual 1-form symmetry and the Coulomb 0-form group is modified by the presence of the operator $O$ which is now a genuine local operator. Its charges $q_a$ described above need to be accounted to compute the new Coulomb 0-form symmetry group.

We can instead ungauge $SU(n)_H$. The gauge group is now

$$\mathcal{G}_R = \prod_i U(n_i) \,. \tag{161}$$

There is a Higgs 0-form symmetry algebra

$$\mathfrak{f}_H = \mathfrak{su}(n)_H \,. \tag{162}$$

Both the UV gauge theory and the IR SCFT have Higgs 0-form symmetry group as

$$\mathcal{F}_H = PSU(n)_H \,. \tag{163}$$

The Coulomb 0-form symmetry group is still as before the ungauging. There is a mixed 't Hooft anomaly between the Higgs and Coulomb 0-form symmetry groups. The anomaly is described completely by the charges $q_a$ of the operator $O$ discussed above, which is now a mixed flavor-gauge monopole operator.

**Example.**   Many of the results of the previous section 3 can be arrived at by a simple application of the above general analysis. The anomaly (104) of $T[SU(n)]$ is a consequence of the fact that the flavor node $\mathfrak{su}(n)_H$ intersects the $\mathfrak{su}(n)_C$ balanced quiver at the location of the anti-fundamental node.

Similarly, the anomaly (109) is a consequence of the fact that the unbalanced $SU(n)_H$ gauge node intersects the $\mathfrak{su}(n)_C$ balanced quiver at the location of the anti-fundamental node. Upon gauging $\mathbb{Z}_n$ 1-form symmetry, it modifies the $PSU(n)_C$ 0-form symmetry to $SU(n)_C$ 0-form symmetry as discussed in the paragraph on special case $m = n$ of section 3.2.3.

## 4.6   Special case 2: Single unbalanced unitary node

In this subsection, we consider the second special case, which occurs when we have a single unbalanced unitary gauge node $U(n)$, and no special unitary gauge nodes. We require the presence of at least one flavor node. The gauge group is taken to be

$$\mathcal{G} = \prod_i U(n_i) \times U(n) \,. \tag{164}$$

There is no 1-form symmetry, and the Coulomb 0-form symmetry algebra of the IR SCFT is

$$\mathfrak{f}_C^{\text{IR}} = \mathfrak{f} \oplus \mathfrak{u}(1)_C \,. \tag{165}$$

The only non-trivial charge under its center $Z \times U(1)_C$ is provided by fundamental gauge monopole operators associated to the $U(1)_C$ gauge node. The charge $q_C$ of such an operator $O$ under $U(1)_C$ is $+1$ and the charge $q_a$ under $Z_a$ is the same as that of the representation

$$\bigotimes_{i \in \mathcal{N}_a} \mathsf{F}_i^{m_i} \,, \tag{166}$$

of $\mathfrak{f}_a$, where $m_i$ is the number of bifundamental hypers between the $U(n)$ gauge node and the balanced unitary gauge node $i \in \mathcal{N}_a$. That is, to compute $q_a$ we simply observe the number of times the $U(n)$ node hits the node $i$ and include that many copies of the representation $\mathsf{F}_i$. This allows us to compute the Coulomb 0-form symmetry group $\mathcal{F}_C^{\text{IR}}$ of the IR SCFT.

**Example.** Some of the results of the previous section 3 can be arrived at by a simple application of the above general analysis. The $U(n)_C$ 0-form symmetry group of $U(n)$ gauging of $T[SU(n)]$ appearing in equation (125) is a straightforward consequence of the fact that the unbalanced $U(n)$ gauge node intersects the balanced $\mathfrak{su}(n)_C$ quiver at the location of anti-fundamental node.

# 5 Consistency checks

In this section we will provide various consistency checks of our results detailed in the previous section. One central application is to magnetic quivers of 4d and 5d SCFTs with 8 supercharges.

## 5.1 Class S

In this subsection, we apply the methods of previous section to deduce the Coulomb 0-form symmetry groups of magnetic quivers (MQs) associated to Class S theories of $A_{n-1}$ type [74]. These MQs, derived in [75], are 3d quiver gauge theories that are mirror to the circle compactification of 4d $\mathcal{N} = 2$ Class S theories. Thus the Coulomb 0-form symmetry groups of these MQs should capture the usual Higgs flavor symmetry groups of 4d $\mathcal{N} = 2$ Class S theories. The computation of flavor symmetry groups of Class S theories was described recently in [66]. We demonstrate a match of results obtained using our methods against the results obtained using their methods, and illustrate it with three examples.

Alternatively, one can view this subsection as providing the correct global form of the MQs of Class S theories of $A_{n-1}$ type. That is, we provide the global form of the gauge group that should be associated to the gauge algebra of the MQ described in [75]. This global form of MQ is deduced by matching symmetries of MQ with the symmetries of the Class S theory.

### 5.1.1 General matching

**Symmetries of class S theory.** Consider a Class S theory arising from the sphere compactification of a 6d $\mathcal{N} = (2,0)$ SCFT of $A_{n-1}$ type with $k$ regular untwisted punctures. Each puncture $\mathcal{P}_i$ is characterized by a partition $\rho_i$ of $n$. Let $\rho_{i,j}$ be the elements of the partition where $j$ takes values in $1 \leq j \leq |\rho_i|$, and $|\rho_i|$ is the number of elements in the partition. We order $\rho_{i,j}$ such that $\rho_{i,j} \geq \rho_{i,j+1}$.

The flavor symmetry algebra $\mathfrak{f}_i$ associated to the puncture $\mathcal{P}_i$ is encoded in the partition $\rho_i$ according to the following standard rule. Define $j_1$ such that $\rho_{i,j} = \rho_{i,1}$ for all $j \leq j_1$. Define $j_2$ such that $\rho_{i,j} = \rho_{i,j_1+1}$ for all $j_1 < j \leq j_2$. We continue defining $j_a$ in this fashion until we reach $j = |\rho_i|$. Let $b$ be the total number of $j_a$. Then,

$$\mathfrak{f}_i = \bigoplus_{a=1}^{b} \mathfrak{su}(j_a - j_{a-1}) \oplus \mathfrak{u}(1)^{b-1}, \tag{167}$$

with $j_0 := 0$, and $\mathfrak{su}(1)$ being the trivial Lie algebra.

Let us define $F_i$ to be the following Lie group associated to the algebra $\mathfrak{f}_i$

$$F_i = \prod_{a=1}^{b} SU(j_a - j_{a-1}) \times U(1)^{b-1}, \tag{168}$$

where $SU(1)$ denotes the trivial group. The flavor symmetry group $\mathcal{F}$ of the Class S theory is obtained as a quotient

$$\mathcal{F} = \frac{\prod_i F_i}{\mathcal{Z}}. \tag{169}$$

Let $Z_F$ be the center of $F := \prod_i F_i$. The group $\mathcal{Z} \subseteq Z_F$ is obtained by taking Pontryagin dual of a surjective map

$$\widehat{Z}_F \to \widehat{\mathcal{Z}} = \widehat{Z}_F / Y_F , \tag{170}$$

where $Y_F$ is a subgroup of $\widehat{Z}_F$ whose computation was described in [66]. First of all, there is a contribution $Y_{F,\mathcal{P}_i} \subseteq Y_F$ coming from each puncture $\mathcal{P}_i$, and then there is a contribution $\widetilde{Y}_F \subseteq Y_F$ coming from all punctures put together. The group $Y_F$ is recovered as the combined span of all $Y_{F,\mathcal{P}_i}$ and $\widetilde{Y}_F$ inside $\widehat{Z}_F$. Finally, as discussed in [76] the 1-form symmetry group of the Class S theory is trivial.

**Review of the magnetic quiver.** Let us now review the magnetic quiver of the Class S theory discussed in [75]. We first associate a sub-quiver to each puncture $\mathcal{P}_i$ which can be described as the following 3d $\mathcal{N} = 4$ Lagrangian theory

$$[\mathfrak{su}(n)_H] \longrightarrow \mathfrak{u}(n_{i,1}) \longrightarrow \mathfrak{u}(n_{i,2}) \longrightarrow \cdots \longrightarrow \mathfrak{u}(n_{i,|\rho_i|-1}), \tag{171}$$

where

$$n_{i,J} = n - \sum_{j=1}^{J} \rho_{i,j} . \tag{172}$$

We see that the gauge nodes for $j_{a-1} < J < j_a$ are balanced for $1 \le a \le b$, and give rise to an emergent $\bigoplus_{a=1}^{b} \mathfrak{su}(j_a - j_{a-1})_C$ Coulomb symmetry in the IR. Moreover, the gauge nodes $J = j_a$ for $1 \le a \le j_{b-1}$ are unbalanced and give rise to $\mathfrak{u}(1)_C^{\oplus(b-1)}$ Coulomb symmetry in the IR. Thus, the contribution of this sub-quiver to the IR Coulomb 0-form symmetry algebra matches the flavor symmetry algebra $\mathfrak{f}_i$ associated to the puncture $\mathcal{P}_i$ shown in (167).

The full magnetic quiver of the Class S theory is then obtained by gauging the diagonal $\mathfrak{su}(n)_H$ symmetry of all the sub-quivers, resulting in the theory

$$\begin{array}{c}
\mathfrak{u}(n_{k,1}) \relbar\joinrel\relbar \cdots \relbar\joinrel\relbar \mathfrak{u}(n_{k,|\rho_k|-1}) \\
\vert \qquad\qquad\qquad \\
\mathfrak{u}(n_{1,|\rho_1|-1}) \relbar\joinrel\relbar \cdots \relbar \mathfrak{u}(n_{1,1}) \relbar\joinrel\relbar \mathfrak{su}(n)_H \qquad\qquad \\
\vert \qquad\qquad\qquad \\
\mathfrak{u}(n_{2,1}) \relbar\joinrel\relbar \cdots \relbar\joinrel\relbar \mathfrak{u}(n_{2,|\rho_2|-1}),
\end{array} \tag{173}$$

where the $\mathfrak{su}(n)_H$ node is unbalanced, and thus the IR Coulomb 0-form symmetry algebra is $\mathfrak{f} = \bigoplus_i \mathfrak{f}_i$, matching with the flavor symmetry algebra of the Class S theory.

**Global form of the magnetic quiver.** What is the gauge group that we should choose? Apriori there are many choices

$$\frac{\prod_{i=1}^{k} \prod_{J=1}^{|\rho_i|-1} U(n_{i,J}) \times SU(n)_H}{\mathbb{Z}_m} , \tag{174}$$

parametrized by divisors $m$ of $n$. The 1-form symmetry of the theory for such a choice is

$$\Gamma^{(1)} = \mathbb{Z}_{n/m} . \tag{175}$$

To match it with the trivial 1-form symmetry of the Class S theory, we are forced to pick the gauge group

$$\mathcal{G} = \frac{\prod_{i=1}^{k} \prod_{J=1}^{|\rho_i|-1} U(n_{i,J}) \times SU(n)_H}{\mathbb{Z}_n} , \tag{176}$$

for the choice $m = n$.

This global form is actually manifest in the following usual presentation of the MQ

$$
\begin{array}{c}
U(n_{k,1}) \longrightarrow \cdots \longrightarrow U(n_{k,|\rho_k|-1}) \\
| \\
U(n_{1,|\rho_1|-1}) \longrightarrow \cdots \longrightarrow U(n_{1,1}) \longrightarrow U(n)_H \\
| \\
U(n_{2,1}) \longrightarrow \cdots \longrightarrow U(n_{2,|\rho_2|-1}),
\end{array}
\tag{177}
$$

with an additional instruction of ungauging a $U(1)$. If we perform this $U(1)$ ungauging on the central $U(n)_H$ node, we are forced to remove also the $\mathbb{Z}_n$ center of the $SU(n)_H$ component of $U(n)_H$ as this $\mathbb{Z}_n$ sits inside the $U(1)$ center of $U(n)_H$ being removed. Due to the presence of bifundamental matter, this $\mathbb{Z}_n$ also sits inside the $U(1)$ center of all the other unitary gauge groups. Thus, performing the $U(1)$ ungauging at $U(n)_H$ node indeed leaves behind the $\mathcal{G}$ gauge group appearing in (176). Similar discussions also appeared in [77], where a discussion regarding other $U(1)$ ungaugings can also be found (see also [78]).

**Computing flavor symmetry group using the magnetic quiver.** Since, by this simple argument based on 1-form symmetry, we have no other possible choice for the gauge group, it better be true that the IR Coulomb 0-form symmetry group for this choice of gauge group matches the flavor symmetry group of the Class S theory.

This is a straightforward application of the special case of our general prescription described in section 4.5. Recall there we also encountered two types of contributions. The first type of contributions came from unbalanced unitary gauge nodes. Collecting all such contributions from the unbalanced unitary gauge nodes situated along the sub-quiver leg associated to the puncture $\mathcal{P}_i$ provide the contribution $Y_{F,\mathcal{P}_i}$ of [66]. The second type of contribution comes from the sole special unitary unbalanced gauge node, which provides the contribution $\widetilde{Y}_F$ of [66]. In this way, we find that the IR Coulomb 0-form symmetry group of the magnetic quiver (with the correct global form) described above matches the flavor symmetry group $\mathcal{F}$ of the Class S theory.

### 5.1.2 Examples

Let us see this matching explicitly for the following three examples.

**Example 1: Trinion $T_n$.** The first example we consider is the 4d trinion $T_n$ theory, obtained as a Class S theory by compactifying 6d $\mathcal{N} = (2,0)$ SCFT of $A_{n-1}$ type on a sphere with 3 maximal regular punctures. Each puncture provides an $\mathfrak{su}(n)$ flavor symmetry algebra. Thus, the total flavor symmetry algebra is

$$
\mathfrak{f} = \mathfrak{su}(n)_1 \oplus \mathfrak{su}(n)_2 \oplus \mathfrak{su}(n)_3 \,,
\tag{178}
$$

with associated $F = SU(n)_1 \times SU(n)_2 \times SU(n)_3$ with center

$$
Z_F = (\mathbb{Z}_n)_1 \times (\mathbb{Z}_n)_2 \times (\mathbb{Z}_n)_3 \,.
\tag{179}
$$

We assume $n > 3$, because as is well known there is an enhancement of the above flavor symmetry algebra to $\mathfrak{f} = \mathfrak{e}_6$ for $n = 3$, in which case the $T_3$ trinion theory coincides with the $E_6$ Minahan-Nemeschansky theory. This was discussed as the first example in section 4.3 of [66].

The associated magnetic quiver is

$$
\begin{array}{ccccc}
& & \mathfrak{u}(n-1) \longrightarrow \mathfrak{u}(n-2) \longrightarrow \cdots \longrightarrow \mathfrak{u}(1) \\
& & | \\
\mathfrak{u}(1) \longrightarrow \cdots \longrightarrow \mathfrak{u}(n-2) \longrightarrow \mathfrak{u}(n-1) \longrightarrow \mathfrak{su}(n)_H \\
& & | \\
& & \mathfrak{u}(n-1) \longrightarrow \mathfrak{u}(n-2) \longrightarrow \cdots \longrightarrow \mathfrak{u}(1) \, ,
\end{array}
\tag{180}
$$

with gauge group

$$
\mathcal{G} = \frac{\prod_{i=1}^{n-1} U(i)^3 \times SU(n)_H}{\mathbb{Z}_n} \, .
\tag{181}
$$

Let us now compute the IR Coulomb 0-form symmetry group using the arguments of section 4.5. Since every unitary gauge node is balanced, we do not have any puncture dependent contribution i.e. $Y_{F,\mathcal{P}_i} = 0$. On the other hand, the contribution $\widetilde{Y}_F$ is obtained from the monopole operator $O$, whose charge under $Z_F$ is the same as that of the representation

$$
\mathsf{F}_1 \otimes \mathsf{F}_2 \otimes \mathsf{F}_3 \, ,
\tag{182}
$$

of $F$, where $\mathsf{F}_i$ is the fundamental representation of $SU(n)_i \subset F$. This is because the $\mathfrak{su}(n)_H$ node hits each balanced sub-quiver $i$ at the node of Dynkin diagram of $\mathfrak{su}(n)_i$ corresponding to the fundamental representation of $\mathfrak{su}(n)_i$.

These contributions match those appearing in [66], and as described there the flavor symmetry group can written as

$$
\mathcal{F} = \frac{SU(n)_1 \times SU(n)_2 \times SU(n)_3}{\mathbb{Z}_n \times \mathbb{Z}_n} \, .
\tag{183}
$$

We refer the reader to [66] for more details regarding the identity of the two $\mathbb{Z}_n$ subgroups appearing in the denominator.

We can also discuss the case of $n = 3$, for which it was argued in [66] that the full flavor symmetry group must be $E_6/\mathbb{Z}_3$ as that is the only possible enhancement of (183) for the $n = 3$ case. We can see this flavor group directly from the MQ, which is obtained by a $U(1)$ ungauging of

$$
\begin{array}{ccccc}
& & U(2) \longrightarrow U(1) \\
& & | \\
U(1) \longrightarrow U(2) \longrightarrow U(3)_H \longrightarrow U(2) \longrightarrow U(1) \, .
\end{array}
\tag{184}
$$

Instead of performing the $U(1)$ ungauging on the $U(3)_H$ gauge node, let us perform it on one of the $U(1)$ gauge nodes. The magnetic quiver can then be expressed as

$$
\begin{array}{ccccc}
& & U(2) \longrightarrow \mathsf{F} \\
& & | \\
U(1) \longrightarrow U(2) \longrightarrow U(3)_H \longrightarrow U(2) \longrightarrow U(1) \, ,
\end{array}
\tag{185}
$$

where we have a fundamental hyper charged under the top $U(2)$ gauge node. Every unitary gauge node is balanced and hence the IR Coulomb 0-form symmetry algebra is

$$
\mathfrak{f} = \mathfrak{e}_6 \, .
\tag{186}
$$

Since there are no unbalanced unitary or special unitary gauge nodes, the IR Coulomb 0-form symmetry group is the centerless global form

$$\mathcal{F} = E_6/\mathbb{Z}_3, \tag{187}$$

of $\mathfrak{f} = \mathfrak{e}_6$.

**Example 2: Free bifundamental hyper.** As a second example, consider the Class S theory obtained by compactifying 6d $\mathcal{N} = (2,0)$ SCFT of $A_{n-1}$ type on a sphere with 2 maximal regular punctures $\mathcal{P}_1, \mathcal{P}_2$ and 1 minimal regular puncture $\mathcal{P}_3$. Each maximal puncture provides an $\mathfrak{su}(n)$ flavor symmetry algebra, while the minimal puncture provides $\mathfrak{u}(1)$ flavor symmetry algebra. Thus, the total flavor symmetry algebra is

$$\mathfrak{f} = \mathfrak{su}(n)_1 \oplus \mathfrak{su}(n)_2 \oplus \mathfrak{u}(1), \tag{188}$$

with associated $F = SU(n)_1 \times SU(n)_2 \times U(1)$ with center

$$Z_F = (\mathbb{Z}_n)_1 \times (\mathbb{Z}_n)_2 \times U(1). \tag{189}$$

The resulting 4d $\mathcal{N} = 2$ theory can be recognized as a free hypermultiplet transforming in bifundamental representation of two $\mathfrak{su}(n)$ factors of $\mathfrak{f}$, with the $\mathfrak{u}(1)$ factor of $\mathfrak{f}$ rotating the bifundamental hyper. This was discussed as the second example in section 4.1 of [66].

The associated magnetic quiver is

$$\mathfrak{u}(1)$$

$$\mathfrak{u}(1) \longrightarrow \cdots \longrightarrow \mathfrak{u}(n-2) \longrightarrow \mathfrak{u}(n-1) \longrightarrow \mathfrak{su}(n)_H \longrightarrow \mathfrak{u}(n-1) \longrightarrow \mathfrak{u}(n-2) \longrightarrow \cdots \longrightarrow \mathfrak{u}(1), \tag{190}$$

with gauge group

$$\mathcal{G} = \frac{U(1)^3 \times \prod_{i=2}^{n-1} U(i)^2 \times SU(n)_H}{\mathbb{Z}_n}. \tag{191}$$

Let us now compute the IR Coulomb 0-form symmetry group using the arguments of section 4.5. Since every unitary gauge node is balanced in the sub-quivers associated to punctures $\mathcal{P}_1$ and $\mathcal{P}_2$, we have $Y_{F,\mathcal{P}_1} = Y_{F,\mathcal{P}_2} = 0$. On the other hand, the $U(1)$ gauge node comprising the sub-quiver associated to $\mathcal{P}_3$ contributes a genuine local operator of charge $n$ under $U(1)$ factor of $Z_F$ (and charge 0 under each $(\mathbb{Z}_n)_i$ factor). This is precisely the contribution $Y_{F,\mathcal{P}_3}$ of [66].

The contribution $\widetilde{Y}_F$ is obtained from the monopole operator $O$, whose charge under $(\mathbb{Z}_n)_1 \times (\mathbb{Z}_n)_2$ factor of $Z_F$ is the same as that of the representation

$$\mathsf{F}_1 \otimes \mathsf{F}_2, \tag{192}$$

of $SU(n)_1 \times SU(n)_2$ factor of $F$, where $\mathsf{F}_i$ is the fundamental representation of $SU(n)_i \subset F$. This is because the $\mathfrak{su}(n)_H$ node hits each balanced sub-quiver $i \in \{1,2\}$ at the node of Dynkin diagram of $\mathfrak{su}(n)_i$ corresponding to the fundamental representation of $\mathfrak{su}(n)_i$. Moreover, $O$ also has charge +1 under $U(1)$ factor of $Z_F$.

These contributions match those appearing in [66], and as described there the flavor symmetry group can written as

$$\mathcal{F} = \frac{SU(n)_1 \times SU(n)_2 \times U(1)}{\mathbb{Z}_n \times \mathbb{Z}_n}. \tag{193}$$

We refer the reader to [66] for more details regarding the identity of the two $\mathbb{Z}_n$ subgroups appearing in the denominator.

**Example 3.** As a final example, let us consider a Class S theory involving more general regular punctures that are neither maximal nor minimal. We are compactifying $A_3 \, \mathcal{N} = (2, 0)$ theory on a sphere with three punctures. The puncture $\mathcal{P}_1$ has partition $\rho_1 = \{2, 1, 1\}$, the puncture $\mathcal{P}_2$ has partition $\rho_2 = \{2, 2\}$, and the puncture $\mathcal{P}_3$ is a maximal puncture with partition $\rho_3 = \{1, 1, 1, 1\}$. This was discussed as the third example in section 4.1 of [66].

The flavor symmetry algebras associated to the punctures are $\mathfrak{f}_1 = \mathfrak{su}(2)_1 \oplus \mathfrak{u}(1)$, $\mathfrak{f}_2 = \mathfrak{su}(2)_2$ and $\mathfrak{f}_3 = \mathfrak{su}(4)$, with the total flavor symmetry algebra being

$$\mathfrak{f} = \mathfrak{su}(2)_1 \oplus \mathfrak{u}(1) \oplus \mathfrak{su}(2)_2 \oplus \mathfrak{su}(4), \tag{194}$$

with associated $F = SU(2)_1 \times U(1) \times SU(2)_2 \times SU(4)$ with center

$$Z_F = (\mathbb{Z}_2)_1 \times U(1) \times (\mathbb{Z}_2)_2 \times \mathbb{Z}_4. \tag{195}$$

The associated magnetic quiver is

$$
\begin{array}{c}
\mathfrak{u}(2) \\
| \\
\mathfrak{u}(1) \text{ —— } \mathfrak{u}(2) \text{ —— } \mathfrak{su}(4)_H \text{ —— } \mathfrak{u}(3) \text{ —— } \mathfrak{u}(2) \text{ —— } \mathfrak{u}(1),
\end{array}
\tag{196}
$$

with gauge group

$$\mathcal{G} = \frac{U(1)^2 \times U(2)^3 \times U(3) \times SU(4)_H}{\mathbb{Z}_4}. \tag{197}$$

Let us now compute the IR Coulomb 0-form symmetry group using the arguments of section 4.5. Since every unitary gauge node is balanced in the sub-quivers associated to punctures $\mathcal{P}_2$ and $\mathcal{P}_3$, we have $Y_{F,\mathcal{P}_2} = Y_{F,\mathcal{P}_3} = 0$. On the other hand, the unbalanced $\mathfrak{u}(2)$ gauge node in the sub-quiver associated to $\mathcal{P}_1$ contributes a genuine local operator whose charge under $(\mathbb{Z}_2)_1$ factor of $Z_F$ is the same as that of the fundamental representation $\mathsf{F}_1$ of $SU(2)_1$. This is because this unbalanced $\mathfrak{u}(2)$ gauge node hits once the Dynkin diagram of $\mathfrak{su}(2)_1$ formed by the $\mathfrak{u}(1)$ gauge node in the sub-quiver associated to $\mathcal{P}_1$. Moreover, this genuine local operator also has charge 4 under the $U(1)$ factor of $Z_F$ which arises from this unbalanced $\mathfrak{u}(2)$ gauge node.

The contribution $\widetilde{Y}_F$ is obtained from the monopole operator $O$, whose charge under $(\mathbb{Z}_2)_1 \times (\mathbb{Z}_2)_2 \times \mathbb{Z}_4$ factor of $Z_F$ is the same as that of the representation

$$\mathsf{F}_2 \otimes \mathsf{F}, \tag{198}$$

of $SU(2)_1 \times SU(2)_2 \times SU(4)$ factor of $F$, where $\mathsf{F}_2$ is the fundamental representation of $SU(2)_2$ and $\mathsf{F}$ is the fundamental representation of $SU(4)$. This is because the $\mathfrak{su}(4)_H$ node does not hit the Dynkin diagram of $\mathfrak{su}(2)_1$, while hitting the $\mathfrak{su}(2)_2$ and $\mathfrak{su}(4)$ Dynkin diagrams at the nodes corresponding to the fundamental representations of $\mathfrak{su}(2)_2$ and $\mathfrak{su}(4)$. Moreover, $O$ also has charge 2 under $U(1)$ factor of $Z_F$.

Since all charges under $U(1)$ factor of $Z_F$ are even, we can scale them half. The scaled contributions match those appearing in [66], and as described there the flavor symmetry group can written as

$$\mathcal{F} = \frac{SU(2)_1 \times U(1) \times SU(2)_2 \times SU(4)}{\mathbb{Z}_4 \times \mathbb{Z}_2}. \tag{199}$$

We refer the reader to [66] for more details regarding the identity of the $\mathbb{Z}_4$ and $\mathbb{Z}_2$ subgroups appearing in the denominator.

Table 1: Rank 2 5d SCFTs. We list the IR gauge theory description as well as the flavor symmetry algebra. We determine the group structure from both 5d and the corresponding 3d magnetic quivers.

| Model | Gauge Theory | Flavor algebra |
|-------|--------------|----------------|
| 2 | $SU(3)_{1/2} + 9\mathbf{F}$ | $\mathfrak{so}(20)$ |
| 5 | $SU(3)_1 + 8\mathbf{F}$ | $\mathfrak{so}(16) \oplus \mathfrak{su}(2)$ |
| 12 | $SU(3)_0 + 6\mathbf{F}$ | $\mathfrak{su}(6) \oplus \mathfrak{su}(2)^2$ |
| 9 | $SU(3)_{3/2} + 7\mathbf{F}$ | $\mathfrak{so}(14) \oplus \mathfrak{u}(1)$ |
| 26 | $SU(3)_2 + 4\mathbf{F}$ | $\mathfrak{su}(5) \oplus \mathfrak{u}(1)$ |

## 5.2  5d SCFTs

5d superconformal field theories (SCFTs) with 8 supercharges are closely related to 3d $\mathcal{N} = 4$ theories. The proposed correspondence is that the Higgs branch of the 5d SCFT is given by the Coulomb branch of the 3d $\mathcal{N} = 4$ IR SCFT arising from the associated magnetic quiver (MQ) [22, 23], which is again a 3d $\mathcal{N} = 4$ quiver gauge theory. This conjecture has passed numerous non-trivial tests. It can be motivated from the 5d brane-web realization of 5d SCFTs [20, 23, 26, 27, 31, 32], but also via a geometric construction of the 5d theory in M-theory on a canonical singularity: in the case of isolated hypersurface singularities, the MQs for the 5d theories can be derived from the geometry [28, 35, 42].

One salient feature of 5d SCFTs is the flavor symmetry, which often is enhanced compared to the flavor symmetry of an IR gauge theory description obtained in the IR after performing a mass deformation (i.e. moving onto the extended Coulomb branch) of the UV 5d SCFT. The simplest class of such models are the Seiberg $E_{n+1}$ theories having UV flavor symmetry $\mathfrak{e}_{n+1}$, which after a mass deformation give rise to $SU(2) + n\mathbf{F}$ gauge theories in the IR with IR flavor symmetry $\mathfrak{so}(2n) \oplus \mathfrak{u}(1)$ [79].

The flavor symmetry is encoded in terms of the magnetic quiver as well: for simplicity let us consider MQs which are built from $\prod_i U(n_i)$ gauge nodes (along with the additional instruction of a ungauging a $U(1)$ for each connected component of the MQ), connected by bifundamentals such that there is a single bifundamental between any two nodes and the resulting quiver has no loops. Then the balanced unitary nodes give rise to the non-abelian part of the flavor symmetry algebra and the unbalanced nodes give rise to the abelian part of the flavor symmetry algebra. The global form can be obtained using the methods in this paper, specifically section 4. Note that the analysis of that section is applied after making a suitable choices of $U(1)$ ungaugings.

In this section we will compare the global form of the flavor symmetries of the 5d SCFTs and their associated 3d MQ theories. The examples we will focus on are rank 2 theories. A complete list of all MQs for rank 2 theories can be found in [32], using the method of [31]. The flavor symmetry can be determined alternatively from geometry as in [68, 80–87]. The models we consider are shown in table 1 and their magnetic quivers are in table 2. The 5d theories have gauge theory descriptions with $SU(3)$ gauge groups and fundamental flavor and thus no 1-form symmetry (which in principle upon dimensional reduction can contribute to the 0-form symmetry of the MQ theory).

### 5.2.1  Flavor symmetry groups from MQs

To derive the flavor symmetry from the MQs we simply apply the special cases of our general analysis discussed in sections 4.5 and 4.6. The MQs are all listed in table 2. All nodes are unitary: balanced nodes are white, unbalanced ones are black. The magnetic quiver is obtained

Table 2: The Magnetic Quivers for the 5d SCFTs listed in table 1. Each node labeled by $n$ corresponds to a $U(n)$ gauge node, and connection lines to bi-fundamentals. The subgraph given by the white nodes is the Dynkin diagram of the non-abelian part of the flavor symmetry algebra (i.e. the balanced nodes). The black are unbalanced nodes. The MQ is obtained by the ungauging of one of the nodes.

| Model | Magnetic Quiver | Flavor Group |
|:---:|:---:|:---:|
| 2 | $\begin{array}{c} \quad\quad\quad\quad 4 \circ \\ \circ-\circ-\circ-\circ-\circ-\circ-\circ-\circ-\circ-\bullet \\ 1\ \ 2\ \ 3\ \ 4\ \ 5\ \ 6\ \ 7\ \ 8\ \ 5\ \ 2 \end{array}$ | $Ss(20)$ |
| 5 | $\begin{array}{c} \quad\quad\quad 3 \circ \\ \circ-\circ-\circ-\circ-\circ-\circ-\circ-\bullet-\circ \\ 1\ \ 2\ \ 3\ \ 4\ \ 5\ \ 6\ \ 4\ \ 2\ \ 1 \end{array}$ | $\frac{Ss(16)\times SU(2)}{\mathbb{Z}_2}$ |
| 12 | $\begin{array}{c} 1\ \ 2\ \ 1 \\ \circ-\bullet-\circ \\ \quad\ \ \ \ | \\ \circ-\circ-\circ-\circ-\circ \\ 1\ \ 2\ \ 3\ \ 2\ \ 1 \end{array}$ | $\frac{SU(6)/\mathbb{Z}_3\times SO(4)}{\mathbb{Z}_2}$ |
| 9 | $\begin{array}{c} 3\ \ \ 1 \\ \circ-\bullet \\ \circ-\circ-\circ-\circ-\circ\ 5 \\ 1\ \ 2\ \ 3\ \ 4\ \ | \\ \circ-\bullet \\ 3\ \ \ 1 \end{array}$ | $\frac{Spin(14)\times U(1)}{\mathbb{Z}_4}$ |
| 26 | $\begin{array}{c} 1\ \bullet \\ \circ-\circ-\circ-\circ \\ 1\ \ 2\ \ 2\ \ 1 \end{array}$ $\begin{array}{c} 1\ \bullet=\bullet\ 1 \\ \circ-\circ-\circ-\circ \\ 1\ \ 1\ \ 1\ \ 1 \end{array}$ | $U(5)$ |

by ungauging a $U(1)$ in each connected component of the listed quiver. There is a choice in this, and we will pick the ungaugings that allow us to apply the analysis of sections 4.5 and 4.6.

To determine the flavor symmetry group from the magnetic quiver for the cases listed in the table 2, we have to simply follow the following rules:

1. Pick a connected component $\alpha$ of the listed magnetic quiver. Determine the set of balanced nodes, which form a non-abelian Lie algebra $\mathfrak{f}_\alpha$. The number of unbalanced nodes $U_\alpha$ determines an abelian Lie algebra $\mathfrak{u}(1)^{|U_\alpha-1|}$. The Coulomb flavor symmetry algebra $\mathfrak{f}$ of the 3d IR SCFT associated to the full magnetic quiver is obtained by picking the component $\beta$ with the maximal associated algebra, i.e. $\mathfrak{f}=\mathfrak{f}_\beta\oplus\mathfrak{u}(1)^{|U_\beta-1|}$.

2. Ungauge a $\mathfrak{u}(1)$ in the each connected component $\alpha$. This is done at the location of an unbalanced node. If the unbalanced node is $\mathfrak{u}(n)$ for $n>1$, then we are left with a special unitary gauge node $\mathfrak{su}(n)$, and the gauge group associated to the connected

component $\alpha$ is

$$\mathcal{G}_\alpha = \frac{\prod_i U(n_i) \times SU(n)}{\mathbb{Z}_n}. \tag{200}$$

That is the center $\mathbb{Z}_n$ of the numerator is automatically removed in the $U(1)$ ungauging process, as discussed in [77] and after equation (177).

On the other hand, if the unbalanced node is $\mathfrak{u}(1)$, then we are left with a fundamental flavor there and the gauge group associated to the connected component $\alpha$ is

$$\mathcal{G}_\alpha = \prod_i U(n_i). \tag{201}$$

After the ungauging, for the cases appearing in table 2, we are left with either no unbalanced nodes or another unbalanced unitary gauge node.

3. The position of the unbalanced unitary or special unitary node determines the representation under $\mathfrak{f}$ of the gauge monopole operators as described in sections 4.5 and 4.6. We account for monopole operators coming from all connected components. From this the global form of the flavor symmetry group is determined, by quotienting out the part of the flavor center (of the simply connected group associated to $\mathfrak{f}$) that acts trivially on the monopole operators.

We will now determine the global form of the flavor symmetry groups in the examples of table 2.

**Model 2.** The balanced (white) nodes in the magnetic quiver form the Dynkin diagram of

$$\mathfrak{f} = \mathfrak{so}(20). \tag{202}$$

There is no abelian factor in the flavor algebra because there is a single unbalanced node (shown in black). We ungauge the $U(1)$ at the location of this unbalanced node and land in the special case of our general analysis discussed in section 4.5. The gauge group is

$$\mathcal{G} = \frac{\prod_{i=1}^{8} U(i) \times U(4) \times U(5) \times SU(2)}{\mathbb{Z}_2}. \tag{203}$$

The unbalanced special unitary node is attached to the spinor node of $\mathfrak{so}(20)$, and thus we have a monopole operator transforming in the spinor representation[8] of $\mathfrak{so}(20)$. Since we do not have a cospinor representation, we can remove the $\mathbb{Z}_2$ subgroup of the $\mathbb{Z}_2 \times \mathbb{Z}_2$ center of Spin(20), which acts on cospinor representation, but leaves the spinor representation invariant. Thus the flavor symmetry group is

$$\mathcal{F} = \text{Spin}(20)/\mathbb{Z}_2 = Ss(20), \tag{204}$$

where the group $Ss(20)$ is a global form of $\mathfrak{so}(20)$ which admits spinor representation but does not admit cospinor or vector representations of $\mathfrak{so}(20)$.

---

[8]More precisely, we are only claiming that we have a monopole operator transforming in a representation of $\mathfrak{so}(20)$ having the same charges under the center of the simply connected group $Spin(20)$ as the spinor representation of $\mathfrak{so}(20)$. However, for brevity here and in what follows, we will blur this distinction, but the reader should keep this in mind.

**Model 5.** Here the balanced nodes form $\mathfrak{f} = \mathfrak{so}(16) \oplus \mathfrak{su}(2)$. There is no abelian factor in the flavor algebra. After ungauging $U(1)$ at the location of the unbalanced $U(2)$ gauge node, we obtain the MQ whose gauge group is

$$\mathcal{G} = \frac{\prod_{i=1}^{6} U(i) \times U(3) \times U(4) \times SU(2) \times U(1)}{\mathbb{Z}_2}. \tag{205}$$

The unbalanced special unitary node is attached to the spinor node of $\mathfrak{so}(16)$ and the fundamental node of $\mathfrak{su}(2)$, implying that we have a monopole operator transforming in representation $(\mathsf{S}, \mathsf{F})$ of $\mathfrak{so}(16) \oplus \mathfrak{su}(2)$, where $\mathsf{S}$ is spinor representation of $\mathfrak{so}(16)$ and $\mathsf{F}$ is fundamental representation of $\mathfrak{su}(2)$. There is again no monopole operator transforming in the cospinor, so we can reduce Spin(16) to $Ss(16)$. The center of $Ss(16)$ is $\mathbb{Z}_2$ which acts non-trivially on the spinor representation, and the $\mathbb{Z}_2$ center of $SU(2)$ acts non-trivially on the fundamental representation. Thus the $(\mathsf{S}, \mathsf{F})$ monopole operator is left invariant by the diagonal $\mathbb{Z}_2$ and we can express the flavor symmetry group as

$$\mathcal{F} = \frac{Ss(16) \times SU(2)}{\mathbb{Z}_2}. \tag{206}$$

**Model 12.** Similarly for model 12, we have

$$\mathfrak{f} = \mathfrak{su}(6) \oplus \mathfrak{su}(2) \oplus \mathfrak{su}(2). \tag{207}$$

The unbalanced special unitary node (obtained after $U(1)$ ungauging) intersects the Dynkin diagrams of simple components of $\mathfrak{f}$ such that we have a monopole operator in representation $(\Lambda^3, \mathsf{F}, \mathsf{F})$ of $\mathfrak{f}$, where $\Lambda^3$ is the 3-index antisymmetric irreducible representation of dimension 20 of $\mathfrak{su}(6)$. Since only $\Lambda^3$ of $\mathfrak{su}(6)$ appears, we can begin with the smallest global form $SU(6)/\mathbb{Z}_3$ of $\mathfrak{su}(6)$ allowing this representation. The center of $SU(6)/\mathbb{Z}_3$ is $\mathbb{Z}_2$ which acts non-trivially on $\Lambda^3$. Similarly, since we only have the bifundamental $(\mathsf{F}, \mathsf{F})$ of $\mathfrak{su}(2) \oplus \mathfrak{su}(2) = \mathfrak{so}(4)$, we can begin with its smallest global form $SO(4)$ allowing this representation. The center of $SO(4)$ is $\mathbb{Z}_2$ which acts non-trivially on $(\mathsf{F}, \mathsf{F})$. Thus, the diagonal $\mathbb{Z}_2$ of the $\mathbb{Z}_2$ centers of $SU(6)/\mathbb{Z}_3$ and $SO(4)$ acts trivially on $(\Lambda^3, \mathsf{F}, \mathsf{F})$ the monopole, leading to the flavor symmetry group

$$\mathcal{F} = \frac{SU(6)/\mathbb{Z}_3 \times SO(4)}{\mathbb{Z}_2}. \tag{208}$$

**Model 9.** The balanced nodes provide an $\mathfrak{so}(14)$ flavor algebra and the unbalanced nodes provide a $\mathfrak{u}(1)$ flavor algebra, since we have 2 unbalanced nodes. The total flavor algebra is thus

$$\mathfrak{f} = \mathfrak{so}(14) \oplus \mathfrak{u}(1)_C. \tag{209}$$

We choose to ungauge one of the unbalanced $U(1)$ nodes. The gauge group is thus

$$\mathcal{G} = \prod_{i=1}^{5} U(i) \times U(3)^2 \times U(1). \tag{210}$$

We have to now apply the analysis of section 4.6. Since the unbalanced $U(1)$ gauge node is attached to the Dynkin diagram of $\mathfrak{so}(14)$ at the location of co-spinor node, the monopole operator associated to the unbalanced $U(1)$ node transforms in the cospinor irreducible representation $\mathsf{C}$ of $\mathfrak{so}(14)$. Simultaneously, the monopole operator also carries a charge $+1$ under a global form $U(1)_C$ of the $\mathfrak{u}(1)_C$ factor of $\mathfrak{f}$ arising from this unbalanced node. Since, the representation $\mathsf{C}$ has charge $-1$ under the $\mathbb{Z}_4$ center of the simply connected group Spin(14)

associated to $\mathfrak{so}(14)$, the monopole operator discussed above is uncharged under the diagonal combination of the $\mathbb{Z}_4$ center of Spin(14) and the $\mathbb{Z}_4$ subgroup of the group $U(1)_C$. The flavor symmetry group is thus

$$\mathcal{F} = \frac{\mathrm{Spin}(14) \times U(1)}{\mathbb{Z}_4}. \tag{211}$$

**Model 26.** There are two connected components in the MQ corresponding to two branches of the moduli space. The first component gives rise to algebra $\mathfrak{f}_1 = \mathfrak{su}(5)$, while the second component gives rise to a larger algebra $\mathfrak{f}_2 = \mathfrak{su}(5) \oplus \mathfrak{u}(1)_C$. The flavor symmetry algebra is thus

$$\mathfrak{f} = \mathfrak{su}(5) \oplus \mathfrak{u}(1)_C, \tag{212}$$

provided by the second component. Ungauging the unbalanced $U(1)$ gauge node in the first component results in an MQ with balanced unitary gauge nodes only, and thus we do not obtain any monopole operator relevant for the analysis of flavor group. Ungauging an unbalanced $U(1)$ gauge node in the second component leaves behind another unbalanced $U(1)$ gauge node which provides a monopole operator transforming in anti-fundamental representation of $\mathfrak{su}(5)$ along with charge $+1$ under $U(1)_C$, leading to the flavor group

$$\mathcal{F} = \frac{SU(5) \times U(1)_C}{\mathbb{Z}_5} = U(5). \tag{213}$$

### 5.2.2 Flavor symmetry groups from string theory constructions

Reference [68] described a computation of flavor symmetry groups of 5d SCFTs using their string theory constructions. The key physical idea involved is as follows. Study the 5d conformal theory on a flat spacetime with a non-conformal vacuum at infinity. More precisely, one chooses a supersymmetric non-conformal vacuum lying in the Coulomb branch of vacua[9] of the 5d SCFT. The theory now flows and in the IR we obtain a 5d supersymmetric gauge theory with an abelian gauge group $U(1)^r$, where $r$ is referred to as the rank of the 5d SCFT. In addition, we have massive BPS particles[10] charged under $U(1)^r$ and the flavor symmetry. The flavor group is then obtained easily by computing flavor charges of gauge invariant combinations of the above charges, namely those linear combinations which have zero $U(1)^r$ gauge charge.

The charges under $U(1)^r$ can be read off from any string theory construction, but the charges under flavor symmetry require us to use "good" string theory constructions, namely those that manifest the full enhanced flavor symmetry algebra of the 5d SCFT. As is well-known, there are two main kinds of string theory constructions of 5d SCFTs: the first kind involves compactifications of M-theory on Calabi-Yau threefolds while the second kind involves intersecting brane configurations in Type IIB superstring theory. There is always a good M-theory construction, which we will now focus on to compute explicitly the flavor symmetry group of the 5d SCFTs appearing in table 1.

In an M-theory construction, the charges of all BPS particles can be captured by charges of a special set of BPS particles arising from M2 branes wrapping irreducible holomorphic curves in the Calabi-Yau threefold. The gauge/flavor charges are described by intersection numbers

---

[9]It is important not to confuse this with the extended Coulomb branch, which is simply referred to as the Coulomb branch in many studies on 5d SCFTs. The extended Coulomb branch is a space obtained by fibering Coulomb branch of vacua on the base space comprising of a family of theories obtained from the 5d SCFT by performing supersymmetric mass deformations. The Coulomb branch of vacua being referred here is the fiber at the origin (namely the point with zero mass deformations) of the base space of extended Coulomb branch.

[10]One might worry about non-BPS excitations and whether they provide any additional charges that can modify the computation. From the string theory constructions, it is possible to argue that they do not provide any new charges.

of these curves with compact/non-compact divisors. Thus the set of data required for computation of flavor groups is a set $\mathcal{C}$ of holomorphic curves (whose charges span charges of all other curves) along with their intersection numbers with compact and non-compact divisors. Recently, a lot of work has been performed on the computation of such a set $\mathcal{C}$ of curves along with their intersection numbers, and crucially the intersection numbers with non-compact divisors capturing enhanced flavor symmetries. These works have used various geometric techniques involving blow-downs of flat [88, 89] (which built upon the work of [90, 91]) and non-flat [80, 82–84, 92] resolutions of non-minimal elliptic fibrations, along with more general local surface geometry structures [86, 87, 93–99] and a comprehensive description of the associated Calabi-Yau Kähler cone structure [80, 82–84, 100] that interpolates between different resolutions via flop transitions.

We now describe the spanning set $\mathcal{C}$ of curves and the relevant charges of BPS particles associated to these curves, using this information to compute the flavor group for all the models appearing in table 1. Detailed computation of the set $\mathcal{C}$ and its intersection numbers can be found in appendix A.

**Model 2.** One can use any of the above described methods to compute a sufficient spanning set $\mathcal{C}$ of curves and its intersection numbers. One finds that $\mathcal{C}$ contains four curves whose charges are

$$
\begin{aligned}
q(C_1) &= \big(0, 1 | 0 \ (\text{mod } 2), 1 \ (\text{mod } 2)\big), \\
q(C_2) &= \big(-2, 2 | 0 \ (\text{mod } 2), 0 \ (\text{mod } 2)\big), \\
q(C_3) &= \big(2, -1 | 1 \ (\text{mod } 2), 1 \ (\text{mod } 2)\big), \\
q(C_4) &= \big(1, -1 | 1 \ (\text{mod } 2), 1 \ (\text{mod } 2)\big),
\end{aligned}
\tag{214}
$$

where the first two charges are under the $U(1)^2$ gauge group, and the last two charges are under the $\mathbb{Z}_2^{\mathsf{S}} \times \mathbb{Z}_2^{\mathsf{C}}$ center of Spin(20) simply connected group associated to the flavor symmetry algebra $\mathfrak{f} = \mathfrak{so}(20)$, where $\mathbb{Z}_2^{\mathsf{S}}$ is the subgroup of the center under which spinor and vector are charged and $\mathbb{Z}_2^{\mathsf{C}}$ is the subgroup of the center under which co-spinor and vector are charged. From this the reader easily sees that all gauge-invariant linear combinations of these curves are either trivially charged under $\mathbb{Z}_2^{\mathsf{S}} \times \mathbb{Z}_2^{\mathsf{C}}$ or have charge $\big(1 \ (\text{mod } 2), 0 \ (\text{mod } 2)\big)$. Thus we are again led to the flavor group $\mathcal{F} = Ss(20)$.

The above curves $C_i$ have a nice physical interpretation as follows. Perform a mass deformation of the 5d SCFT such that it flows in the IR to 5d $\mathcal{N} = 1$ gauge theory with gauge group $Sp(2)$ and 9 fundamental hypers. Then, the curve $C_1$ gives rise to a BPS instanton of the gauge theory. The curves $C_2$ and $C_3$ give rise to W-bosons of $Sp(2)$ upon moving onto the Coulomb branch of this gauge theory. Finally, the curve $C_4$ gives rise to one of the 9 hypers.

**Model 5.** The analysis is similar to the above case. We again have four curves $C_i$ which have the same physical interpretation after mass deforming to 5d $\mathcal{N} = 1$ gauge theory with gauge group $Sp(2)$ and 8 fundamental hypers. One can use any of the above described methods to compute that these curves have charges

$$
\begin{aligned}
q(C_1) &= \big(0, 1 | 1 \ (\text{mod } 2), 0 \ (\text{mod } 2), 0 \ (\text{mod } 2)\big), \\
q(C_2) &= \big(-2, 2 | 0 \ (\text{mod } 2), 0 \ (\text{mod } 2), 0 \ (\text{mod } 2)\big), \\
q(C_3) &= \big(2, -1 | 0 \ (\text{mod } 2), 0 \ (\text{mod } 2), 1 \ (\text{mod } 2)\big), \\
q(C_4) &= \big(1, -1 | 1 \ (\text{mod } 2), 1 \ (\text{mod } 2), 0 \ (\text{mod } 2)\big),
\end{aligned}
\tag{215}
$$

where the first two charges are under the $U(1)^2$ gauge group, the next two charges are under the $\mathbb{Z}_2^{\mathsf{S}} \times \mathbb{Z}_2^{\mathsf{C}}$ center of Spin(16) simply connected group associated to $\mathfrak{so}(16) \subset \mathfrak{f}$, and the last

charge is under the $\mathbb{Z}_2$ center of $SU(2)$ simply connected group associated to $\mathfrak{su}(2) \subset \mathfrak{f}$. From this the reader easily sees that all gauge-invariant linear combinations of these curves are either trivially charged under $\mathbb{Z}_2^S \times \mathbb{Z}_2^C \times \mathbb{Z}_2$ or have charge $\big(1 \,(\mathrm{mod}\ 2), 0 \,(\mathrm{mod}\ 2), 1 \,(\mathrm{mod}\ 2)\big)$. Thus we are again led to the flavor group described in (206).

**Model 12.** It is again sufficient to consider four curves $C_i$ which have similar physical interpretation as above after mass deforming to 5d $\mathcal{N} = 1$ gauge theory with gauge group $SU(3)$, Chern-Simons level 0, and 6 fundamental hypers. One can use any of the above described methods to compute that these curves have charges

$$
\begin{aligned}
q(C_1) &= \big(0, 0 | 0 \,(\mathrm{mod}\ 6), 0 \,(\mathrm{mod}\ 2), 0 \,(\mathrm{mod}\ 2)\big), \\
q(C_2) &= \big(-1, 2 | 0 \,(\mathrm{mod}\ 6), 0 \,(\mathrm{mod}\ 2), 1 \,(\mathrm{mod}\ 2)\big), \\
q(C_3) &= \big(2, -1 | 0 \,(\mathrm{mod}\ 6), 1 \,(\mathrm{mod}\ 2), 0 \,(\mathrm{mod}\ 2)\big), \\
q(C_4) &= \big(1, -1 | 1 \,(\mathrm{mod}\ 6), 0 \,(\mathrm{mod}\ 2), 0 \,(\mathrm{mod}\ 2)\big),
\end{aligned}
\tag{216}
$$

where the first two charges are under the $U(1)^2$ gauge group, the next charge is under the $\mathbb{Z}_6$ center of $SU(6)$ simply connected group associated to $\mathfrak{su}(6) \subset \mathfrak{f}$, and the last two charges are under the $\mathbb{Z}_2^S \times \mathbb{Z}_2^C$ center of $\mathrm{Spin}(4) = SU(2) \times SU(2)$ simply connected group associated to $\mathfrak{so}(4) = \mathfrak{su}(2) \oplus \mathfrak{su}(2) \subset \mathfrak{f}$. From this the reader can see that all gauge-invariant linear combinations of these curves are either trivially charged under $\mathbb{Z}_6 \times \mathbb{Z}_2^S \times \mathbb{Z}_2^C$ or have charge $\big(3 \,(\mathrm{mod}\ 6), 1 \,(\mathrm{mod}\ 2), 1 \,(\mathrm{mod}\ 2)\big)$. Thus we are again led to the flavor group described in (208).

**Model 9.** It is again sufficient to consider four curves $C_i$ which have similar physical interpretation as above after mass deforming to 5d $\mathcal{N} = 1$ gauge theory with gauge group $Sp(2)$ and 7 fundamental hypers. One can use any of the above described methods to compute that these curves have charges

$$
\begin{aligned}
q(C_1) &= \big(0, 1 | 3 \,(\mathrm{mod}\ 4), 0\big), \\
q(C_2) &= \big(-2, 2 | 0 \,(\mathrm{mod}\ 4), 0\big), \\
q(C_3) &= \big(2, -1 | 0 \,(\mathrm{mod}\ 4), -1\big), \\
q(C_4) &= \big(1, -1 | 2 \,(\mathrm{mod}\ 4), 0\big),
\end{aligned}
\tag{217}
$$

where the first two charges are under the $U(1)^2$ gauge group, the next charge is under the $\mathbb{Z}_4$ center of $Spin(14)$ simply connected group associated to $\mathfrak{so}(14) \subset \mathfrak{f}$, and the last charge is under the $U(1)$ group associated to $\mathfrak{u}(1) \subset \mathfrak{f}$. From this the reader can see that all gauge-invariant linear combinations of these curves are either trivially charged under $\mathbb{Z}_4 \times U(1)$ or have charge a multiple of $\big(1 \,(\mathrm{mod}\ 4), -1\big)$. Thus we are again led to the flavor group described in (211).

**Model 26.** It is again sufficient to consider four curves $C_i$ which have similar physical interpretation as above after mass deforming to 5d $\mathcal{N} = 1$ gauge theory with gauge group $SU(3)$, Chern-Simons level 2 and 7 fundamental hypers. One can use any of the above described methods to compute that these curves have charges

$$
\begin{aligned}
q(C_1) &= \big(-1, 1 | 4 \,(\mathrm{mod}\ 5), 0\big), \\
q(C_2) &= \big(-1, 2 | 1 \,(\mathrm{mod}\ 5), 0\big), \\
q(C_3) &= \big(2, -1 | 0 \,(\mathrm{mod}\ 5), -1\big), \\
q(C_4) &= \big(1, -1 | 1 \,(\mathrm{mod}\ 5), 0\big),
\end{aligned}
\tag{218}
$$

where the first two charges are under the $U(1)^2$ gauge group, the next charge is under the $\mathbb{Z}_5$ center of $SU(5)$ simply connected group associated to $\mathfrak{su}(5) \subset \mathfrak{f}$, and the last charge is under the $U(1)$ group associated to $\mathfrak{u}(1) \subset \mathfrak{f}$. From this the reader can see that all gauge-invariant linear combinations of these curves are either trivially charged under $\mathbb{Z}_5 \times U(1)$ or have charge a multiple of $\big(1 \ (\mathrm{mod}\ 5), -1\big)$. Thus we are again led to the flavor group described in (213).

## 5.3 3d mirror symmetry

As a final consistency check, we can apply our methods to compute and match symmetries and anomalies of two 3d $\mathcal{N} = 4$ gauge theories that are related by 3d mirror symmetry. Let us illustrate this with the general example of 3d $\mathcal{N} = 4$ gauge theory

$$U(m) \text{ --- } [\mathfrak{su}(2m)_H], \tag{219}$$

namely $U(m)$ gauge theory with $2m$ fundamental hypers. The Higgs flavor symmetry algebra is

$$\mathfrak{f}_H = \mathfrak{su}(2m)_H, \tag{220}$$

rotating the fundamental hypers, and the IR Coulomb flavor symmetry algebra is

$$\mathfrak{f}_C^{\mathrm{IR}} = \mathfrak{su}(2)_C, \tag{221}$$

because the $U(m)$ gauge node is balanced. The Higgs 0-form symmetry group is

$$\mathcal{F}_H = PSU(2m)_H, \tag{222}$$

because the full center $\mathbb{Z}_{2m}$ of $SU(2m)_H$ is a subgroup of the $U(1)$ center of the $U(m)$ gauge group. From the analysis of section 4.5, the IR Coulomb 0-form symmetry group is

$$\mathcal{F}_C^{\mathrm{IR}} = SO(3)_H, \tag{223}$$

as there are no unbalanced unitary or special unitary gauge nodes. From the analysis of section 4.5 we also see that there is mixed flavor-gauge monopole operator having winding number 1 around a $U(1)$ subgroup of the maximal torus of $PSU(2m)_H$ and a charge under the $\mathbb{Z}_2$ center of $SU(2)_C$ which is the same as that of the fundamental representation of $SU(2)_C$. This is because the flavor node $[\mathfrak{su}(2m)_H]$ hits the Dynkin diagram of $\mathfrak{su}(2)_C$ at the node corresponding to fundamental representation of $\mathfrak{su}(2)_C$. This translates to a mixed 't Hooft anomaly of the IR SCFT between the Higgs and Coulomb 0-form symmetry groups of the form

$$\mathcal{A}_4^{\mathrm{IR}} = \exp\left( \pi i \int w_2^H \cup w_2^C \right), \tag{224}$$

where $w_2^C$ is the $\mathbb{Z}_2$ valued second Stiefel-Whitney class of the background $SO(3)_C$ bundle and $w_2^H$ is the $\mathbb{Z}_{2m}$ valued obstruction class for lifting background $PSU(2m)_H$ bundles to $SU(2m)_H$ bundles.

Now consider its 3d $\mathcal{N} = 4$ mirror gauge theory [4]

$$[\mathfrak{su}(2)_H]$$
$$|$$
$$U(1) \text{ ---- } U(2) \text{ --- } \cdots \text{ --- } U(m-1) \text{ ---- } U(m) \text{ ---- } U(m-1) \text{ ---- } \cdots \text{ ---- } U(2) \text{ ---- } U(1). \tag{225}$$

The Higgs flavor symmetry algebra is

$$\mathfrak{f}_H = \mathfrak{su}(2)_H, \tag{226}$$

rotating the two fundamental hypers of $U(m)$ gauge group, and the IR Coulomb flavor symmetry algebra is

$$\mathfrak{f}_C^{\text{IR}} = \mathfrak{su}(2m)_C \,, \tag{227}$$

because all the unitary gauge nodes are balanced. The Higgs 0-form symmetry group is

$$\mathcal{F}_H = SO(3)_H \,, \tag{228}$$

because the center $\mathbb{Z}_2$ of $SU(2)_H$ is a subgroup of the $U(1)$ center of each unitary gauge group. From the analysis of section 4.5, the IR Coulomb 0-form symmetry group is

$$\mathcal{F}_C^{\text{IR}} = PSU(2m)_H \,, \tag{229}$$

as there are no unbalanced unitary or special unitary gauge nodes. From the analysis of section 4.5 we also see that there is mixed flavor-gauge monopole operator having winding number 1 around the maximal torus of $SO(3)_H$ and a charge under the $\mathbb{Z}_{2m}$ center of $SU(2m)_C$ which is the same as that of the irreducible representation of $\mathfrak{su}(2m)_C$ whose highest weight has a single non-zero Dynkin coefficient, namely the $m$-th one with $d_m = 1$. This is because the flavor node $[\mathfrak{su}(2)_H]$ hits the Dynkin diagram of $\mathfrak{su}(2m)_C$ at the node corresponding to this representation of $\mathfrak{su}(2m)_C$. This translates to a mixed 't Hooft anomaly of the IR SCFT between the Higgs and Coulomb 0-form symmetry groups of the form

$$\mathcal{A}_4^{\text{IR}} = \exp\left( \pi i \int w_2^C \cup w_2^H \right) , \tag{230}$$

where $w_2^H$ is the $\mathbb{Z}_2$ valued second Stiefel-Whitney class of the background $SO(3)_H$ bundle and $w_2^C$ is the $\mathbb{Z}_{2m}$ valued obstruction class for lifting background $PSU(2m)_C$ bundles to $SU(2m)_C$ bundles.

Thus, we have seen explicitly that the symmetry and anomaly properties of the two mirror theory are the same up to the exchange of labels $C \longleftrightarrow H$.

# 6 Some generalizations

In this section, we discuss a few generalizations of the general considerations of this paper. We will discuss two different types of generalizations:

1. In the first generalization, we will allow ourselves to perform an $\mathcal{N} = 2$ gauging of flavor symmetries of 3d $\mathcal{N} = 4$ theories along with the addition of a Chern-Simons level. We will see that the Chern-Simons level induces a 't Hooft anomaly purely for the 1-form symmetry, which is novel feature that we have not encountered in the $\mathcal{N} = 4$ theories that we studied in this paper. Related models have appeared in [72], motived from the study of $T[M_3]$ compactifications of 6d theories.

2. In the second generalization, we will allow ourselves to perform gaugings of discrete subgroups of flavor symmetries of 3d $\mathcal{N} = 4$ theories. We will see that this opens up the possibility of having non-trivial 2-group symmetries within the context of the study of 3d $\mathcal{N} = 4$ theories involving unitary and special unitary gauge groups. We will also encounter the presence of mixed 't Hooft anomalies between these 2-group symmetries and 0-form symmetries of the 3d $\mathcal{N} = 4$ theory.

## 6.1 $\mathcal{N} = 2$ gauging of T[SU($n$)]

In this subsection we study $\mathcal{N} = 2$ gaugings (possibly with Chern-Simons levels) of $\mathfrak{su}(n)_H$ Higgs flavor symmetry of T[SU($n$)]. We begin with $n = 2$ and later generalize to arbitrary $n$. We find that none of these have a non-trivial 2-group symmetry.

**Symmetries.** Consider the 3d $\mathcal{N} = 2$ theory obtained by an $\mathcal{N} = 2$ gauging of the $\mathfrak{su}(2)_H$ flavor symmetry of the 3d $\mathcal{N} = 4$ theory T[SU(2)] by an $SU(2)_H$ gauge group. In addition, we turn on a Chern-Simons level $k$ for the $SU(2)_H$ gauge group while preserving the $\mathcal{N} = 2$ supersymmetry. Due to this Chern-Simons level, non-fractional gauge monopole operators (which are genuine local operators) start transforming in representations of $SO(3)_H$. For such monopole operators to be gauge invariant, they must arise at the ends of Wilson line defects associated to representations of $SO(3)_H$. However, this clearly does not impact the 1-form symmetry, and we have

$$\Gamma^{(1)} = \mathbb{Z}_2 \,, \tag{231}$$

just as for the case of $\mathcal{N} = 4$ $SU(2)_H$ gauging. The 0-form symmetry is also the same as for the $\mathcal{N} = 4$ gauging

$$\mathcal{F} = SO(3)_C \,. \tag{232}$$

One can argue just as for the case of $\mathcal{N} = 4$ $SU(2)_H$ gauging that $\mathbb{Z}_2$ and $SO(3)_C$ do not combine to form a 2-group symmetry with a non-trivial Postnikov class.

**Purely 1-form anomaly.** Consider instead a fractional gauge monopole operator $O$ labeled by a co-character of $SU(2)_H$ having winding number half around its maximal torus. Due to Chern-Simons level $k$, the operator $O$ transforms in a representation $R$ of $SU(2)_H$ with charge

$$k \ (\text{mod } 2)\,, \tag{233}$$

under the $\mathbb{Z}_2$ center of $SU(2)_H$. For $O$ to be gauge invariant, it must arise at the end of of a Wilson line defect associated to representation $R$. Using the analysis of [12], this fact is equivalent to a 't Hooft anomaly for the 1-form symmetry of the form

$$\mathcal{A}_4^{(1)} = \exp\left( \pi i k \int \frac{\mathcal{P}(B_2)}{2} \right), \tag{234}$$

where $\mathcal{P}(B_2)$ is the Pontryagin square of $B_2$ and is a $\mathbb{Z}_4$ valued class. This class is even on spin manifolds, and one can hence define a $\mathbb{Z}_2$ valued class $\frac{1}{2}\mathcal{P}(B_2)$. The anomaly is this $\mathbb{Z}_2$ valued. It vanishes for $k$ even, but is non-trivial for $k$ odd.

**Mixed 1-form 0-form anomaly.** The fractional gauge monopole operator $O$ also transforms in a representation of $SU(2)_C$ that is not a representation of $SO(3)_C$, just as for the case of $\mathcal{N} = 4$ gauging of T[SU(2)]. This is equivalent to a mixed 't Hooft anomaly between the $\mathbb{Z}_2$ 1-form and $SO(3)_C$ 0-form symmetries, and the full 't Hooft anomaly can be expressed as

$$\mathcal{A}_4 = \exp\left( \pi i \int k \frac{\mathcal{P}(B_2)}{2} + B_2 \cup w_2^C \right). \tag{235}$$

**Generalization to T[SU($n$)].** It is straightforward to generalize to arbitrary $n$. We are studying 3d $\mathcal{N} = 2$ theory obtained by an $\mathcal{N} = 2$ gauging of the $\mathfrak{su}(n)_H$ Higgs flavor symmetry of the 3d $\mathcal{N} = 4$ theory T[SU($n$)] by an $SU(n)_H$ gauge group. In addition, we turn on a Chern-Simons level $k$ for the $SU(n)_H$ gauge group while preserving the $\mathcal{N} = 2$ supersymmetry. The non-fractional monopole operators transform in representations of $PSU(n)_H$ and so the 1-form symmetry is

$$\Gamma^{(1)} = \mathbb{Z}_n \,, \tag{236}$$

just as for the case of $\mathcal{N} = 4$ $SU(n)_H$ gauging. The 0-form symmetry is also the same as for the $\mathcal{N} = 4$ gauging

$$\mathcal{F} = PSU(n)_C \,. \tag{237}$$

One can argue just as for the case of $\mathcal{N} = 4$ $SU(n)_H$ gauging that $\mathbb{Z}_n$ and $PSU(n)_C$ do not combine to form a 2-group symmetry with a non-trivial Postnikov class.

A fractional gauge monopole operator $O$ labeled by a co-character of $SU(n)_H$ having winding number $1/n$ around its maximal torus transforms, due to Chern-Simons level $k$, in a representation $R$ of $SU(n)_H$ with charge

$$k \,(\text{mod } n), \tag{238}$$

under the $\mathbb{Z}_n$ center of $SU(n)_H$, implying a 't Hooft anomaly for the 1-form symmetry of the form

$$\mathcal{A}_4^{(1)} = \exp\left( \frac{2\pi i k}{n} \int \frac{\mathcal{P}_{\sigma(n)}(B_2)}{2} \right), \tag{239}$$

where $\sigma(n) = 0, 1$ depending on whether $n$ is even, odd respectively, and

$$\begin{aligned}
\frac{\mathcal{P}_0(B_2)}{2} &:= \frac{\mathcal{P}(B_2)}{2}, \\
\frac{\mathcal{P}_1(B_2)}{2} &:= B_2 \cup B_2.
\end{aligned} \tag{240}$$

The Pontryagin square $\mathcal{P}(B_2)$ is $\mathbb{Z}_{2n}$ valued and is even for spin manifolds. As a consequence, its half $\mathcal{P}(B_2)/2$ is $\mathbb{Z}_n$ valued. On the other hand, $B_2 \cup B_2$ is naturally $\mathbb{Z}_n$ valued.

The fractional gauge monopole operator $O$ also transforms in a representation of $SU(n)_C$ with charge $-1 \,(\text{mod } n)$ under its $\mathbb{Z}_n$ center, just as for the case of $\mathcal{N} = 4$ gauging of T[$SU(n)$]. This is equivalent to a mixed 't Hooft anomaly between the $\mathbb{Z}_n$ 1-form and $PSU(n)_C$ 0-form symmetries, and the full 't Hooft anomaly can be expressed as

$$\mathcal{A}_4 = \exp\left( \frac{2\pi i}{n} \int k \frac{\mathcal{P}_{\sigma(n)}(B_2)}{2} - B_2 \cup w_2^C \right). \tag{241}$$

## 6.2  T[SU(2)]/$\mathbb{Z}_2^C$ and its gaugings

In this subsection, we study the gauging of a $\mathbb{Z}_2$ subgroup of the $SO(3)_C$ 0-form symmetry of T[SU(2)], which leads to a theory with a non-trivial 2-group symmetry. We also find a mixed anomaly between this 2-group symmetry and the residual Coulomb 0-form symmetry. Finally, we study the $\mathcal{N} = 4$ gauging of $\mathfrak{su}(2)_H$ Higgs flavor symmetry of the theory T[SU(2)]/$\mathbb{Z}_2^C$ obtained after the $\mathbb{Z}_2$ gauging of T[SU(2)].

Let us consider the 3d $\mathcal{N} = 4$ Lagrangian theory

$$U(1) \overset{2}{\underline{\quad\quad}} \left[\mathfrak{su}(2)_H\right], \tag{242}$$

which denotes a theory having a $U(1)$ gauge group along with 2 hypermultiplets of charge 2 that are rotated by an $\mathfrak{su}(2)_H$ Higgs flavor symmetry algebra. This theory can be obtained by gauging the $\mathbb{Z}_2$ subgroup, denoted $\mathbb{Z}_2^C$, of $U(1)_C$ Coulomb 0-form symmetry of the Lagrangian theory (56) discussed earlier.

We are also interested in the 3d $\mathcal{N} = 4$ SCFT that the above Lagrangian theory flows to. We call this SCFT T[SU(2)]/$\mathbb{Z}_2^C$ because it can be obtained by gauging a $\mathbb{Z}_2$ subgroup, denoted $\mathbb{Z}_2^C$, of the $SO(3)_C$ 0-form symmetry of the 3d $\mathcal{N} = 4$ SCFT T[SU(2)]. This is a consequence of the fact that the UV theory (242) is obtained by gauging $\mathbb{Z}_2$ subgroup of $U(1)_C$ 0-form symmetry of the theory (56), and this $U(1)_C$ symmetry embeds as the maximal torus of $SO(3)_C$ 0-form symmetry of the corresponding IR SCFT T[SU(2)].

**1-form symmetry.**     The symmetries and anomalies of the UV theory were discussed in detail in section 7.4 of [12], which we review and use to deduce symmetry and anomalies of the IR SCFT $T[SU(2)]/\mathbb{Z}_2^C$. First of all, there is a

$$\Gamma^{(1)} = \mathbb{Z}_2 \,, \tag{243}$$

1-form symmetry coming from the fact that Wilson lines of odd $U(1)$ charges cannot be screened in the UV theory. This becomes the 1-form symmetry of the IR SCFT. This 1-form symmetry can also be understood as the dual symmetry arising from the perspective of $\mathbb{Z}_2^C$ 0-form gauging.

**Higgs 0-form symmetry.**     The Higgs 0-form symmetry algebra of (242) is

$$\mathfrak{f}_H = \mathfrak{su}(2)_H \,, \tag{244}$$

and the Higgs 0-form symmetry group is

$$\mathcal{F}_H = SO(3)_H \,, \tag{245}$$

because the genuine local operators charged under $\mathfrak{su}(2)_H$ are gauge-invariant combinations of hypermultiplets which all have trivial charge under $\mathbb{Z}_2$ center of $SU(2)_H$. The IR SCFT admits the same Higgs 0-form symmetry group.

**Coulomb 0-form symmetry.**     We label the Coulomb 0-form symmetry group of (242) arising from the $U(1)$ gauge node as

$$\mathcal{F}_C = U(1)'_C = U(1)_C/\mathbb{Z}_2 \,, \tag{246}$$

to distinguish it from the $U(1)_C$ 0-form symmetry of the theory (56).

   The IR SCFT admits the same Coulomb 0-form symmetry group. It can be checked easily using the analysis of [5] that there is no enhancement due to monopole operators. Alternatively, one can understand it from the point of view of gauging $\mathbb{Z}_2^C$ subgroup of $SO(3)_C$ 0-form symmetry group of $T[SU(2)]$. To compute the residual 0-form symmetry after this gauging, we first compute the commutant of $\mathbb{Z}_2^C$ in $SO(3)_C$, which is the maximal torus $U(1)_C$ of $SO(3)_C$, and then we mod out the commutant by $\mathbb{Z}_2^C$ to find that the residual 0-form symmetry is $U(1)'_C$.

**2-group symmetry.**     There is a non-trivial 2-group symmetry formed by $\mathbb{Z}_2$ 1-form symmetry and $SO(3)_H$ 0-form symmetry, with Postnikov class

$$\delta B_2 = w_3^H = \text{Bock}(w_2^H) \,, \tag{247}$$

where $w_3^H$ is the third Stiefel-Whitney class of background $SO(3)_H$ bundles, which can be obtained by applying Bockstein homomorphism associated to the non-split short exact sequence

$$0 \to \mathbb{Z}_2 \to \mathbb{Z}_4 \to \mathbb{Z}_2 \to 0 \,, \tag{248}$$

on the second Stiefel-Whitney class $w_2^H$.

   This 2-group symmetry is a consequence of the fact that even though Wilson line operators of even charge can be screened, the local operators responsible for screening them have different $\mathfrak{su}(2)_H$ representations depending on whether the charge of Wilson line is a multiple of 4 or not. The non-genuine local operators living at the ends of Wilson line operators of charge $4m$ form $SO(3)_H$ representations, while the non-genuine local operators living at the ends of Wilson line operators of charge $4m + 2$ form $\mathfrak{su}(2)_H$ representations that are not allowed representations of $SO(3)_H$.

   The IR SCFT $T[SU(2)]/\mathbb{Z}_2^C$ also carries this 2-group symmetry.

**Mixed 2-group 0-form 't Hooft anomaly.** The structure group of the gauge theory (242) involving the Higgs 0-form symmetry is

$$\mathcal{S} = \frac{U(1) \times SU(2)_H}{\mathbb{Z}_4}, \tag{249}$$

where the $\mathbb{Z}_4$ in the denominator is obtained by combining the $\mathbb{Z}_4$ subgroup of $U(1)$ with the $\mathbb{Z}_2$ center of $SU(2)_H$. The $\mathbb{Z}_4$ in the denominator can actually be identified with the $\mathbb{Z}_4$ group appearing in the short exact sequence (248) appearing in the description of the 2-group symmetry of the theory.

We can thus consider a mixed flavor-gauge monopole operator $O$ associated to a co-character of $\mathcal{S}$ with winding number $1/4$ around $U(1)$ and winding number $1/2$ around the maximal torus of $SU(2)_H$. This monopole operator $O$ is a solitonic defect associated to the 2-group symmetry described above because its obstruction to being lifted to a combination of purely (and non-fractional) gauge and purely flavor monopole operator is captured by the $\mathbb{Z}_4$ group in the denominator of the structure group $\mathcal{S}$ which as remarked above is associated to the 2-group symmetry. See [12] for a more details regarding such solitonic defects.

The monopole operator $O$ has charge $q = 1/4$ under $U(1)'_C$. From the analysis of [12], this fact is equivalent to a mixed 't Hooft anomaly between the 2-group and the Coulomb 0-form symmetry of the form

$$\mathcal{A}_4 = \exp\left( \frac{\pi i}{2} \int B_w^H \cup \Big[ c_1\big(U(1)'_C\big) \,(\text{mod } 4) \Big] \right), \tag{250}$$

where $B_w$ is a $\mathbb{Z}_4$ valued background field associated to the 2-group symmetry comprised out of the $\mathbb{Z}_2$ valued background field $B_2$ for 1-form symmetry and the second Stiefel-Whitney class $w_2^H$ for background $SO(3)_H$ 0-form symmetry bundles, and $c_1\big(U(1)'_C\big)$ is the first Chern class for background $U(1)'_C$ 0-form symmetry bundles.

The above anomaly for (242) descends to an anomaly in the IR SCFT $T[SU(2)]/\mathbb{Z}_2^C$.

**Gauging $SU(2)_H$.** Consider performing $\mathcal{N} = 4$ gauging of $\mathfrak{su}(2)_H$ symmetry of $T[SU(2)]/\mathbb{Z}_2^C$ by an $SU(2)_H$ gauge group

$$T[SU(2)]/\mathbb{Z}_2^C \;\rule[0.5ex]{2em}{0.4pt}\; SU(2)_H. \tag{251}$$

In the $T[SU(2)]/\mathbb{Z}_2^C$ theory we have a line operator $L$ that cannot be screened, but its square $2L$ can be screened such that a non-genuine local operator living at the end of $2L$ transforms in a representation of $SU(2)_H$ that is not a representation of $SO(3)_H$. After gauging $SU(2)_H$, this non-genuine local operator needs to be attached to a $SU(2)_H$ Wilson line in the same representation. Thus, after gauging $SU(2)_H$, even $2L$ is not screened. As a consequence, the 1-form symmetry group is

$$\Gamma^{(1)} = \mathbb{Z}_4. \tag{252}$$

The 2-group background $B_w$ in $T[SU(2)]/\mathbb{Z}_2^C$ can be identified with the 1-form symmetry background $B_2'$ in (251).

The 0-form symmetry group remains $U(1)'_C$, and the mixed 't Hooft anomaly between 2-group and Coulomb 0-form symmetries of $T[SU(2)]/\mathbb{Z}_2^C$ becomes a mixed 't Hooft anomaly between 1-form and 0-form symmetries of (251) of the form

$$\mathcal{A}_4 = \exp\left( \frac{\pi i}{2} \int B_2' \cup \Big[ c_1\big(U(1)'_C\big) \,(\text{mod } 4) \Big] \right). \tag{253}$$

## Acknowledgments

We thank Antoine Bourget, Marcus Sperling, Jingxiang Wu and Zhenghao Zhong for discussions on related topics.

**Funding information**   The work of MB is supported by the EPSRC Early Career Fellowship EP/T004746/1 "Supersymmetric Gauge Theory and Enumerative Geometry", STFC Research Grant ST/T000708/1 "Particles, Fields and Spacetime", and the Simons Collaboration on Global Categorical Symmetries. This work is supported by the European Union's Horizon 2020 Framework through the ERC grants 682608 (LB and SSN) and 787185 (LB). SSN is supported in part by the "Simons Collaboration on Special Holonomy in Geometry, Analysis and Physics" and the EPSRC Open Fellowship EP/X01276X/1.

**Note.**   A paper with related results will appear at the same time in [101]. We thank the authors for coordinating submission.

## A   Geometric computations for 5d SCFTs

In this appendix, we provide more details on the geometric computations leading to the charges $q(C_i)$ of BPS particles arising from curves $C_i$ in Calabi-Yau threefolds involved in the construction of 5d SCFTs appearing in table 1.

All the models we consider can be obtained by decoupling from the following 6d geometry, which is a collision between a $D_{10}$ Kodaira singularity and an $I_1$ smooth fiber, tuned so that the collision results in a non-minimal singularity. The fibration is best described in terms of a so-called Tate model

$$y^2 + b_1 U x y + b_3 U^5 \delta_1^2 = x^3 + b_2 U V x^2 + b_4 U^5 \delta_1 x + b_6 U^{10} \delta_1^4. \tag{A.1}$$

Here $U = 0$ is the $D_{10}$ Kodaira fiber, and $V = 0$ the locus above which the $I_1$ singular fiber is located. We furthermore blowup the locus $U = V = 0$ by inserting a rational curve, and denote the exceptional section of that by $\delta_1$. Non-flat resolution of this model was performed in [82] and the geometry is given in figure 1, which we reproduced from said paper.

The compact surfaces are denoted by $S_i$ and are glued along $U = 0$, which is a degree $(-2, 0)$ curve. The numbers in the bracket indicate the self-intersection of the curve in the divisor. The geometry shown is that of the marginal theory, i.e. the 6d parent SCFT (on the tensor branch, which is modeled by the curve $\delta_1$) on a circle. Only once we start decoupling matter (which geometrically corresponds to performing blowdowns), do we get a theory that is a genuinely 5d SCFT fixed point.

The flavor symmetry is obtained from the geometry as follows: The non-abelian flavor symmetry algebra is obtained from the intersection of compact $S_a$ and non-compact $\mathbf{N}_i$ divisors. The complete intersection curves $S_a \cdot \mathbf{N}_i$ will have a normal bundle degree, which if it is $(-2, 0)$ will indicate that the non-compact divisors are ruled by these curves and correspond to the Cartans of the flavor symmetry algebra (for a more refined discussion see [82]). On the other hand, curves that have normal bundle $(-1, -1)$ are inside the compact divisors, and correspond to hypermultiplets in any associated 5d $\mathcal{N} = 1$ gauge theory description, e.g. with gauge group $Sp(2)$ and 10 fundamental hypers.

The intersection pattern of the $(-2, 0)$ curves gives the Dynkin diagram of the flavor symmetry algebra, however in order to determine the flavor symmetry group, we need to also determine the charges of the hypers under the flavor and gauge symmetry. Additionally, we

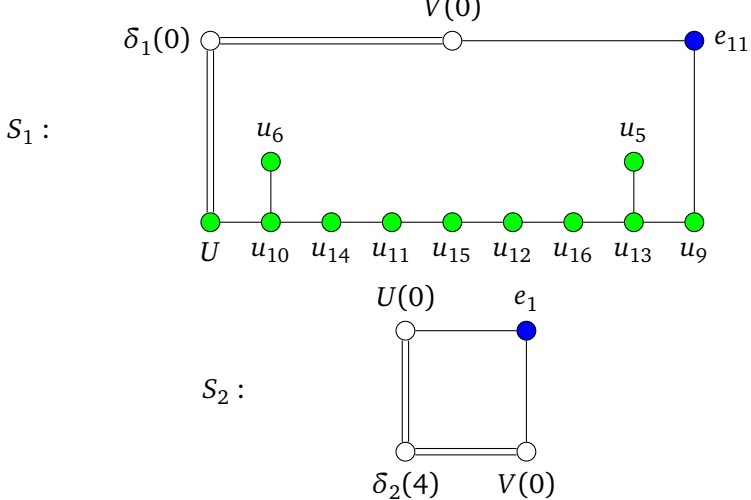

Figure 1: The surface geometry for the marginal theory of type $D_{10} - I_1$, which gives rise to the rank 2 5d SCFTs discussed in this section (the labels for the curves is chosen in accord with the derivation of the geometry in [82]). The two compact surfaces are denoted by $S_i$ and the collection of rational curves and their intersections are shown in the figure. The self-intersection of the curves in each surface is either shown next to the sections $u_i$, $U, V$ $\delta_i$, or is $-2$ for green and $-1$ for blue curves.

need to determine the charges of W-bosons and instantons under the flavor and gauge symmetry. A convenient way of doing so is to convert the resolution information present in figure 1 into a local surface geometry describing explicitly the various compact and non-compact divisors present in the Calabi-Yau along with the intersections of these divisors.

The surface geometry associated to figure 1, i.e. the KK-theory, is

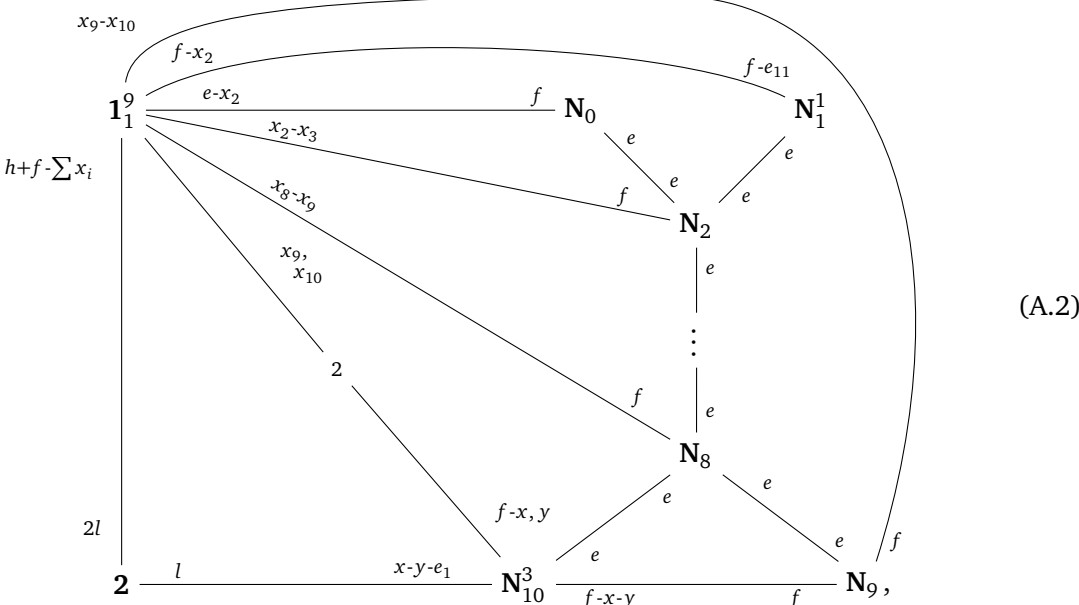

$$(A.2)$$

where we used the notation $e_1$ and $e_{11}$ for the $(-1)$ curves that can be flopped to decouple hyper-multiplets – in accordance with the notation in figure 1. The figure shows the compact surfaces $S_i$ having been represented as Hirzebruch surfaces $\mathbf{i}_d^b$ where the subscript $d$ is the degree of the Hirzebruch surface and the superscript $b$ is the number of additional blowups

performed on it. An edge between two surfaces denotes an intersection between the two surfaces and the labels on the edges describe the curves in the two surfaces that are glued at the locus of the intersection. Multiple edges are denoted by a number between the edge. A curve $e$ inside a surface denotes the base of the corresponding $\mathbb{P}^1$ fibration. This is a compact curve for the Hirzebruch surfaces and a non-compact curve for the non-compact surfaces $\mathbf{N}_i$. A curve $f$ inside a surface denotes the fiber of the corresponding $\mathbb{P}^1$ fibration. This is a compact curve for both compact and non-compact surfaces. The curves $x_i, x, y$ denote various blowups, and we finally we have defined the curve $h := e + df$. Moving forward we denote $e$ curve of surface $S_i$ as $e_i$ and $f$ curve of surface $S_i$ as $f_i$.

After a single flop (of the curve $e_{11}$) the surface geometry can be expressed as

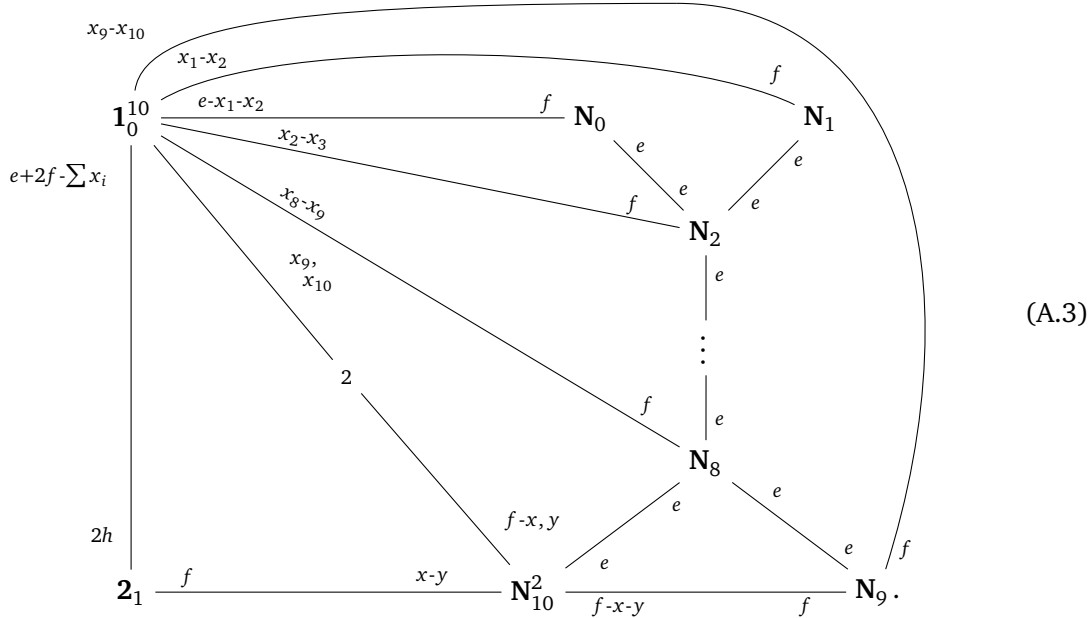

$$\text{(A.3)}$$

The non-compact surfaces $\mathbf{N}_i$ form the Dynkin diagram of the affine Lie algebra $\mathfrak{so}(20)^{(1)}$ which is associated to the fact that this is actually a 6d SCFT with $\mathfrak{so}(20)$ flavor symmetry compactified on a circle.

**Model 2.** This model is obtained by blowing down the curve $f_1 - x_1$. This means that we first flop $f_1 - x_1$ and then taking the volume of the flopped curve infinity. This effectively decouples a hypermultiplet. The effect of this blowdown is that a $\mathbb{P}^1$ fibered non-compact surface intersecting $f_1 - x_1$ does not remain $\mathbb{P}^1$ fibered anymore. In (A.3), $\mathbf{N}_1$ is the only such non-compact surface since

$$(f_1 - x_1) \cdot \mathbf{N}_1 = (f_1 - x_1) \cdot_{S_1} (x_1 - x_2) = 1, \tag{A.4}$$

where the intersection number $(f_1 - x_1) \cdot \mathbf{N}_1$ is in the Calabi-Yau threefold and the intersection number $(f_1 - x_1) \cdot_{S_1} (x_1 - x_2)$ is in the surface $S_1$. The above intersection number is a consequence of the facts that $f_1 \cdot_{S_1} x_i = x_j \cdot_{S_1} x_i = 0$ and $x_i \cdot_{S_1} x_i = -1$. Later we will also use $e_i \cdot_{S_i} e_i = -d_i$, where $d_i$ is the degree of the Hirzebruch surface $S_i$.

After the blowdown we obtain the surface geometry

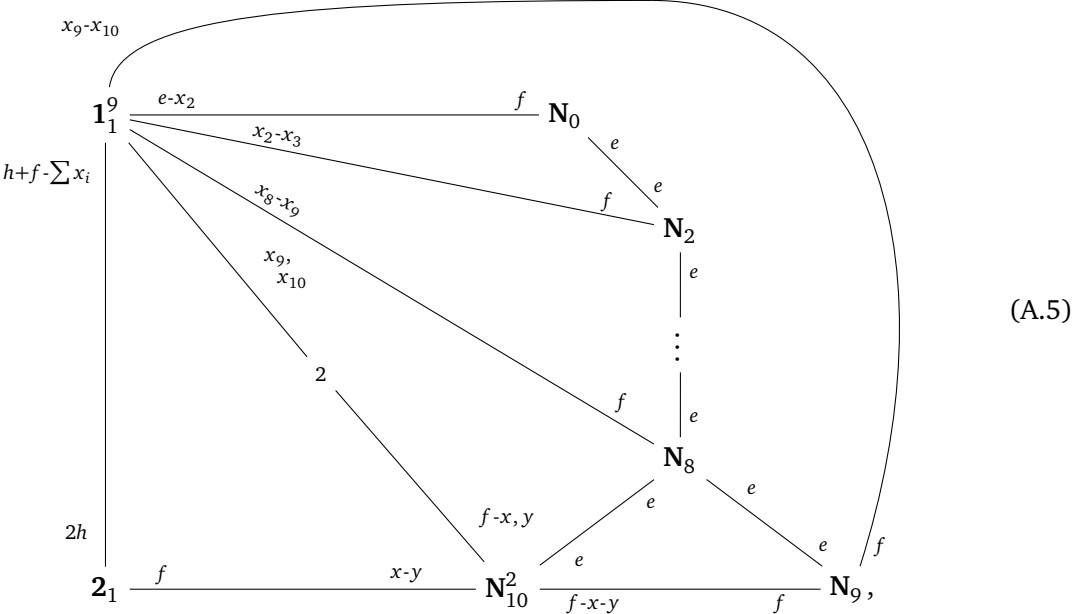

$$(A.5)$$

where we have only kept the $\mathbb{P}^1$ fibered non-compact surfaces. Taking the gluings into account, we see that all compact curves are (possibly non-positive) linear combinations of the curves $e_i$, $f_i$ and the blowups $x_i$. Additionally due to the gluing between $S_1$ and $S_2$, we can express $e_1$ as a linear combination of[11] the $e_2$, $f_i$ and $x_i$.

We have the identifications

$$C_1 = e_2, \qquad C_2 = f_2, \qquad C_3 = f_1, \qquad C_4 = x_i, \qquad (A.6)$$

where we can choose any $x_i$ because the charges under gauge and flavor centers are the same for all $x_i$.

Let us now compute the charges. The charge $q_i(C)$ of a compact curve $C$ under $U(1)$ gauge group associated to $S_i$ is computed as

$$q_i(C) = -C \cdot S_i, \qquad (A.7)$$

which for a genus zero curve is $2 + C \cdot_{S_i} C$ if $C$ lives in $S_i$. This implies

$$q_2(C_1) = 1, \qquad q_2(C_2) = 2, \qquad q_1(C_3) = 2, \qquad q_1(C_4) = 1. \qquad (A.8)$$

On the other hand, if $C$ is in surface $S_j$ then $C \cdot S_i$ is computed as $C \cdot_{S_j} C_{j,i}$ where $C_{j,i}$ is the gluing curve in $S_j$ to $S_i$. This implies

$$q_1(C_1) = 0, \qquad q_1(C_2) = -2, \qquad q_2(C_3) = -1, \qquad q_2(C_4) = -1. \qquad (A.9)$$

Additionally, we have

$$-C_1 \cdot \mathbf{N}_i = \delta_{i,10}. \qquad (A.10)$$

That is, $C_1$ intersects the non-compact surfaces along the cospinor node $\mathbf{N}_{10}$ and hence it has the same charges under the center of $Spin(20)$ as the cospinor representation of $Spin(20)$. Similarly, since $C_2$ has no non-trivial intersections with any $\mathbf{N}_i$, it does not contribute any non-trivial charge under the center of $Spin(20)$. On the other hand, we have

$$-C_3 \cdot \mathbf{N}_i = \delta_{i,0}, \qquad (A.11)$$

---

[11]Note that the labels 1 and 2 are not interchangeable here. We cannot write $e_2$ in terms of $e_1$, $f_i$ and $x_i$.

implying that it has the same charges under the center of $Spin(20)$ as the vector representation of $Spin(20)$. Finally, for any $x_i$ the reader can compute that it has the same charges under the center of $Spin(20)$ as the vector representation of $Spin(20)$. For example choosing $C_4 = x_9$, we have

$$-C_4 \cdot \mathbf{N}_i = \delta_{i,9} + \delta_{i,10} \,, \tag{A.12}$$

which means it transforms both under $\mathbb{Z}_2^{\mathsf{S}}$ and $\mathbb{Z}_2^{\mathsf{C}}$ subgroups of the center of $Spin(20)$. This reproduces the charges claimed in (214).

**Model 5.** Its surface geometry can be obtained from the surface geometry for model 2 by blowing down the curve $f - x_2$, resulting in the surface geometry

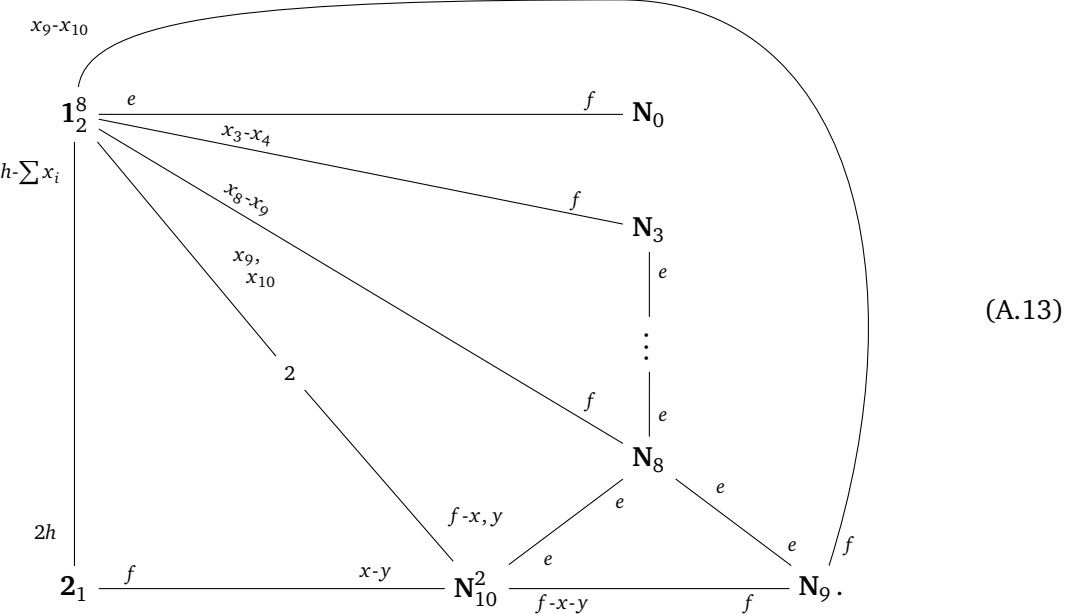

$$\tag{A.13}$$

Now $\mathbf{N}_0$ generates the $\mathfrak{su}(2) \subset \mathfrak{f}$ and the other $\mathbf{N}_i$ generate the $\mathfrak{so}(16) \subset \mathfrak{f}$.

We have the same identifications as in (A.6). The gauge charges are the same as for model 2. The flavor center charge of $C_1$ is the same as spinor of $Spin(16)$ as it intersects the spinor node $\mathbf{N}_{10}$. The flavor center charge of $C_2$ is trivial as it does not intersect any $\mathbf{N}_i$. The flavor center charge of $C_3$ is the same as fundamental of $SU(2)$ as it intersects $\mathbf{N}_0$. Finally, the flavor center charge of $C_4$ is the same for all $x_i$ and can be easily seen to be that for vector of $Spin(16)$ by choosing $C_4 = x_3$.

**Model 9.** Its surface geometry can be obtained from the surface geometry for model 5 by blowing down the curve $f - x_3$, resulting in the surface geometry

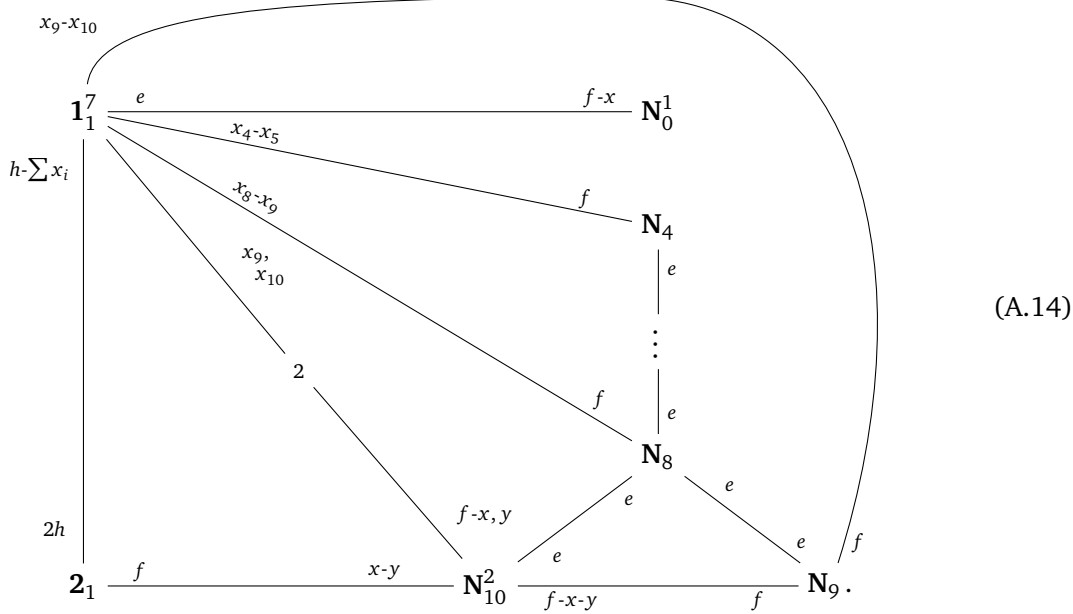

$$(A.14)$$

The blowdown process causes $N_3$ and $N_0$ to become non-$\mathbb{P}^1$ fibered non-compact surfaces. $N_3$ intersects the $\mathbb{P}^1$ fibers of remaining $\mathbb{P}^1$ fibered non-compact surfaces, and hence cannot give rise to an independent $\mathfrak{u}(1)$ flavor. However, $N_0$ does not intersect the $\mathbb{P}^1$ fibers of remaining $\mathbb{P}^1$ fibered non-compact surfaces and generates a $\mathfrak{u}(1)$ factor in the flavor symmetry algebra $\mathfrak{f}$. It should be noted that the curves $f$ and $x$ of $N_0$ both have infinite volume, but their difference $f - x$ has finite volume. The other $N_i$ shown above generate the $\mathfrak{so}(14) \subset \mathfrak{f}$.

We have the same identifications as in (A.6). The gauge charges are the same as for the previous models. The flavor center charge of $C_1$ is the same as cospinor of $Spin(14)$ as it intersects the cospinor node $N_{10}$. The flavor center charge of $C_2$ is trivial as it does not intersect any $N_i$. The curve $C_3$ carries charge $-1$ under $U(1)$ flavor as it intersects $N_0$. Finally, the flavor center charge of $C_4$ is the same for all $x_i$ and can be easily seen to be that for vector of $Spin(14)$ by choosing $C_4 = x_4$.

**Model 12.** Its surface geometry can be obtained from the surface geometry for model 5 by blowing down curves $x_9, x_{10}$, resulting in the surface geometry

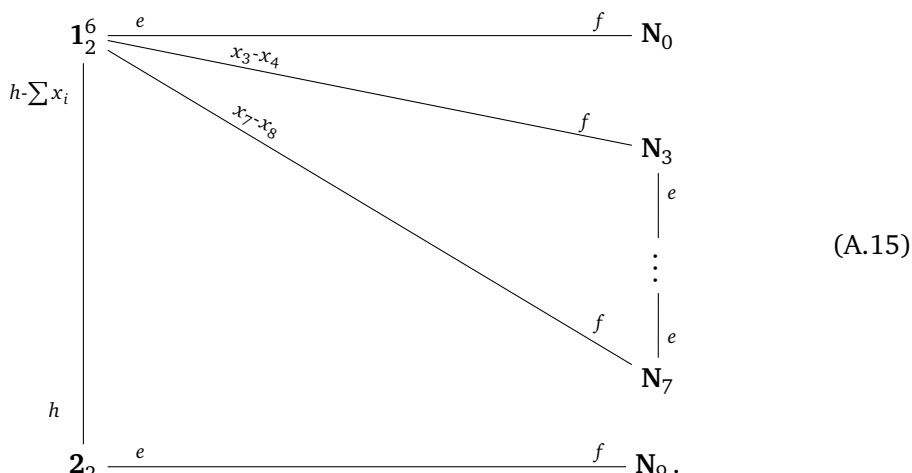

$$(A.15)$$

We have the same identifications as in (A.6). The gauge charges are modified for curves living in $S_2$. The flavor center charge of $C_1$ is trivial as its intersection number $\mathbf{N}_9$ is even. The flavor center charge of $C_2$ is the same as fundamental of $SU(2)$ corresponding to $\mathbf{N}_9$ as the intersection number of $C_2$ with $\mathbf{N}_9$ is odd. Similarly, The flavor center charge of $C_3$ is the same as fundamental of $SU(2)$ corresponding to $\mathbf{N}_0$. Finally, the flavor center charge of $C_4$ is the same for all $x_i$ and can be easily seen to be that of fundamental of $SU(6)$ by choosing $C_4 = x_3$.

**Model 26.** Its surface geometry can be obtained from the surface geometry for model 9 by performing some blowdowns of both types of curves $f - x_i$ and $x_i$, resulting in the surface geometry

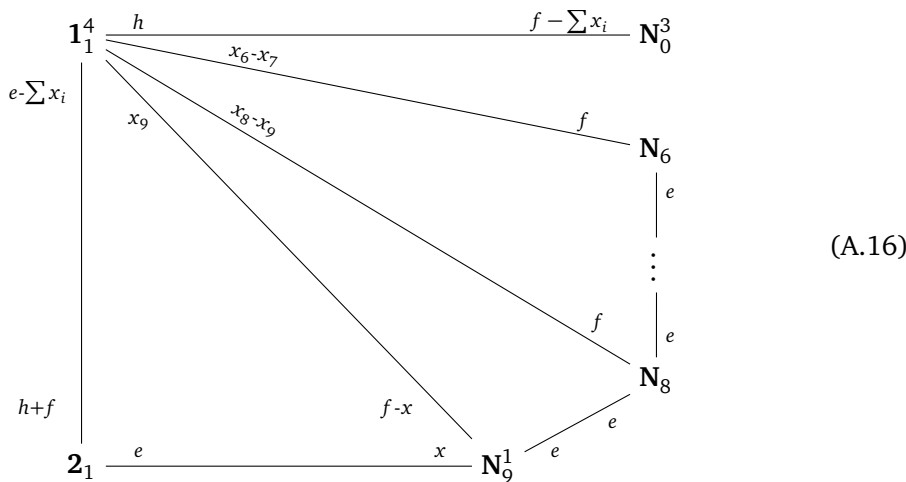

$$\text{(A.16)}$$

$\mathbf{N}_0$ generates a $\mathfrak{u}(1)$ factor in the flavor symmetry algebra $\mathfrak{f}$. Its curves $f$ and $x_i$ are non-compact, but $f - \sum x_i$ is compact. The other $\mathbf{N}_i$ shown above generate the $\mathfrak{su}(5) \subset \mathfrak{f}$.

We have the same identifications as in (A.6). The gauge charges are computed as above. The flavor center charge of $C_1$ is the same as anti-fundamental of $SU(5)$ as it has intersection $-1$ with the anti-fundamental node $\mathbf{N}_9$. The flavor center charge of $C_2$ is the same as fundamental of $SU(5)$ as it has intersection $+1$ with the anti-fundamental node $\mathbf{N}_9$. The curve $C_3$ carries charge $-1$ under $U(1)$ flavor as it intersects $\mathbf{N}_0$. Finally, the flavor center charge of $C_4$ is the same for all $x_i$ and can be easily seen to be that for fundamental of $SU(5)$ by choosing $C_4 = x_6$.

More generally for all descendants of the marginal geometry figure 1, we can determine the flavor symmetry group following similar reasoning. See tables in appendix A of [82] for the flavor algebras.

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
