# Peer review of "Generalized Symmetries and Anomalies of 3d N=4 SCFTs"

_SciPost Physics, doi:SciPost Phys. 16, 080 (2024)_

## Round 2 · Referee Report · Anonymous (Referee 1) · 2023-10-8

Report

There are additional grammatical typos:

Requested changes

1- At the beginning of p.18: The symmetries are summarises as follows $\to$ The symmetries are summarised as follows 2- At the end of p.31: transforming in same representations $\to$ transforming in \emph{the} same representations. Even below there are many places that miss \emph{the} in front of same 3- Above (3.67): is is $\to$ is 4- Below (6.21): this non-genuine local operator needs to attached to $\to$ this non-genuine local operator needs to \emph{be} attached to 5- In p.66: The intersection pattern of the $(-2,0)$ curves give $\to$ The intersection pattern of the $(-2,0)$ curves gives 6- Above (A.4): The affect of this blowdown $\to$ The effect of this blowdown

---

## Round 2 · Referee Report · Anonymous (Referee 1) · 2023-10-8

Strengths

The paper is a well-structured, detailed, and strong contribution to the study of generalized symmetries and 't Hooft anomalies in 3d \( \mathcal{N}=4 \) SCFTs. The technical depth, combined with solid consistency checks, makes it an important work in the area. After the following points are addressed, I believe the paper would be a good fit for publication in SciPost.

Report

The manuscript presents a thorough study of generalized global symmetries and their 't Hooft anomalies in 3d \(\mathcal{N}=4\) superconformal field theories (SCFTs). An essential feature of this study is the method used to deduce generalized symmetries and their anomalies for the Coulomb branch from the UV Lagrangian. The primary focus of the study is on good quiver gauge theories with (special) unitary gauge groups. As a concrete example, the T[SU(\(n\))] theories are first studied from this viewpoint in great details. Then, more general quiver theories are considered. The authors make their results even more concrete by presenting various consistency checks, using magnetic quiver techniques.

Requested changes

1- The paper \cite{Benini:2018reh} is one of the pioneering papers which provide a detailed examination of 2-groups and their 't Hooft anomalies. It is important to cite this work.

2- The hat notation appears first in Eqn.(2.10), which is supposed to indicate the Pontryagin dual. This notation should be mentioned.

3- In the first line of the first bullet in p.11, $\lambda_j=0$ is supposed to be $m_j=0$.

4- Below Eqn.(4.11), oft-called $\to$ so-called

5- Above Eqn.(4.13): "The charge $q_i$ of $O$ under $U(1)_i$ for $i\in \mathcal {U}$ is $n_i$" $\to$ "The charge of $O$ under $U(1)_i$ for $i\in \mathcal {U}$ is $q_i$". Anyhow, the use of $n_i$ should be avoided since it is used as a rank of a unitary group right below.

6- Eqn.(4.13) gives the definition of Dynkin coefficients for topological symmetry $\mathfrak{f}_a$. It is unclear why it involves the summation over unbalanced unitary nodes in $\mathcal{U}$. The reason needs to be elaborated. If it is a typo, it should be corrected.

Attachment

---

## Round 2 · Referee Report · Anonymous (Referee 2) · 2023-10-28

Report

This article serves as a useful reference on generalized global symmetries and 't Hooft anomalies in 3d $\mathcal{N}=4$ SCFTs. It contains many explicit results with clear explanations. The referee sees that the discussion on the $U(1)$ gauge theory with hypermultiplets of charge $q$ has been improved significantly from the first version on the arXiv.  In the referee's opinion, it deserves publication after minor correction suggested by the other referees.

---

## Round 3 · Referee Report · Anonymous (Referee 2) · 2024-1-29

Report

The authors have modified the manuscript according to the referees' comments satisfactorily. The article is recommended to be published in SciPost.

---

## Round 3 · Referee Report · Anonymous (Referee 1) · 2024-2-7

Report

All the points raised by the referees were addressed. I would recommend the manuscript for the publication.

---

## Round 3 · Author Response

We thank very much the referees for their useful comments. All of their comments have been addressed.

To report 1:
Point 6: This is because the U(1) topological symmetries associated to unbalanced nodes neighbouring the balanced nodes contribute to the cartan of the non-abelian IR topological symmetry associated with the balanced nodes. This is a well-known phenomenon, and follows straightforwardly from the analysis of Gaiotto and Witten. A simple example of this phenomenon is exhibited already in eq (3.37).

---

## Round 3 · List of Changes

The changes suggested by the referees have been performed. The reference \cite{Benini:2018reh} has been added at the beginning of section 2.2.3

---

## Editorial Decision

published